# KnowMT-Bench: Benchmarking Knowledge-Intensive Long-Form Question Answering in Multi-Turn Dialogues

## Abstract

Multi-Turn Long-Form Question Answering (MT-LFQA) is a key application paradigm of Large Language Models (LLMs) in knowledge-intensive domains. However, existing benchmarks are limited to single-turn dialogue, while multi-turn dialogue benchmarks typically assess other orthogonal capabilities rather than knowledge-intensive factuality. To bridge this critical gap, we introduce **KnowMT-Bench**, the *first* benchmark designed to systematically evaluate MT-LFQA for LLMs across knowledge-intensive fields, including medicine, finance, and law. To faithfully assess the model's real-world performance, KnowMT-Bench employs a dynamic evaluation setting where models generate their own multi-turn dialogue histories given logically progressive question sequences. The factual capability and information delivery efficiency of the *final-turn* answer are then evaluated via a human-validated automated pipeline. Our experiments on a diverse suite of LLMs show a clear degradation in both factual capability and information delivery efficiency within multi-turn contexts. We further probe the underlying causes and find that contextual noise, particularly relevant misinformation, along with increasing context length and the structure of the dialogues, substantially contributes to this degradation. In addition, experimental results in mitigation strategies demonstrate that structural context refinement and RAG can effectively alleviate these issues, with RAG notably capable of reversing this performance degradation. These findings underscore the importance of our benchmark for evaluating and enhancing LLMs' conversational factual capabilities in real-world applications. Code and data is available at KnowMT-Bench.

## 1 Introduction

Large Language Models (LLMs) are increasingly being used in highly specialized domains such as medicine, finance, and law, partially replacing costly expert consultations and significantly lowering the barrier to accessing professional knowledge (Wu et al., 2023; Huang et al., 2023; Singhal et al., 2025). In particular, real-world consultations are often progressive and complex, requiring multi-turn dialogues to pinpoint a user's core needs and then delivering a detailed long-form answer that synthesizes information across multiple key points (Kurtz & Silverman, 1996; CFP Board, 2020). Building on these observations, we formalize such a challenge as Multi-Turn Long-Form Question Answering (MT-LFQA): an open-domain QA task that requires the model to synthesize multiple facts into a paragraph-level answer for the final-turn question, given the context of dialogue history.

As these specialized domains are inherently knowledge-intensive and often high-stakes, the answers provided must be factually comprehensive and accurate, while exhibiting minimal factual hallucination. While numerous single-turn Long-Form Question Answering (LFQA) benchmarks have emerged, such as K-QA in medicine (Manes et al., 2024), FinTextQA in finance (Chen et al., 2024b), and cLegal-QA in law (Wang et al., 2025), the challenges are substantially amplified in a multi-turn context. In MT-LFQA, the dialogue history can introduce redundant information, which acts as noise to compromise the model's ability to generate a long-form answer adhering to these standards (Laban et al., 2025). As demonstrated in Figure 1, the model that produces factually sound answers in the single-turn setting generates a significant factual error within the multi-turn context. Concurrently, the volume of non-factual content increases, obscuring key information, which degrades the overall utility of the answer (Zhou & Shen, 2024; Hackenburg et al., 2025). Therefore, the single-turn setup of existing LFQA benchmarks cannot faithfully assess a model's performance in the more challenging MT-LFQA scenario.

Figure 1: Illustration of Single-Turn vs. Multi-Turn LFQA on Llama-3.3-70B-Instruct, with correct facts , irrelevant statements , and factual hallucinations .

Existing multi-turn benchmarks are also misaligned with the specific challenges of MT-LFQA. First, conventional conversational QA benchmarks such as QuAC (Choi et al., 2018), CoQA (Reddy et al., 2019) are designed for short, often extractive, answers, making them unable to effectively assess the integration of multiple facts into a paragraph-level answer (see Table 1 for a detailed comparison of the QA benchmark). Second, contemporary dialogue benchmarks for LLMs also prove unsuitable, as they either adopt evaluation paradigms like LLM-as-a-judge (Zheng et al., 2023; Bai et al., 2024), which are inadequate for rigorous factuality assessment by relying on the judge's own fallible parametric knowledge (Fu et al., 2023; Chen et al., 2024a), or they prioritize orthogonal capabilities like instruction-following (He et al., 2024), fairness (Fan et al., 2024), thereby diluting the focus on core LFQA competences. This clear gap necessitates a purpose-built benchmark to systematically measure fact capability and information delivery efficiency within MT-LFQA.

To bridge this critical gap, we introduce **KnowMT-Bench**, the first benchmark designed to conduct a systematic study of MT-LFQA. We ground our research in medicine, finance, and law, which are the common domains for specialized consultation. Our benchmark is thus founded on 801 evidence-grounded LFQA instances from these domains. To simulate a realistic human-LLMs interaction, the benchmark requires models to generate their own dialogue history following logically progressive human-authored question sequences. The final-turn answers are assessed using a comprehensive framework that leverages an automated fine-grained, Natural Language Inference (NLI)-based pipeline inspired by previous works (Manes et al., 2024; Jeong et al., 2024), to analyze factual capability and information delivery efficiency. The reliability of the automated pipeline is supported by validating each step with human experts.

Our experiments on a diverse suite of LLMs reveal that multi-turn contexts pose a severe challenge: model factual capability shows a pronounced degradation when shifting from single-turn to multi-turn LFQA, accompanied by a significant increase in verbosity that reduces information delivery efficiency (Section 4.2). Our analysis further reveals that this decline is driven by the contextual noise, particularly relevant misinformation, along with increasing context length and the structure of the dialogues (Section 4.3). In addition, we explore some mitigation strategies and demonstrate that structural context refinement and RAG can effectively alleviate these issues, with RAG notably capable of reversing this performance degradation (Section 5). These findings highlight the limitations of single-turn evaluations and underscore the necessity of our benchmark for assessing and improving the conversational robustness of LLMs under real-world knowledge-intensive applications.

In summary, our main contributions are as follows: (1) We introduce **KnowMT-Bench**, the first benchmark designed for the systematic evaluation of LLMs in MT-LFQA. (2) We design and validate a comprehensive **evaluation framework** for MT-LFQA, employing a human-validated, automated pipeline to assess both **factual capability** and **information delivery efficiency**. (3) Our experimental results reveal a pronounced degradation in both model factual capability and information delivery efficiency within the multi-turn contexts. Crucially, we identify that the decline is primarily attributable to the combination of contextual noise, context length and the structure of the dialogues, and demonstrate that structural context refinement and RAG can serve as effective methods of mitigating this performance degradation.

## 2 TASK DEFINITION

As an early systematic study of MT-LFQA, we begin by formalizing this task, introducing notation to facilitate our analysis, and delimiting the scope of evaluation. We first formalize the single-turn LFQA task and then extend it to the multi-turn setting, which is central to this work.

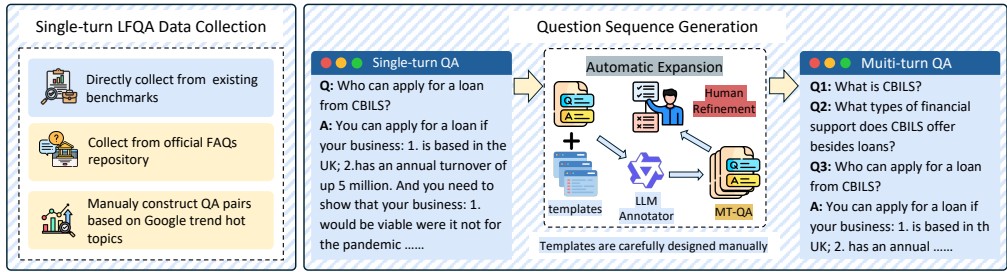

Figure 2: Overview of the data construction pipeline: including collecting single-trun LFQA pairs and expanding them into MT-LFQA instances.

**LFQA** is an open-domain QA task where a model is required to synthesize multiple facts into a paragraph-level answer. We formalize this setting as follows: the input is a set of knowledge-intensive questions $\mathcal{Q} = \{q_1, q_2, \ldots, q_N\}$. For each question $q_i \in \mathcal{Q}$, the ground-truth consists of a set of must-have facts $\mathcal{F}_i = \{f_{i1}, f_{i2}, \ldots, f_{iM_i}\}$, with the collection of all fact sets denoted by $\mathbb{F} = \{\mathcal{F}_1, \mathcal{F}_2, \ldots, \mathcal{F}_N\}$. These facts are composed into free-form ground-truth answers $\mathcal{G} = \{g_1, g_2, \ldots, g_N\}$, where each $g_i$ provides a complete and non-redundant representation of $\mathcal{F}_i$. To evaluate this task, a QA model $\mathcal{M}$ is applied to the question set $\mathcal{Q}$ to generate answers $\mathcal{A} = \{a_1, a_2, \ldots, a_N\}$, where each $a_i = \mathcal{M}(q_i)$. The generated answers $\mathcal{A}$ are then compared against the ground-truth answers $\mathcal{G}$, or equivalently against the supporting facts $\mathbb{F}$.

**MT-LFQA** is defined as the task that performs LFQA where the model is conditioned on the preceding conversational histories, and its definition naturally extends the single-turn setting formalized above. To formalize this task, we consider a set of $K$ dialogues $\mathcal{D} = \{d_1, \ldots, d_K\}$. Each dialogue $d_k \in \mathcal{D}$ consists of a sequence of $N_k$ turns, where $N_k$ is the total number of turns in dialogue $d_k$. The conversational context for the final turn is the preceding history, denoted as $H_k = (q_1^{(k)}, a_1^{(k)}, \ldots, q_{N_k-1}^{(k)}, a_{N_k-1}^{(k)})$. The final-turn question $q_{N_k}^{(k)}$ is a single-turn LFQA question from the set $\mathcal{Q}$. For a given $q_{N_k}^{(k)}$, we denote its corresponding ground-truth fact set and reference answer as $\mathcal{F}_j$ and $g_j$ respectively, where $q_{N_k}^{(k)} = q_j$ for some index $j \in \{1, \ldots, N\}$.

In task MT-LFQA, a model $\mathcal{M}$ is required to generate a factual and complete long-form answer for the final-turn question $q_{N_k}^{(k)}$, conditioned on the context $H_{N_k-1}^{(k)}$. The model's output is $a_{N_k}^{(k)} = \mathcal{M}(H_k, q_{N_k}^{(k)})$, where $H_k$ can be provided under two settings. In a *static context* setting, the history is pre-defined, composed of question-answer pairs authored by humans or generated by a model. In a *dynamic context* setting, which simulates an interactive session, the history is constructed on-the-fly by having the model $\mathcal{M}$ generate each answer in response to a pre-defined sequence of questions. Finally, the generated answer $a_{N_k}^{(k)}$ is then evaluated against its ground-truth ($\mathcal{F}_j$ and/or $g_j$), following the same assessment protocol as in single-turn LFQA.

## 3 KNOWMT-BENCH

Based on the task definition, this section introduces our benchmark, **KnowMT-Bench**, along with its data creation and evaluation pipeline, as illustrated in Figures 2 and Figures 3. We first curate high-quality, evidence-grounded single-turn LFQA instances and then expand them with multi-turn question sequences designed for dynamic evaluation. For assessment, we employ an automated, NLI-based pipeline, inspired by previous LFQA benchmarks (Manes et al., 2024), to score the final-turn answer. This human-validated pipeline allows us to evaluate model performance through a suite of metrics capturing both fact capability and information delivery efficiency.

### 3.1 SINGLE-TURN LFQA DATA COLLECTION

We first curate a high-quality set of single-turn LFQA instances. Each instance is a QA pair $(q_i, g_i)$, where $q_i \in \mathcal{Q}$ is a knowledge-intensive question and $g_i \in \mathcal{G}$ is its ground-truth answer. To ensure reliability, each $g_i$ is supported by an authoritative evidence set $\mathcal{E}_i = \{e_{i1}, \ldots, e_{iK_i}\}$ extracted from trusted sources such as official websites or expert-curated documents.

Our benchmark focuses on three representative specialized domains: medicine, finance, and law, and draws data from three sources: (i) prior LFQA benchmarks, (ii) authoritative financial-legal FAQs,

Table 1: Comparison between KnowMT-Bench with existing QA benchmarks. # Avg. Tokens refer to the token counts of the ground truth answer, computed by the GPT-4o tokenizer. *: Since FintextQA is partially open-sourced, we directly report the result in their paper, which is 75 **words**.

| Benchmark | # Avg. Turns | # Avg. Tokens | Multi-Turn | Open-Domain | Across-Domain |
|---|---|---|---|---|---|
| CoQA (Reddy et al., 2019) | 15.97 | 2.52 | ✓ | ✗ | ✓ |
| QASA (Lee et al., 2023) | 1 | 50 | ✗ | ✗ | ✗ |
| K-QA (Manes et al., 2024) | 1 | 119.89 | ✗ | ✓ | ✗ |
| MedLFQA (Jeong et al., 2024) | 1 | 132.86 | ✗ | ✓ | ✗ |
| FintextQA (Chen et al., 2024b) | 1 | 75* | ✗ | ✓ | ✗ |
| **KnowMT-Bench (ours)** | 2.98 | 95.85 | ✓ | ✓ | ✓ |

and (iii) finance-related trending topics. In the medical domain, we include all 201 labeled QA pairs from the K-QA benchmark (Manes et al., 2024), after removing redundant content to align with our task definition. For the financial-legal domain, we collect 116 QA pairs from the SEC FAQ repository[1] and the policy-focused subset of FinTextQA (Chen et al., 2024b), while filtering out trivial single-point answers and repairing missing jurisdictional context, resulting in 184 pairs. To broaden coverage, we further sample finance-related trending topics from Google Trends[2], categorize them, and construct 300 QA pairs through manual annotation with authoritative references, including their official website or encyclopedia verified by human experts.

In total, this process yields **801 high-quality single-turn LFQA instances** spanning three domains: finance (579), law (278), and medicine (209). Notably, 33.1% of instances are multi-domain, with 261 cases primarily located at the finance-legal intersection. Detailed annotation procedures and additional statistics are provided in Appendix D.1.

## 3.2 QUESTION SEQUENCE GENERATION

To mirror real conversational patterns, we analyze the ShareGPT-Chinese-English-90k dataset (shareAI, 2023) and find the following distribution for knowledge-intensive dialogues up to five turns: 2-turn (38.5%), 3-turn (38.2%), 4-turn (14.0%), and 5-turn (5.4%). Since dialogues longer than 5 turns occur at a negligible rate ($\leq 5\%$), we merge them into 5 turns and set the maximum dialogue length to $N_{max} = 5$. In our benchmark, dialogue lengths are drawn from this empirical distribution of 2–5 turns (37.45%, 37.45%, 14.98%, and 10.11%, respectively).

For each single-turn question $q_j \in \mathcal{Q}$, we generate a multi-turn question sequence $\mathbf{q}^{(d)} = (q_1^{(d)}, \ldots, q_{N_d}^{(d)})$ of a sampled length $N_d \in \{2, \ldots, 5\}$, with the final question $q_{N_d}^{(d)} = q_j$. The preceding questions $q_{1:N_d-1}^{(d)}$ are created under a human-in-the-loop paradigm where combining LLM-based generation with manual review ensured that each sequence adheres to three key principles: (1) **Progressive Context Building**: Questions gradually establish background or narrow the scope. (2) **Intent Preservation**: The sequence naturally leads to the final question $q_j$ without semantic drift. (3) **No Answer Leakage**: Preceding questions do not reveal or hint at the answer to $q_j$. In total, the procedure produced 801 question sequences. We provide more details in Appendix D.2.

During evaluation, we evaluate models using a dynamic setting where the model self-generates its own dialogue history. Specifically, for a given question sequence $\mathbf{q}^{(d)}$, the history for the final turn, $H_{N_d}^{(d)}$, is constructed by recursively generating each intermediate answer:

$$a_t^{(d)} = \mathcal{M}(q_1^{(d)}, a_1^{(d)}, \ldots, q_{t-1}^{(d)}, a_{t-1}^{(d)}, q_t^{(d)}), \quad \text{for } t = 1, \ldots, N_d - 1. \tag{1}$$

Finally, the model generates the targeted answers $a_{N_d}^{(d)} = \mathcal{M}(H_{N_d}^{(d)}, q_{N_d}^{(d)})$ for MT-LFQA.

## 3.3 EVALUATION FRAMEWORK

To systematically evaluate model performance in MT-LFQA, we introduce a comprehensive evaluation framework, which has two components: a two-stage automated pipeline to assess factual alignment, and a three-dimensional metric suite to quantify performance from the pipeline's outputs. The evaluation focuses on the final-turn response ($a_j$) in a dialogue, measuring it against the ground-truth answer ($g_j$) and the ground-truth must-have facts set $\mathcal{F}_j$.

---

[1]https://www.sec.gov/answers/faqs.htm
[2]https://trends.google.com/trends/

Figure 3: Overview of the two-stage natural language inference-based evaluation. Here, **GTA** refers to **G**round **T**ruth **A**nswer, and the detailed prompts for evaluation are provided in Appendix J.

### 3.3.1 TWO-STAGE EVALUATION PIPELINE

The core of our framework is a two-stage, NLI-based evaluation pipeline, which is designed to assess factual consistency at a fine-grained level. In the first stage, we employ **Qwen2.5-32B-Instruct** (Team, 2024) as a **decomposer** to break down long-form answers into minimal, self-contained factual units. For the ground-truth answers ($g_j$), this process yields a set of atomic facts ($\mathcal{F}_j$), which subsequently undergoes manual verification and refinement to establish the gold standard. During testing, the model-generated answer ($a_j$) is dynamically decomposed into a corresponding set of atomic statements ($\mathcal{S}_j$). In the second stage, we use **Qwen2.5-14B-Instruct** as the **NLI-based evaluator** to assess the factual consistency between ground-truth and generated answers. To avoid the quadratic computational complexity, $O(|\mathcal{S}_j| \cdot |\mathcal{F}_j|)$, of an exhaustive comparison between atomic statement sets, we adopt an efficient symmetric approach. Completeness is measured by judging each gold-standard fact $f \in \mathcal{F}_j$ against the full model-generated answer $a_j$. Conversely, correctness is measured by judging each generated statement $s \in \mathcal{S}_j$ against the full ground-truth answer $g_j$. The evaluator classifies each relationship as **Entailment**, **Contradiction**, or **Neutral**.

To ensure the reliability of our pipeline, we conduct a human validation on a sample of 100 generated answers, randomly drawn from representative LLMs across single-turn or multi-turn settings. For the **decomposition stage**, we compare the decomposer's output against human decomposition on these 100 answers. The process demonstrates high fidelity, with a Symmetric Mean Absolute Percentage Error (SMAPE) of **18.1%** in statement counts and an omission rate of **5.9%**. Errors mainly arose from under-segmentation rather than semantic distortion. For the **judgment stage**, these dialogues were used to construct 1,687 evaluation NLI-pairs. The agreement between our NLI-based evaluator and the resulting gold annotations from majority voting among three annotators reached an F1-score of **83.6%**, confirming that our pipeline provides a reliable measure of factual consistency. Further details on the human annotation process, including the models sampled and the prompts utilized, are available in Appendix D.3 and Appendix J, respectively.

### 3.3.2 THREE-DIMENSIONAL METRIC FRAMEWORK

The NLI judgments are aggregated into a suite of metrics organized under three distinct dimensions.

**Factuality** This dimension quantifies the correctness and completeness of the must-have fact provided. It is based on **Factual Precision** ($\mathbf{P_f}$), the fraction of generated statements that are entailmented, and **Factual Recall** ($\mathbf{R_f}$), the fraction of ground-truth facts covered. These are combined into the **Factual F1** ($\mathbf{S_f}$) score for a comprehensive assessment.

$$\mathbf{R_f} = \frac{1}{|\mathcal{D}|} \sum_{j \in \mathcal{D}} \frac{|\mathcal{F}_j^+|}{|\mathcal{F}_j|}, \quad \mathbf{P_f} = \frac{1}{|\mathcal{D}|} \sum_{j \in \mathcal{D}} \frac{|\mathcal{S}_j^+|}{|\mathcal{S}_j|}, \quad \mathbf{S_f} = \frac{2\mathbf{P_f}\mathbf{R_f}}{\mathbf{P_f} + \mathbf{R_f}} \tag{2}$$

**Reliability (Factual Hallucination)** This dimension measures the extent of factual hallucination. Analogous to factuality, it is quantified using the **False Claim Rate** ($\mathbf{P_{fc}}$), the fraction of generated statements that are contradicted, and the **Misrepresentation Rate** ($\mathbf{R_m}$), the fraction of ground-truth facts contradicted by the ($a_j$). These are unified into the **Hallucination F1** ($\mathbf{S_h}$) score.

$$\mathbf{R_m} = \frac{1}{|\mathcal{D}|} \sum_{j \in \mathcal{D}} \frac{|\mathcal{F}_j^-|}{|\mathcal{F}_j|}, \quad \mathbf{P_{fc}} = \frac{1}{|\mathcal{D}|} \sum_{j \in \mathcal{D}} \frac{|\mathcal{S}_j^-|}{|\mathcal{S}_j|}, \quad \mathbf{S_h} = \frac{2\mathbf{P_{fc}}\mathbf{R_m}}{\mathbf{P_{fc}} + \mathbf{R_m}} \tag{3}$$

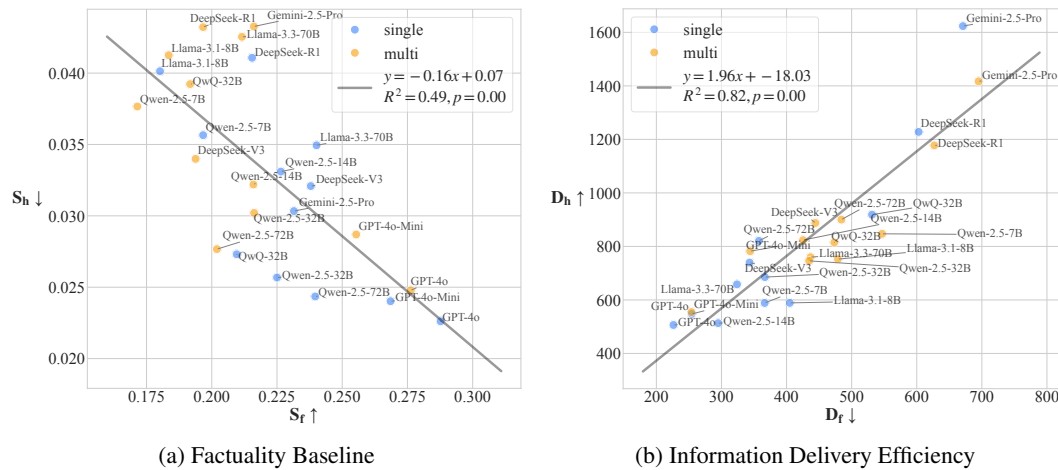

(a) Factuality Baseline        (b) Information Delivery Efficiency

Figure 4: Results for LLMs Performance on Factuality and Information Delivery Efficiency.

**Information Delivery Efficiency** This dimension assesses the utility of the model's response by measuring the token cost of conveying information. This provides an intuitive measure of efficiency, as users directly interact with the token count. We report $\mathbf{D_f}$, the average tokens per correctly entailed fact, $\mathbf{D_h}$, the average tokens per contradicted fact, and $\mathbf{D_R}$, the average tokens per to cover the entire set of ground-truth facts. Lower values indicate higher efficiency.

$$\mathbf{D_f} = \frac{1}{|\mathcal{D}|} \sum_{j \in \mathcal{D}} \frac{T(a_j)}{|\mathcal{F}_j^+|}, \quad \mathbf{D_h} = \frac{1}{|\mathcal{D}|} \sum_{j \in \mathcal{D}} \frac{T(a_j)}{|\mathcal{F}_j^-|}, \quad \mathbf{D_R} = \frac{1}{|\mathcal{D}|} \sum_{j \in \mathcal{D}} \frac{T(a_j)}{r_f(j)}, \tag{4}$$

where $T(a_j)$ is the token length of $a_j$, and $r_f(j) = \frac{|\mathcal{F}_j^+|}{|\mathcal{F}_j|}$ is the factual recall for $a_j$. If the denominator of any term equals zero, i.e. $|\mathcal{F}_j^+| = 0$, $|\mathcal{F}_j^-| = 0$, or $r_f(j) = 0$, that term is estimated by $\max_{k \in \mathcal{D}} \frac{T(a_k)}{|\cdot|}$ of the corresponding metric.

## 4 EXPERIMENTS

### 4.1 EXPERIMENTAL SETUP

We evaluate a broad set of popular LLMs, including the DeepSeek series: DeepSeek-V3-0324, denoted as **DeepSeek-V3** (Liu et al., 2024) and DeepSeek-R1-0528, denoted as **DeepSeek-R1** (Guo et al., 2025), OpenAIs GPT models (Achiam et al., 2023): gpt-4o-mini-2024-07-18, denoted as **GPT-4o mini**, and gpt-4o-2024-08-06, denoted as **GPT-4o**, Metas Llama family (Touvron et al., 2023): Llama-3.1-8B-Instruct, denoted as **Llama-3.1-8B** and Llama-3.3-70B-Instruct, denoted as **Llama-3.3-70B**, the Qwen family (Team, 2024): Qwen-2.5-7B/14B/32B/72B-Instruct, denoted collectively as **Qwen-2.5-7B/14B/32B/72B**, as well as **QwQ-32B**, and **Gemini-2.5-Pro** (Comanici et al., 2025). Token counts are computed using tiktoken's gpt-4o tokenizer.[3] All experiments are conducted on NVIDIA A800 GPUs. See detailed experiment settings in Appendix G.

### 4.2 MAIN RESULTS

To reveal the relations between different performance dimensions, we map all model performances into scatter plots (Figure 4). We present four primary metrics (also used in the experiments that follow); detailed numerical results and additional metric values are provided in Appendix H. A clear observation from the plots is a systematic shift in performance when moving from single-turn to multi-turn settings. Specifically, in Figure 4a, most models exhibit a top-left shift from their single-turn to multi-turn counterparts, indicating a decrease in the Factual F1 score ($\mathbf{S_f}$) and an increase in the Hallucination F1 score ($\mathbf{S_h}$). Similarly, in Figure 4b, the points generally shift to the top-right, which means that models require more tokens to convey correct facts (higher $\mathbf{D_f}$), while their generated factual errors also become sparser (higher $\mathbf{D_h}$). This pronounced degradation in factuality, coupled with the challenge of efficiently delivering correct information in multi-turn

---

[3]https://github.com/openai/tiktoken

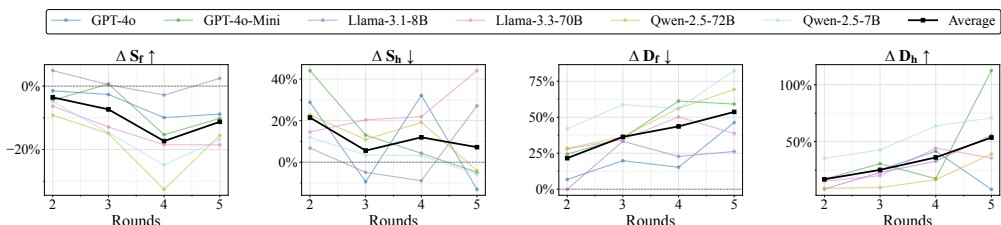

Figure 5: Relative difference between multi-turn and single-turn across models.

dialogues, highlights the unique challenges posed by the MT-LFQA task. Within this general trend, several other patterns are also apparent: proprietary models such as GPT-4o define the performance frontier, and larger models generally outperform smaller ones within the same family. However, it is noteworthy that models optimized for Chain-of-Thought (e.g., DeepSeek-R1 to DeepSeek-V3, QWQ-32B to Qwen-2.5-32B) show no advantage in fact capability, suggesting that current CoT mechanisms may not directly translate to more factual answers.

In addition, the relationships between data points also reveal two underlying correlations. First, Figure 4a illustrates a moderate negative correlation between the Factual F1 score ($S_f$) and the Hallucination F1 score ($S_h$) ($R^2 = 0.49$, t-test $p < 0.001$), which suggests that as a model's factuality improves, its tendency to factual hallucinate generally decreases. This points to a potential strategy for mitigating hallucinations: enhancing a model's intrinsic knowledge may be an effective path to reducing false claims. Second, Figure 4b reveals a strong positive correlation between the token cost per correct fact ($D_f$) and per contradicted fact ($D_h$) ($R^2 = 0.82$, t-test $p < 0.001$). This indicates that most models tend to be uniformly concise or verbose, rather than dynamically adjusting their efficiency based on the correctness of the information. For instance, the GPT-4o family is characterized by low costs on both metrics, demonstrating high efficiency. An interesting outlier in the single-turn setting is Gemini-2.5-Pro, which deviates significantly from the regression line; its $D_h$ is exceptionally high for its given $D_f$, successfully pushing this trade-off boundary. Notably, this desirable characteristic disappears in the multi-turn dialogue setting, where its performance aligns with the general trend. Therefore, a key direction for future research is to design models or strategies that can break this trade-off by achieving a low $D_f$ while simultaneously increasing $D_h$ within the multi-turn dialogue. A detailed, holistic four-quadrant analysis is provided in Appendix H.2.

### 4.3 THE SOURCE OF MULTI-TURN PERFORMANCE DEGRADATION

In the main experiment, we observed a systematic performance degradation in MT-LFQA. To disentangle the root causes, we analyze three different properties of the dialogue context: the contextual length, the multi-turn template structure, and the noise level within the history.

**Effect of dialogue length.** To isolate the impact of length, we conduct two controlled experiments. **(i) Number of Rounds.** We vary the turn count and report the relative change of each metric $m \in \{S_f, S_h, D_f, D_h\}$ against the single-turn baseline, calculated as $\Delta m = (m_{\text{multi}} - m_{\text{single}})/m_{\text{single}}$ (Figure 5). **(ii) Context Length.** For each multi-turn instance, we constrain the dialogue history preceding the final question to varying token budgets. We then regenerate the final-turn answer based on each truncated context (Figure 6). A consistent trend emerges: the information delivery efficiency degrades monotonically as the context lengthens. This suggests that longer contexts induce a dilution effect, making models increasingly verbose. However, factual capability ($S_f$) shows a slight overall decrease, and hallucination ($S_h$) exhibits no clear length dependence. Thus, **dialogue length primarily dictates the information delivery efficiency rather than the factual capability.**

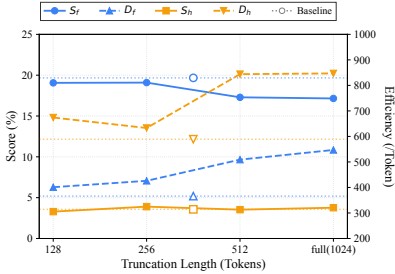

Figure 6: Impact of context length on Qwen-2.5-7B performance.

**Effect of multi-turn structure.** To examine the effect of the conversational format, we compare the standard multi-turn setting against a *concatenated-history* variant. Specifically, for each dialogue, we integrate all previous turns and the final-turn question into a single long prompt, instructing the model to regenerate the final-turn answer. As shown in Figure 7, this variant exhibits a distinct behavioural shift: it achieves the highest factual performance, surpassing even the single-

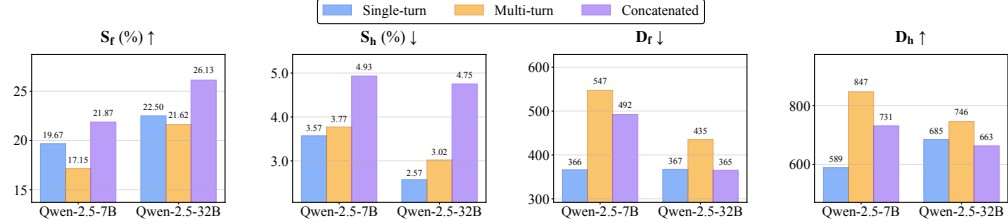

Figure 7: Comparison between standard multi-turn and a concatenated history setting.

turn baseline, and significantly improves efficiency. However, this comes at the cost of increased hallucination. This suggests that the multi-turn template imposes a conservative constraint, making the model more cautious and verbose. Removing this structure yields more efficient fact delivery but at a higher error risk, indicating that the format primarily shapes generation strategy.

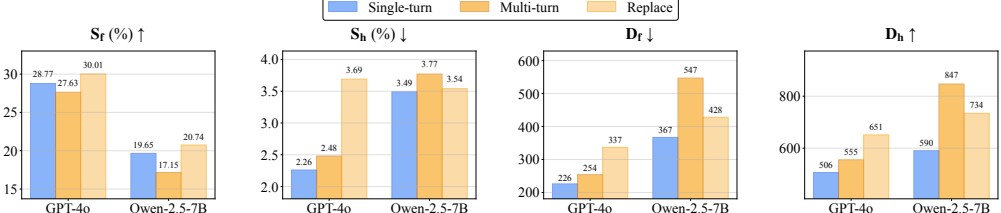

Figure 8: Wapping experiment between strong model (GPT-4o) and weaker model (Qwen-2.5-7B). It also underscores our dynamic evaluation setting more faithfully captures real model performance.

**Noise in dialogue history.** Finally, we investigate the intrinsic quality of the context. We approach this via two complementary analyses focusing on natural variations and synthetic perturbations. First, utilizing natural variations in history quality, we conduct a bidirectional substitution experiment (Figure 8). In the upgrading direction, conditioning weaker models on histories generated by a stronger model (GPT-4o) significantly improves factuality and reliability, while also increasing information density. This confirms that a cleaner context directly boosts performance. In the downgrading direction, where GPT-4o is conditioned on weaker histories (Qwen-2.5-7B), we observe a distinct "disruption" effect. The strong model, faced with noisy context, loses its calibration: it becomes unconstrained and verbose, leading to higher factuality but at the severe cost of significantly increased hallucinations. This demonstrates that high-quality history acts as a necessary guardrail for reliability; without it, even strong models degrade into aggressive but ungrounded generation. For more results of interpretability experiments, see Appendix H.7.

Second, we employ a controlled noise-injection protocol (Figure 9). To examine how noise affects known facts, we first curate a "knowledge-verified" baseline by filtering for 85 instances where GPT-4o exhibits perfect factual mastery in the single-turn setting ($S_f \geq 60\%, S_h = 0$). Keeping the final question fixed, we manipulate *only the content of the previous turn* to construct three distinct history types: *Irrelevant Noise* (correct but unrelated), *Irrelevant Error* (incorrect and unrelated), and *Relevant Error* (incorrect and related). We evaluate these perturbations under both standard multi-turn and concatenated-history settings and observe a **universal degradation pattern**: as noise severity increases, factuality drops monotonically while hallucinations rise sharply. Notably, *relevant errors* induce the most catastrophic degradation, which is accompanied by a significant deterioration in information delivery efficiency, suggesting that models become confused and verbose when struggling to reconcile conflicting context with their internal knowledge. This trend persists regardless of the

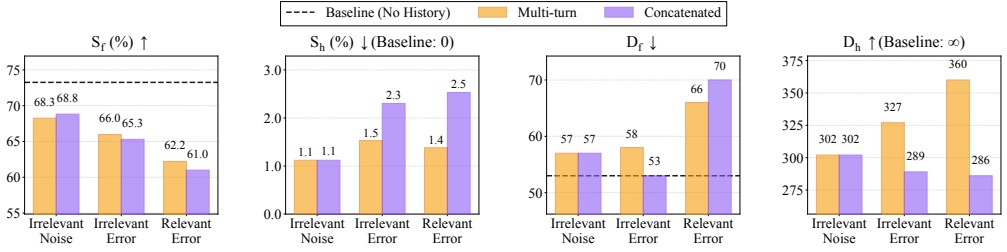

Figure 9: Performance degradation under controlled injections of noise

Table 2: Results across different domains. Blocks are ordered as **non-domain** (base) followed by a consolidated **domain-specific** block. Headers annotate the applicable domain-specific model: Finance (Fin-R1) and Medical (HuatuoGPT). Underlined values denote improvements over the base model under the same domain and turn setting. White rows indicate single-turn results, while gray rows indicate multi-turn results.

| Finance | | | | Non-finance | | | | Medical | | | | Non-medical | | | |
|---|---|---|---|---|---|---|---|---|---|---|---|---|---|---|---|
| $S_f$ (%)↑ | $S_h$ (%)↓ | $D_f$↓ | $D_h$↑ | $S_f$ (%)↑ | $S_h$ (%)↓ | $D_f$↓ | $D_h$↑ | $S_f$ (%)↑ | $S_h$ (%)↓ | $D_f$↓ | $D_h$↑ | $S_f$ (%)↑ | $S_h$ (%)↓ | $D_f$↓ | $D_h$↑ |
| *Qwen-2.5-7B* | | | | | | | | | | | | | | | |
| 16.61 | 4.22 | 402.95 | 576.60 | 21.73 | 1.40 | 302.15 | 421.90 | 22.02 | 1.41 | 291.35 | 421.31 | 16.62 | 4.15 | 402.20 | 578.08 |
| 14.44 | 4.38 | 604.55 | 831.75 | 19.58 | 1.30 | 438.16 | 718.07 | 20.29 | 1.31 | 392.80 | 710.41 | 14.30 | 4.31 | 606.08 | 833.95 |
| *Domain-specific: Fin-R1* | | | | | | | | *Domain-specific: HuatuoGPT* | | | | | | | |
| 15.74 | 4.18 | 588.71 | 841.60 | 21.90 | 1.73 | 472.22 | 918.13 | 26.67 | 1.74 | 431.49 | 655.61 | 17.91 | 4.41 | 538.26 | 725.75 |
| 14.32 | 4.94 | 535.13 | 785.13 | 18.95 | 1.60 | 306.43 | 729.56 | 25.65 | 1.07 | 413.10 | 624.44 | 17.67 | 4.08 | 571.21 | 757.99 |

input structure, confirming that contextual noise, specifically relevant misinformation, operates as the primary determinant of knowledge grounding, overriding the effects of template or length.

# 5 MITIGATION STRATEGIES

To address the performance degradation in MT-LFQA, we investigate mitigation strategies along two directions. The first direction focuses on **Knowledge Fortification**, aiming to enhance the model's factual grounding to make it more robust against hallucination and noise. We explore this via leveraging **extrinsic knowledge** through Retrieval-Augmented Generation (RAG) (Section 5.1) and strengthening **intrinsic knowledge** through domain-specific finetuning (Section 5.2). The second direction targets **Contextual Denoising**, where we move beyond passive context consumption. Instead, we employ **inference-time interventions** to actively filter, refine, or restructure the dialogue history, thereby directly attenuating the interference from noisy contexts (Section 5.3).

## 5.1 EFFECT OF RAG STRATEGIES

To evaluate the effect of extrinsic knowledge, we test four RAG strategies: **Base** (retrieval at the final turn using the last-turn query), **Last** (retrieval at the final turn using the full dialogue), **Rounds** (retrieval at each turn using the current query), and **All** (retrieval at each turn using the full previous history as the query). In the single-turn setting, since there are no other turns and dialogue history, only the Base settings are available. Detailed settings are provided in Appendix G.

As shown in Table 3, the baseline model without RAG confirms the performance degradation in its factual capacity ($S_f$ and $S_h$) from single-turn to multi-turn set-

Table 3: Comparison of With versus Without RAG on **Qwen-2.5-7B**. Gray and white rows indicate as above. **Bold** numbers highlight the best results under the multi-turn RAG setting. Underlined values denote improvements over the baseline under the same turn setting (single-turn or multi-turn).

| Setting | $S_f$ (%)↑ | $S_h$ (%)↓ | $D_f$↓ | $D_h$↑ |
|---|---|---|---|---|
| **w/o RAG** | | | | |
| Original | 19.67 | 3.57 | 366.41 | 588.70 |
| Original | 17.15 | 3.77 | 546.79 | 847.17 |
| **w/ RAG** | | | | |
| Base | 42.92 | 1.98 | 193.42 | 451.58 |
| Base | 42.55 | 2.43 | 269.65 | **882.00** |
| Last | 41.59 | 2.78 | 231.50 | 601.78 |
| Rounds | **43.15** | **2.14** | **215.84** | 677.44 |
| All | 39.97 | 2.48 | 245.53 | 561.50 |

tings. The introduction of RAG provides a substantial improvement in factual capacity for both settings. Among these strategies, **Rounds** is the best strategy in the multi-turn setting, achieving the highest factuality ($S_f$), the fewest hallucinations ($S_h$), and the best correct information delivery efficiency ($D_f$). Notably, this strategy is so effective that it reverses the performance degradation on some dimension: it enables the model to achieve a higher factuality score in the multi-turn setting than even the RAG-enhanced single-turn baseline. This highlights RAG's capacity not merely to mitigate noise, but to actively leverage the multi-turn structure by grounding each step with factual evidence, thus preventing the accumulation of noise that characterizes the non-RAG setting. In contrast, the **Last** and **All** strategies, which use the full dialogue history as the queries, are more likely to accumulate noise and therefore underperform.

## 5.2 EFFECT OF DOMAIN-SPECIFIC FINETUNING

First, we evaluate the effect of intrinsic knowledge by examining the effect of domain-specific finetuning. Accordingly, we evaluate two Qwen-2.5-7B derivatives: **Fin-R1**[4] (Liu et al., 2025) for

---
[4] https://huggingface.co/SUFE-AIFLM-Lab/Fin-R1

finance and **HuatuoGPT**[5] (Chen et al., 2024c) for medicine. Table 2 indicates that HuatuoGPT outperforms the baseline in both single- and multi-turn settings, with the improvement being more pronounced in the multi-turn context. It suggests that injecting domain knowledge not only improves factuality but also suppresses noise accumulated in the dialogue history, leading to substantially better multi-turn performance. In particular, the improvement is stronger for the medical domain than the non-medical domain. In contrast, the Fin-R1 model's performance in the *finance domain* does not show consistent superiority over the generalist baseline. In summary, these results indicate that domain specialization can be an effective strategy for enhancing factual capability to suppress noise accumulated in multi-turn dialogues, as demonstrated by HuatuoGPT. However, the case of Fin-R1 highlights that such benefits are not guaranteed.

## 5.3 PROMPT-BASED STRATEGIES

Building on our previous analysis, we explored various prompt-based mitigation strategies to address the identified drivers of degradation. We began with a straightforward intervention, explicitly instructing the model: "You may refer to the previous conversation for context, but rely primarily on your internal knowledge." This *Simple Prompt* yielded only slight improvements in factuality and efficiency while leaving hallucination levels virtually unchanged, suggesting that explicit instructions alone are insufficient to filter out noise. Moving to a structural intervention, *Self-Refinement* Madaan et al. (2023) successfully reduced hallucinations; however, it induced excessive conservatism, resulting in a significant trade-off where factual recall notably declined.

Table 4: Performance comparison of **Qwen2.5-32B** with prompt-based mitigation strategies. Gray rows indicate multi-turn settings. Underlined values denote improvements relative to the multi-turn baseline.

| Strategy | $S_f$ (%) ↑ | $S_h$ (%) ↓ | $D_f$ ↓ | $D_h$ ↑ |
|---|---|---|---|---|
| **Baselines** | | | | |
| Original (Single) | 22.50 | 2.57 | 367 | 685 |
| Original (Multi) | 21.62 | 3.02 | 435 | 746 |
| | | | | |
| **Interventions** | | | | |
| + Simple Prompt | 21.89 | 3.05 | 417 | 764 |
| + Summarization | 24.05 | 3.28 | 259 | 419 |
| + Selection | 24.10 | 3.19 | 319 | 590 |
| + Self-Refine | 19.84 | 2.90 | 525 | 899 |

In contrast, strategies that fundamentally restructure the context proved significantly more effective. In both *Summarization* and *Context Selection*, the model processes the preceding dialogue history, either by compressing or filtering it, and concatenates the refined context with the current question to generate the response. These approaches successfully counteract the negative impacts of excessive context length and rigid dialogue templates. By presenting a concise and focused context, both strategies alleviate the model's "conservative" constraint, thereby restoring information delivery efficiency and fostering the generation of more factual content. *Context Selection*, however, demonstrates a distinct advantage; by actively excluding noise rather than merely compressing it, it emerges as the most effective strategy, delivering the strongest recovery in factual capability.

## 6 CONCLUSION

In this work, we addressed the critical gap in evaluating LLMs for knowledge-intensive, MT-LFQA by introducing **KnowMT-Bench**, a new benchmark featuring a dynamic evaluation protocol and a fine-grained, NLI-based assessment framework. Our experiments across a diverse suite of LLMs reveal a consistent multi-turn decline in both *information-delivery efficiency* and *factual capability*. Probing analyses trace this degradation to contextual noise, especially relevant misinformation accumulated from self-generated history, together with increasing context length and dialogue structure. Mitigation studies further show that structural context refinement and RAG alleviate these failures. These findings motivate three concrete avenues for future model development: (1) **Intrinsic Contextual Denoising** equip models with inference-time evidence triage to actively select, compress, and discard irrelevant or self-generated context rather than consuming it passively; (2) **Decoupling Verbosity from Reasoning** design objectives, training signals, and decoding controls that preserve factual density and calibration while constraining length, thereby breaking the empirical trade-off between concision and correctness; and (3) **Robustness to Error Propagation** integrate uncertainty estimation, conflict detection, and self-correction/rollback to contain early-turn hallucinations before they cascade downstream. **KnowMT-Bench** paves the way for developing more sophisticated models and intervention strategies to improve reliability in real-world, knowledge-intensive dialogues.

---

[5] https://huggingface.co/FreedomIntelligence/HuatuoGPT-o1-7B

ETHICS STATEMENT

This research complies with ethical standards. It utilizes datasets that are either synthetic or publicly available, and contains no sensitive or personally identifiable information. The study involves no direct human subjects, nor does it pose any privacy or security concerns. All methodologies and experiments are conducted in accordance with applicable laws and established research integrity practices. There are no conflicts of interest, no undue influence from external sponsorship, and no concerns related to discrimination, bias, or fairness. Moreover, this research does not lead to any harmful insights or applications.

REPRODUCIBILITY STATEMENT

We have taken steps to ensure the reproducibility of the results presented in this paper. The experimental settings, including datasets and models, are thoroughly described in Section 4.1 and Appendix G. Source code will be made publicly available upon acceptance.

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

Table 5: Conversational QA and LFQA benchmark. Columns show average ground-truth length, average number of turns, and whether dialogs are multi-turn, answers are long-form, and the task is open-domain (✓=yes, ✗=no). Notation: t = tokens, w = words, c = Chinese characters, -=not provided in the original paper.

| Benchmark | # Avg. Length | # Avg. Turns | Multi-turn | Long-Form | Open-Domain |
|---|---|---|---|---|---|
| QuAC (Choi et al., 2018) | 14.6 t | 7.2 | ✓ | ✗ | ✗ |
| CoQA (Reddy et al., 2019) | 2.52 t | 15.97 | ✓ | ✗ | ✗ |
| DoQA (Campos et al., 2020) | 12.99 t | 4.48 | ✓ | ✗ | ✗ |
| Doc2Dial (Feng et al., 2020) | 21 t | 12 | ✓ | ✗ | ✗ |
| MultiDoc2Dial (Feng et al., 2021) | 21.6 t | 6.36 | ✓ | ✗ | ✗ |
| ConvFinQA (Chen et al., 2022) | - | 3.67 | ✓ | ✗ | ✗ |
| TopiOCQA (Adlakha et al., 2022) | 11.75 w | 12 | ✓ | ✗ | ✓ |
| InsCoQA (Wu et al., 2024) | - | 3.11 | ✓ | ✗ | ✗ |
| QASA (Lee et al., 2023) | 50 t | 1 | ✗ | ✓ | ✗ |
| FinTextQA (Chen et al., 2024b) | 75 w | 1 | ✗ | ✓ | ✓ |
| K-QA (Manes et al., 2024) | 119.89 t | 1 | ✗ | ✓ | ✓ |
| MedLFQA (Jeong et al., 2024) | 132.86 t | 1 | ✗ | ✓ | ✓ |
| cLegal-QA (Wang et al., 2025) | 93 c | 1 | ✗ | ✓ | ✓ |
| **KnowMT-Bench (ours)** | 75.75 w / 95.85 t | 2.98 | ✓ | ✓ | ✓ |

## APPENDIX

## A    LLM USAGE STATEMENT

LLMs were used for prose refinement (grammar, phrasing) and code edits (formatting). The authors reviewed all LLM suggestions and take full responsibility for the paper.

## B    RELATED WORK

### B.1    MULTI-TURN DIALOGUES BENCHMARKS FOR LLMS

Evaluating LLMs in multi-turn dialogues is a critical and active area of research. Early benchmarks such as MT-Bench (Zheng et al., 2023) and MT-Bench++(Sun et al., 2024) assess general conversational quality using an LLM-as-judge approach. Subsequent works have introduced more diverse evaluation paradigms. For instance, BotChat Duan et al. (2024) evaluates alignment with human conversational patterns, while others, including MT-Eval (Kwan et al., 2024) and MT-Bench-101 (Bai et al., 2024), propose multi-dimensional frameworks to assess specific capabilities like instruction adherence and context utilization.

Beyond general benchmarks, a range of specialized benchmarks have been proposed to probe distinct abilities within multi-turn dialogue. TurnBench-MS (Zhang et al., 2025b) and WIL (Banatt et al., 2024) are designed to assess iterative multi-step reasoning, while Multi-IF (He et al., 2024) and StructFlowBench (Li et al., 2025) focus on the instruction-following ability of LLMs. MINT (Wang et al., 2023) explicitly evaluates LLMs ability to incorporate external tools and language feedback during multi-turn interactions. Additionally, some benchmarks evaluate critical risks in multi-turn dialogues, with FairMT-Bench (Fan et al., 2024) measuring fairness and bias propagation. While comprehensive, these benchmarks do not specifically focus on the factual capability of LLMs within multi-turn, knowledge-grounded dialogues. Our work addresses this critical gap by systematically assessing factual capability in such contexts.

### B.2    LONG-FORM QUESTION ANSWERING BENCHMARKS FOR LLMS

LFQA is a knowledge-base open-domain question answering task, particularly in specialized domains like medicine, finance, and law. Evaluation paradigms for LFQA have evolved over time. Some benchmarks in LFQA, such as FinTextQA (Chen et al., 2024b) and cLegal-QA (Wang et al., 2025) rely on surface-level similarity metrics like ROUGE (Lin, 2004), which may not correlate well with human evaluation (Xu et al., 2023). To improve reliability, subsequent works such as LEXam (Fan et al., 2025) adopted an LLM-as-judge paradigm.

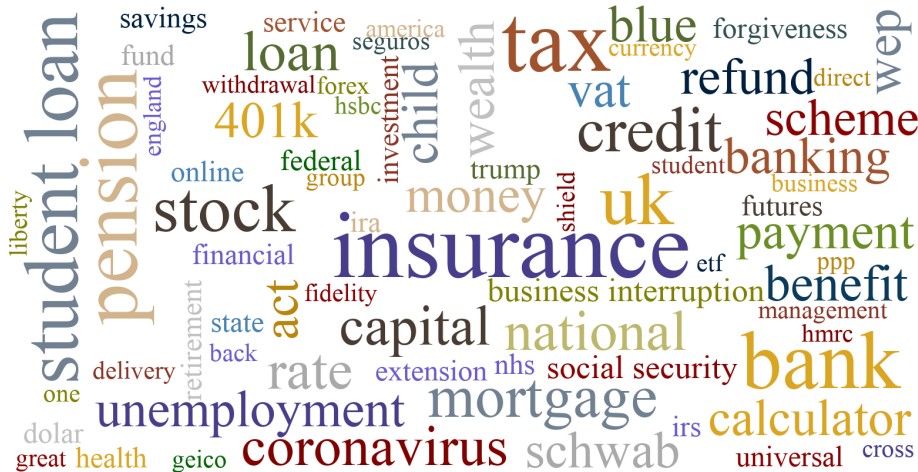

Figure 10: Financial Topics on Google Trends.

A more fine-grained and interpretable approach was introduced by K-QA and MedLFQA (Manes et al., 2024; Jeong et al., 2024), which first decomposes reference answers into atomic facts and then uses NLI-based methods to check for entailment and contradiction at the fact level. This paradigm offers enhanced interpretability by assessing factuality on explicit, human-understandable statements. Our benchmark builds on this NLI-based paradigm but makes some key advancements. We introduce a new dimension for **information delivery efficiency** to evaluate content effectiveness, and we extend the LFQA task to a multi-turn dialogue setting for the first time to make it more closely resemble professional consultation scenarios.

## C    DETAILED COMPARISON TO MORE QA BENCHMARK

Table 5 summarizes widely used conversational QA and long-form QA benchmarks in terms of average answer length, number of turns, and task characteristics. Most conversational QA datasets emphasize multi-turn interaction but contain relatively short answers, while long-form QA benchmarks are typically single-turn and open-domain with longer responses. Our KnowMT-Bench bridges these two directions by combining multi-turn dialogue with long-form, open-domain answers, reflecting more realistic information-seeking scenarios.

## D    ANNOTATION DETAILS

### D.1    ADDITIONAL DETAILS OF SINGLE-TURN LFQA DATA COLLECTION

For the medical domain, we included all 201 QA pairs from the K-QA benchmark (Manes et al., 2024). Since the original answers often contained both *must-to-have* supporting facts and *nice-to-have* details, we manually removed the redundant segments beyond the must-have facts while ensuring that the resulting ground-truth answers remained coherent and fluent. The supporting evidence was derived from authoritative sources such as institutional websites used during K-QA annotation.

For the financial-legal domain, we collect 116 QA pairs from the official FAQ repository maintained by the U.S. Securities and Exchange Commission (SEC)[6] and 184 QA pairs from the policy-focused subset of FinTextQA (Chen et al., 2024b). Several quality-control steps were applied, including the removal of trivial answers consisting only of affirmation, negation, or phrase-level responses, as well as the manual addition of missing jurisdictional context (e.g., "Hong Kong" in entries from HKMA) to resolve ambiguities. The resulting curated subset thus covers major financial jurisdictions, including Hong Kong, the European Union, and the United States.

---

[6]https://www.sec.gov/answers/faqs.htm

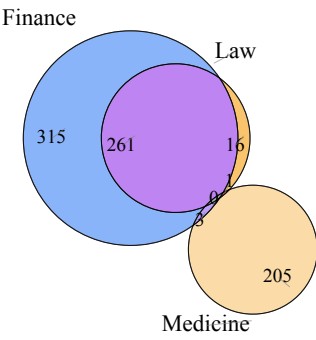 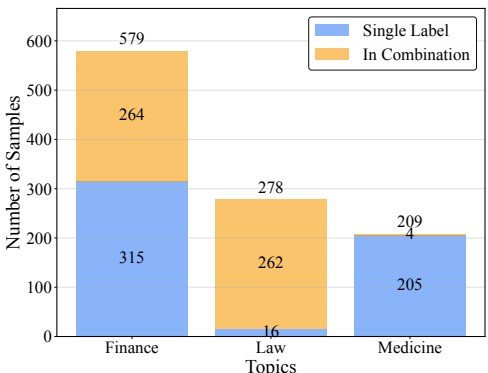

Figure 11: Topic Distribution.

To align the benchmark with topics of broad public interest, we further collect "trending" and "rising" terms from Google Trends[7], focusing on the "Finance" category and its subcategories in the United States and the United Kingdom over the five years preceding February 26, 2025. After de-duplication and filtering, this process yielded 272 unique finance-related topics. We manually classified these topics into three categories: institutions and products, policies and events, and concepts (for more detail, see Appendix I.1), and then constructed QA pairs through annotation. Eighteen topics were randomly selected to receive two QA pairs, yielding a total of 300 QA pairs.

We recruited six annotators, all graduate students in either Finance or Computer Science, and instructed them to formulate complete, non-redundant questions and answers for each topic by referring to authoritative sources such as official institutional websites, Encyclopædia Britannica, and Investopedia, and to record the source URL as supporting evidence. A rigorous verification process was then conducted to ensure accuracy, clarity, and consistency across all QA pairs, including spot-checking answers against their cited references.

### D.2 ADDITIONAL DETAILS FOR MULTI-TURN DIALOGUE GENERATION

To reduce annotation effort, we designed a set of expansion templates: $\mathcal{T}_2$ for two-turn dialogues ($|\mathcal{T}_2| = 8$) and $\mathcal{T}_3$ for three-turn dialogues ($|\mathcal{T}_3| = 10$).(for more detail, see Appendix J.1, all templates are displayed in the prompt) Using these templates,

Figure 12: Distribution of turn count

**Qwen2.5-32B-Instruct** was applied to automatically expand single-turn questions into two- and three-turn sequences. Prompts are provided in Appendix J.1. A subset of these expansions was then manually extended into four- and five-turn dialogues, ensuring natural progression and quality.

All generated question sequences were manually reviewed to avoid answer leakage, preserve the intent of the final question, and ensure cross-turn consistency. We also identified and revised cases where multiple questions corresponded to substantially overlapping supporting-fact sets, thereby maintaining diversity and factual coverage. Figure 12 reports the realized distribution.

---

[7]https://trends.google.com/trends/

```
Example of Multi-Turn Questions With Answer and Reference

{
      "sample_id": "9461cce2",
      "domain": ["finance"],
      "single-turn question": "Compared to living in other parts of the UK, what two
    additional documents are required when taxing a vehicle in Northern Ireland?",
      "multi-turn questions": [
        "How has vehicle taxation evolved in the United Kingdom over the past century
    ?",
        "What are the current vehicle taxation policies in place in the UK today?",
        "Compared to living in other parts of the UK, what two additional documents are
     required when taxing a vehicle in Northern Ireland?"
      ],
      "answer": "When taxing a vehicle at a Post Office in Northern Ireland, you need
    to provide two additional documents: a paper copy of your insurance certificate or
     cover note, and an original MOT test certificate or evidence of a Temporary
    Exemption Certificate (TEC).",
      "must_have": [
        "When taxing a vehicle at a Post Office in Northern Ireland, a paper copy of
    your insurance certificate is required",
        "When taxing a vehicle at a Post Office in Northern Ireland, a cover note is
    required as an alternative to an insurance certificate",
        "When taxing a vehicle at a Post Office in Northern Ireland, an original MOT
    test certificate is required",
        "When taxing a vehicle at a Post Office in Northern Ireland, evidence of a
    Temporary Exemption Certificate (TEC) is required as an alternative to an MOT test
     certificate"
      ],
      "source": "HOT-FINANCE-TOPIC",
      "url": [
        "https://www.gov.uk/vehicle-tax"
      ]
}
```

## D.3 ADDITIONAL DETAILS FOR AUTOMATIC EVALUATION

### D.3.1 VALIDATION OF THE EVALUATION PIPELINE

To validate our two-stage evaluation pipeline, we conducted a human annotation study, illustrated in Fig. 13. For the Atomic Decomposition stage, we sampled 100 model answers drawn from four representative LLMs of varying scales (Qwen2.5-14B-Instruct, Llama-3.1-8B-Instruct, DeepSeek-V3-0324, and GPT-4o-2024-08-06), covering single-turn or multi-turn settings. Each answer was decomposed into atomic statements by our decomposer model (Qwen2.5-32B-Instruct) and compared against human annotations. As shown in the top part of Fig. 13, the decomposition achieved high fidelity, with an SMAPE of **18.1%** in statement counts and an omission rate of only **5.9%**, indicating that discrepancies were mainly due to under-segmentation rather than semantic distortion.

For the Factual Consistency Judgment stage, the same 100 dialogues were decomposed into 1,687 evaluation items, consisting of atomic statements paired with the opposing full-text answers. These were labeled by our evaluator (Qwen2.5-14B-Instruct) and independently annotated by three human experts. As shown in the bottom part of Fig. 13, human annotators

Table 6: Performance comparison on the Factual Consistency Judgment stage.

| Model | Accuracy | F1 | Precision | Recall |
|---|---|---|---|---|
| Qwen-2.5-14B | **0.83** | **0.84** | 0.84 | **0.83** |
| GPT-4o | 0.81 | 0.82 | **0.84** | 0.81 |
| DeepSeek-V3 | 0.76 | 0.78 | 0.84 | 0.76 |

achieved substantial agreement (pairwise Cohens $\kappa$ values of 0.60, 0.62, and 0.63; Fleiss $\kappa = 0.62$). Furthermore, we constructed a gold standard by majority voting over the three human annotations. The agreement between this gold standard and individual annotators was consistently high, with Cohens $\kappa$ values of 0.80, 0.79, and 0.81, indicating strong alignment between the aggregated ground truth and expert annotations.

We then assessed the performance of various models against this gold standard, as detailed in Table 6. The results show that Qwen-2.5-14B achieves the most favorable performance among the candidates, leading in both accuracy and F1-score. Consequently, considering its strong performance and computational efficiency, we selected Qwen-2.5-14B as the designated evaluator for our

pipeline. We conjecture that this outcome may be attributed to the nature of the task; while larger models possess more powerful general reasoning abilities, they might be prone to overly complex inference paths for a constrained judgment task, potentially introducing instability. A well-calibrated, medium-sized model, such as Qwen-2.5-14B, may follow a more direct and consistent reasoning process, rendering it more reliable for this specific application.

### D.3.2 INVESTIGATING BIAS OF DECOMPOSER AND EVALUATOR

A critical consideration for our pipeline is whether the chosen evaluator, Qwen-2.5-14B, exhibits any bias, particularly a self-preference for models within its own family. To investigate this, we conducted a detailed analysis of the evaluation outcomes for each target model, with performance metrics presented in Table 7a. The metrics are tightly clustered across all models, with F1-scores ranging from 0.81 to 0.85. Notably, the model from the evaluator's own family, Qwen2.5-14B, does not receive a disproportionately high score.

Table 7: Analysis of Evaluator Impartiality. We find no evidence of bias, as confirmed by non-parametric tests showing statistically insignificant performance differences between the models.

(a) Performance breakdown per model, as judged by the Qwen-2.5-14B evaluator. OR refers to Ommision Rate.

| Model | Acc. | F1 | Prec. | Rec. | SMAPE(%) | OR(%) |
|---|---|---|---|---|---|---|
| DeepSeek-V3 | 0.85 | 0.85 | 0.87 | 0.85 | 13.12 | 6.34 |
| LLaMA3-8B | 0.84 | 0.85 | 0.86 | 0.84 | 20.66 | 2.54 |
| GPT-4o | 0.83 | 0.83 | 0.83 | 0.83 | 19.70 | 7.52 |
| Qwen2.5-14B | 0.81 | 0.81 | 0.81 | 0.81 | 18.10 | 5.90 |

(b) Pairwise significance tests (p-values from Mann-Whitney U tests) on model accuracy for Evaluator.

| Model | DS-V3 | L3-8B | GPT-4o | Q2.5-14B |
|---|---|---|---|---|
| DS-V3 | - | .761 ns | .445 ns | .183 ns |
| L3-8B | .761 ns | - | .674 ns | .347 ns |
| GPT-4o | .445 ns | .674 ns | - | .602 ns |
| Q2.5-14B | .183 ns | .347 ns | .602 ns | - |

*Note:* ns denotes $p \geq 0.01$.

To further validate this observation with a method robust to non-normal data distributions, we performed pairwise non-parametric Mann-Whitney U tests on the models' accuracy scores. As the answers for each question were generated by a single, randomly assigned model, the groups of scores for each model are independent, making this test appropriate. The results, summarized in Tab. 7b, show that all comparisons yield p-values well above the 0.01 threshold. Furthermore, the calculated effect sizes for all pairs were negligible ($r < 0.03$), indicating that the observed differences lack practical importance. In addition, a Kruskal-Wallis test yields $H = 1.9552, p = 5.82 \times 10^{-1}$, indicating no significant overall difference among models. These statistical evidences strongly support the conclusion that our evaluation pipeline operates impartially and does not systematically favor any specific model architecture or family.

Analogous to the evaluator check, we test whether the Qwen-based Decomposer systematically favors or penalizes some models when extracting atomic facts from generated answers. For each model, we compute decomposition count error against human gold using SMAPE, then perform (1) a KruskalWallis test across models and (2) pairwise Mann-Whitney U tests (see the results in Table 8). The Kruskal-Wallis test yields $H = 6.1147, p = 1.062 \times 10^{-1}$, showing no significant global difference across model families. Pairwise tests again indicate that no pair is significant at the stricter 0.01

Table 8: Pairwise significance tests (p-values from Mann-Whitney U tests) on model SMAPE for Decomposer.

| Model | DS-V3 | L3-8B | GPT-4o | Q2.5-14B |
|---|---|---|---|---|
| DS-V3 | - | .477 ns | .237 ns | .104 ns |
| L3-8B | .477 ns | - | .086 ns | .030 |
| GPT-4o | .237 ns | .086 ns | - | .841 ns |
| Q2.5-14B | .104 ns | .030 | .841 ns | - |

*Note:* ns denotes $p \geq 0.01$.

level, and only one pair reaches 0.05 with a small effect size. This suggests that any between-model variation in decomposition error is minor, and we find no strong evidence that the Decomposer systematically favors or penalizes specific model families.

### D.3.3 INVESTIGATING ROBUSTNESS OF DECOMPOSER AND EVALUATOR

To test the robustness of our evaluation pipeline. We conduct cross-model validation by replacing the original LLM with GPT-4o at two different stages of the pipeline, the decomposer and the evaluator, and then re-evaluate four representative LLMs (GPT-4o, DeepSeek-V3, Llama-3.3-70B, and

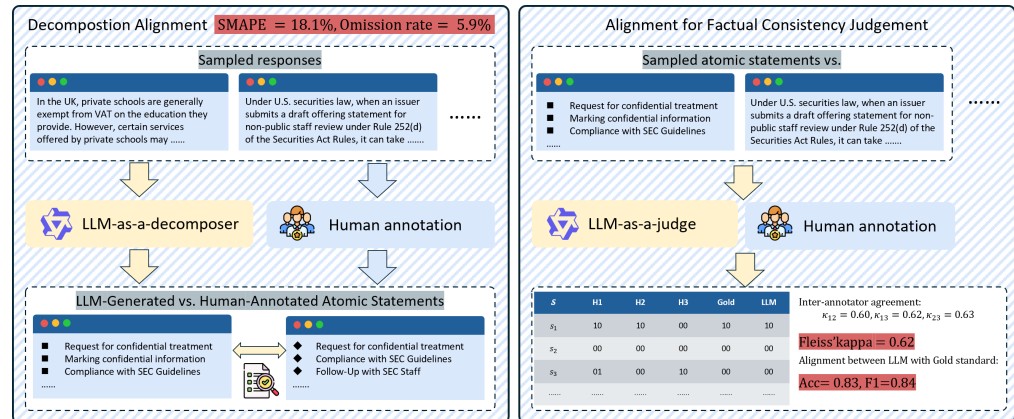

Figure 13: Human validation of our evaluation pipeline. (**left**) Alignment between LLM-based and human-annotated atomic decompositions. (**right**) Alignment between LLM-based and human annotations for factual consistency judgments.

Table 9: Model Ranking ($\mathbf{S_f}$) Consistency across Evaluators and Decomposer Variants. OR refers to Original Rank.

| OR | Model (Mode) | $\mathbf{S_f}$ | GPT-4o Eval (Rank) | GPT-4o Decomp (Rank) |
|---|---|---|---|---|
| 1 | GPT-4o (Single) | 28.77% | 38.06% (1) | 29.54% (1) |
| 2 | GPT-4o (Multi) | 27.63% | 37.86% (2) | 29.24% (2) |
| 3 | Llama-3.3-70B (Single) | 24.02% | 32.95% (4) | 24.41% (5) |
| 4 | Qwen-2.5-72B (Single) | 23.97% | 32.37% (5) | 24.47% (4) |
| 5 | DeepSeek-V3 (Single) | 23.80% | 33.82% (3) | 24.51% (3) |
| 6 | Llama-3.3-70B (Multi) | 21.16% | 29.26% (6) | 21.50% (6) |
| 7 | Qwen-2.5-72B (Multi) | 20.20% | 28.54% (8) | 20.97% (7) |
| 8 | DeepSeek-V3 (Multi) | 19.38% | 29.19% (7) | 19.54% (8) |

Qwen-2.5-72B) under both single-turn and multi-turn settings. The results are shown in Table 9 and Table 10.

**Stability of the Decomposer** We first test the stability of the Atomic Fact Decomposition stage by replacing the original Qwen-2.5-32B Decomposer with GPT-4o, while keeping the Evaluator fixed. we find near-perfect agreement: (i) Pearson correlation $r = 0.996$, this extremely high value indicates that the atomic facts extracted by the Qwen-based decomposer are highly aligned with those extracted by GPT-4o, and that the choice of decomposer explains almost all variance in the final scores. (ii) Spearman rank correlation $\rho = 0.905$, which confirms that changing the Decomposer does not significant alter the relative comparison between models.

**Robustness of the Evaluator** We further replace the original Qwen-2.5-14B Evaluator with GPT-4o, while keeping the Decomposer fixed, and compare the resulting factual F1 scores ($\mathbf{S_f}$) We observe strong statistical consistency between the two Evaluators: (i) Pearson correlation $r = 0.984$, which indicates an strong linear relationship between the two sets of scores. (ii) Spearman rank correlation $\rho = 0.905$, shows that the **relative ranking** of models is almost identical under the two Evaluators. Specifically, as shown in Table R1, both Evaluators (Qwen-based and GPT-4o-based) consistently agree that GPT-4o is the top-tier model, and they preserve the overall separation between the proprietary model and the open-source models in both single-turn and multi-turn evaluation. The rankings of the open-source models only exhibit minor permutations, without changing our qualitative conclusions.

In summray, our main conclusion, that models factual ability degrades in multi-turn dialogue, remains stable under all validation configurations. As shown in Table R2, regardless of whether we

Table 10: Cross-Model Validation  Detailed Metrics for Evaluator and Decomposer Variants.

| Model | Setting | Configuration | $S_f$ (%) | $S_h$ (%) | $D_f$ | $D_h$ |
|---|---|---|---|---|---|---|
| GPT-4o | Single | Original | 28.77 | 2.26 | 226 | 506 |
| GPT-4o | Single | w/ GPT-4o Eval | 38.06 | 3.36 | 179 | 434 |
| GPT-4o | Single | w/ GPT-4o Decomp | 29.54 | 2.55 | 226 | 507 |
| GPT-4o | Multi | Original | 27.63 | 2.48 | 254 | 555 |
| GPT-4o | Multi | w/ GPT-4o Eval | 37.86 | 3.11 | 214 | 532 |
| GPT-4o | Multi | w/ GPT-4o Decomp | 29.24 | 2.26 | 253 | 495 |
| DeepSeek-V3 | Single | Original | 23.80 | 3.21 | 343 | 740 |
| DeepSeek-V3 | Single | w/ GPT-4o Eval | 33.82 | 4.19 | 285 | 722 |
| DeepSeek-V3 | Single | w/ GPT-4o Decomp | 24.51 | 3.32 | 340 | 739 |
| DeepSeek-V3 | Multi | Original | 19.38 | 3.40 | 444 | 888 |
| DeepSeek-V3 | Multi | w/ GPT-4o Eval | 29.19 | 4.32 | 373 | 837 |
| DeepSeek-V3 | Multi | w/ GPT-4o Decomp | 19.54 | 3.40 | 445 | 877 |
| Llama-3.3-70B | Single | Original | 24.02 | 3.49 | 324 | 628 |
| Llama-3.3-70B | Single | w/ GPT-4o Eval | 32.95 | 5.12 | 272 | 628 |
| Llama-3.3-70B | Single | w/ GPT-4o Decomp | 24.41 | 3.58 | 326 | 653 |
| Llama-3.3-70B | Multi | Original | 21.16 | 4.25 | 437 | 760 |
| Llama-3.3-70B | Multi | w/ GPT-4o Eval | 29.26 | 6.00 | 383 | 732 |
| Llama-3.3-70B | Multi | w/ GPT-4o Decomp | 21.50 | 4.16 | 435 | 775 |
| Qwen-2.5-72B | Single | Original | 23.97 | 2.44 | 358 | 821 |
| Qwen-2.5-72B | Single | w/ GPT-4o Eval | 32.37 | 3.92 | 334 | 791 |
| Qwen-2.5-72B | Single | w/ GPT-4o Decomp | 24.47 | 2.49 | 366 | 822 |
| Qwen-2.5-72B | Multi | Original | 20.20 | 2.77 | 484 | 900 |
| Qwen-2.5-72B | Multi | w/ GPT-4o Eval | 28.54 | 4.05 | 434 | 872 |
| Qwen-2.5-72B | Multi | w/ GPT-4o Decomp | 20.97 | 2.86 | 484 | 900 |

replace the Evaluator, the Decomposer, or both, all models (including GPT-4o) exhibit a consistent drop in factual metrics ($R_f$, $P_f$, $S_f$) when moving from the single-turn to the multi-turn setting. Furthermore, the evaluation framework of KnowMT-Bench is not tied to a specific architecture. While we chose the Qwen family primarily for reproducibility and computational efficiency, our cross-model validation demonstrates that using a stronger proprietary model (GPT-4o) in place of Qwen leads to the same qualitative conclusions.

# E QUALITY ASSURANCE AND ANNOTATION WORKLOAD

We designed a workflow to ensure data usability and reliability. The workflow covered construction, alignment, and evaluation for both single-turn and multi-turn data, and we recorded human effort and time cost at each stage. Clear guidelines were enforced at each stage: single-turn data required "complete, non-redundant and multi-point" answers; multi-turn data required "no leakage, intent preservation, and cross-turn consistency"; atomic-level fact decomposition required "no omission, no over-bundling, no fabrication." Across all stages, annotation and verification were performed by independent individuals, with results cross-checked to ensure reliability. All annotators and checkers had backgrounds in finance or computer science.

## E.1 SINGLE-TURN LFQA ANNOTATION (300 FINANCE TOPICS)

We collected 300 new daily financial QA pairs (1-2 per topic) based on Google Trends topics. First, annotators retrieved authoritative sources (through official websites related to these topics or expert-verified encyclopedic sites such as Investopedia or the Encyclopedia Britannica) and wrote complete, non-redundant multi-point LFQA pairs based on these sources. Each QA pair required

10-20 minutes on average. A few difficult items took more than 30 minutes. Six graduate annotators participated. The total effort was about **80 person-hours**. All 300 pairs were manually checked against authoritative sources in the final review.

For items from SEC FAQ or FinTextQA, we performed consistency checks: removing trivial yes/no or phrase-level answers; adding necessary jurisdictional context (e.g., "Hong Kong"). These edits ensure alignment with the task definition of "complete, non-redundant and multi-point"

### E.2 MULTI-TURN DIALOGUE GENERATION AND REVIEW

We expanded single-turn questions into 2-3 turns question sequences using templates $\mathcal{T}_2, \mathcal{T}_3$ with **Qwen2.5-32B-Instruct**. Part of the question sequences were manually extended to 4-5 turns. All dialogues were reviewed to ensure no leakage, intent preservation, and cross-turn consistency. Three annotators participated in this stage, and their results were cross-checked to ensure consistency.

### E.3 ATOMIC FACT CONSTRUCTION AND ALIGNMENT

We decomposed ground-truth answers into *atomic* factual statements (facts) with **Qwen2.5-32B-Instruct**, followed by manual alignment in two rounds: **Round 1:** three annotators checked for missing information, under-decomposition (multiple claims in one), or extraneous content (unsupported). Average 10-20 minutes per item; about **200 person-hours** total. **Round 2:** the same three annotators cross-reviewed each other's annotations. They examined one another's outputs, discussed any disagreements to reach consensus, and updated the annotations accordingly, taking about **100 person-hours**.

### E.4 HUMAN BENCHMARK FOR CONSISTENCY EVALUATION

Three annotators independently labeled atomic statement pairs (atomic statement vs. free-form answer). From 100 dialogues, we obtained 1,687 atomic evaluation items (Fig. 13). Inter-annotator agreement was substantial: pairwise Cohen's $\kappa$ of **0.60**, **0.62**, and **0.63**; Fleiss' $\kappa = $ **0.62**. Gold labels were created by majority vote. Agreement between individual annotators and the gold labels was higher: Cohen's $\kappa$ of **0.80**, **0.79**, and **0.81** (mean **0.80**), showing the gold labels are *highly consistent* with each expert judgment.

### E.5 SUMMARY

Across the multi-stage annotation workflow, we enforced actionable guidelines at every step and adopted an "annotation, independent review, and cross-check" loop to control bias and leakage risks. Concretely: single-turn LFQA required "complete, non-redundant, multi-point" answers; multi-turn dialogues emphasized "no leakage, intent preservation, and cross-turn consistency"; atomic fact decomposition enforced "no omission, no over-bundling, no fabrication." All stages were carried out by individuals with finance or computer science backgrounds, with independent annotators and checkers mutually validating each others work. Taken together, these procedures yield a high-quality benchmark dataset, covering single- and multi-turn settings with atomic fact alignments and human gold labels. The dataset is suitable for automated factuality evaluation and conducive to reproducibility and extension.

## F EMPIRICAL VALIDATION OF DIALOGUE NATURALNESS

While our data construction pipeline incorporates rigorous human-in-the-loop review (as detailed in Appendix D), relying on LLM-assisted expansion for multi-turn sequences entails a potential risk of introducing synthetic artifacts or stylistic biases. To assess the naturalness of our generated dialogues and ensure they align with real-world interaction patterns, we conducted a **Complex Scenario Suitability Study** following the protocol established by StructFlowBench (Li et al., 2025).

To ensure an equitable comparison, we aligned our evaluation protocol with established baselines. Given that the reference benchmark, StructFlowBench (Li et al., 2025) and MT-Bench-101 (Bai et al., 2024), utilize GPT-4o for dialogue generation, we similarly selected the dialogue generated

by GPT-4o within KNOWMT-BENCH for this assessment. Following the experimental setup of StructFlowBench, we randomly sampled 50 dialogues from each benchmark and employed GPT-4o as an impartial evaluator. The assessment was conducted across three distinct dimensions derived from the StructFlowBench rubric, rated on a Likert scale of 1–5:

- **Logical Coherence:** Measures whether the dialogue maintains semantic consistency and logical progression without abrupt or unjustified contextual shifts.
- **Goal Clarity:** Assesses the transparency of the user's intent and the system's adherence to the task objective throughout the interaction.
- **Transition Naturalness:** Evaluates the fluidity of inter-turn transitions, specifically detecting mechanical or forced phrasings typical of synthetic text.

Furthermore, to quantify the distributional alignment with high-quality data, we also introduce the **Confusion Factor (CF)** from StructFlowBench. This composite indicator is defined as the proportion of dialogue samples that achieve an average score of $\geq 4.0$ across the three dimensions, serving as a robust proxy for the density of high-fidelity interactions.

The comparative results are presented in Table 11. KNOWMT-BENCH achieves an overall mean score of 4.43, which is comparable to StructFlowBench (4.47) and MT-Bench-101 (4.51). Crucially, in terms of the Confusion Factor, our benchmark scores **0.78**, surpassing MT-Bench-101 (0.74) and approaching the specialized StructFlowBench (0.82).

These findings substantiate that the multi-turn sequences in KNOWMT-BENCH, despite utilizing model-assisted expansion, possess high logical coherence and transition fluidity. The high CF score further confirms that our human verification protocols effectively filtered out low-quality synthetic artifacts, yielding a benchmark that faithfully reflects the complexity and naturalness of real-world knowledge-intensive interaction between humans and LLMs.

Table 11: Results of the Complex Scenario Suitability Study. We compare KNOWMT-BENCH against StructFlowBench and MT-Bench-101 across three qualitative dimensions and the composite Confusion Factor (CF). All scores are on a 1-5 scale except for CF, which is a ratio.

| Dataset | Logical Coherence | Goal Clarity | Transition Naturalness | Overall Mean | Confusion Factor (CF) |
|---|---|---|---|---|---|
| StructFlowBench (Li et al., 2025) | 4.78 | 4.46 | 4.18 | 4.47 | 0.82 |
| MT-Bench-101 (Bai et al., 2024) | 4.50 | 4.52 | 4.50 | 4.51 | 0.74 |
| **KnowMT-Bench (Ours)** | 4.72 | 4.46 | 4.12 | 4.43 | 0.78 |

## G  DETAILED EXPERIMENT SETTING

**Basic Settings**  Based on the dataset construction process outlined in the previous section, we generat both multi-turn and single-turn dialogues based on questions. For multi-turn experiments, we use each model's *chat template* to format dialogue history and we set *max new tokens* to **1024** for each round. For Gemini, we restricted the chain-of-thought (CoT) output length to **256**. To ensure reproducibility, we applied **greedy decoding** (temperature=0, top_p=1), disabling sampling and beam search. For models with CoT reasoning, we standardized answer extraction: (i) if a clear final answer is present, only that answer is retained for evaluation; (ii) if no explicit answer is generated (for example, due to truncation), the entire reasoning output is treated as the answer. This policy was applied uniformly in both multi-turn and single-turn settings.

**RAG Settings**  For retrieval-augmented generation tasks, we adopt a two-stage retrieve–then–rerank pipeline. Each QA instance was associated with its own reference, which served as the *retrieval candidate pool*. Texts in this pool were segmented using the **SentenceSplitter** from llama-index[8] with a chunk size of 512 and an overlap of 128, ensuring consistent coverage of context across

---

[8] https://github.com/run-llama/llama_index

Table 12: Results for all models. Sections explicitly indicate **Single-turn**, **Multi-turn**, and the separate **Diff** block (Multi vs. Single). **Bold** values mark, *within each block*, the best score per column (for ↑ higher is better; for ↓ lower is better). Green in **Diff** indicates improvement and red indicates degradation.

| Model | Factuality | | | Hallucination | | | Efficiency | | |
|---|---|---|---|---|---|---|---|---|---|
| | $R_f$ (%) ↑ | $P_f$ (%) ↑ | $S_f$ (%) ↑ | $R_m$ (%) ↓ | $P_{fc}$ (%) ↓ | $S_h$ (%) ↓ | $D_f$ ↓ | $D_h$ ↑ | $D_R$ ↓ |
| **Single-turn** | | | | | | | | | |
| Gemini-2.5-Pro | **57.32** | 17.79 | 23.15 | **3.30** | 10.96 | 3.03 | 671 | **1623** | 4267 |
| GPT-4o | 42.08 | **31.81** | **28.77** | 3.40 | 6.87 | **2.26** | **226** | 506 | 1992 |
| GPT-4o-mini | 39.50 | 30.60 | 26.85 | 4.00 | 6.55 | 2.40 | 254 | 548 | **1684** |
| DeepSeek-R1 | 52.02 | 17.48 | 21.55 | 4.34 | 13.93 | 4.11 | 603 | 1228 | 5890 |
| DeepSeek-V3 | 48.51 | 20.68 | 23.80 | 3.79 | 9.58 | 3.21 | 343 | 740 | 3065 |
| Qwen-2.5-72B | 43.21 | 22.50 | 23.97 | 3.45 | 8.20 | 2.44 | 358 | 821 | 3163 |
| Llama-3.3-70B | 41.59 | 23.85 | 24.02 | 4.87 | 8.99 | 3.49 | 324 | 658 | 2548 |
| Qwen-2.5-32B | 37.72 | 23.30 | 22.50 | 3.40 | 8.58 | 2.57 | 367 | 685 | 2832 |
| QwQ-32B | 38.35 | 23.83 | 20.96 | 5.37 | **6.54** | 2.73 | 531 | 919 | 7673 |
| Qwen-2.5-14B | 36.77 | 24.05 | 22.65 | 4.14 | 9.33 | 3.31 | 295 | 513 | 2674 |
| Llama-3.1-8B | 32.98 | 20.17 | 18.02 | 11.94 | 8.52 | 4.01 | 405 | 589 | 3609 |
| Qwen-2.5-7B | 34.38 | 20.55 | 19.67 | 4.67 | 9.09 | 3.57 | 366 | 589 | 3171 |
| **Multi-turn** | | | | | | | | | |
| Gemini-2.5-Pro | **51.59** | 17.23 | 21.62 | 4.77 | 12.34 | 4.33 | 695 | **1417** | 5109 |
| GPT-4o | 42.68 | **28.51** | **27.63** | 3.55 | **7.05** | 2.48 | **254** | 555 | **2439** |
| GPT-4o-mini | 41.10 | 26.53 | 25.54 | 4.08 | 8.04 | 2.87 | 344 | 782 | 2702 |
| DeepSeek-R1 | 50.67 | 15.17 | 19.67 | 4.51 | 13.97 | 4.32 | 627 | 1177 | 4445 |
| DeepSeek-V3 | 49.06 | 15.65 | 19.38 | 3.96 | 10.12 | 3.40 | 444 | 888 | 5213 |
| Qwen-2.5-72B | 44.51 | 17.59 | 20.20 | **3.33** | 9.28 | 2.77 | 484 | 900 | 3749 |
| Llama-3.3-70B | 41.34 | 20.15 | 21.16 | 5.46 | 10.41 | 4.25 | 437 | 760 | 3170 |
| Qwen-2.5-32B | 40.80 | 20.34 | 21.62 | 3.53 | 9.14 | 3.02 | 435 | 746 | 2968 |
| QwQ-32B | 39.00 | 19.78 | 19.17 | 5.38 | 10.03 | 3.92 | 473 | 815 | 5250 |
| Qwen-2.5-14B | 39.17 | 21.11 | 21.60 | 4.03 | 10.07 | 3.22 | 425 | 824 | 3672 |
| Llama-3.1-8B | 34.67 | 18.85 | 18.35 | 7.25 | 9.39 | 4.13 | 479 | 753 | 4105 |
| Qwen-2.5-7B | 36.21 | 15.68 | 17.15 | 4.33 | 10.65 | 3.77 | 547 | 847 | 4476 |
| **Diff (Multi vs. Single)** | | | | | | | | | |
| Gemini-2.5-Pro | -10.0% | -3.2% | -6.6% | +44.8% | +12.5% | +42.6% | +3.6% | -12.7% | +19.7% |
| GPT-4o | +1.4% | -10.4% | -4.0% | +4.2% | +2.6% | +9.6% | +12.2% | +9.6% | +22.4% |
| GPT-4o-mini | +4.1% | -13.3% | -4.9% | +1.9% | +22.8% | +19.5% | +35.3% | +42.9% | +60.4% |
| DeepSeek-R1 | -2.6% | -13.2% | -8.7% | +3.9% | +0.3% | +5.2% | +4.0% | -4.1% | -24.5% |
| DeepSeek-V3 | +1.1% | -24.3% | -18.6% | +4.7% | +5.6% | +5.9% | +29.4% | +20.0% | +70.1% |
| Qwen-2.5-72B | +3.0% | -21.8% | -15.7% | -3.4% | +13.3% | +13.6% | +35.4% | +9.7% | +18.5% |
| Llama-3.3-70B | -0.6% | -15.5% | -11.9% | +12.1% | +15.7% | +21.8% | +34.9% | +15.4% | +24.4% |
| Qwen-2.5-32B | +8.2% | -12.7% | -3.9% | +3.8% | +6.5% | +17.6% | +18.6% | +8.8% | +4.8% |
| QwQ-32B | +1.7% | -17.0% | -8.5% | +0.2% | +53.4% | +43.7% | -10.9% | -11.3% | -31.6% |
| Qwen-2.5-14B | +6.5% | -12.2% | -4.6% | -2.6% | +8.0% | -2.7% | +44.2% | +60.5% | +37.3% |
| Llama-3.1-8B | +5.1% | -6.5% | +1.8% | -39.3% | +10.2% | +2.8% | +18.2% | +27.8% | +13.7% |
| Qwen-2.5-7B | +5.3% | -23.7% | -12.8% | -7.4% | +17.1% | +5.6% | +49.2% | +43.9% | +41.1% |

segments. We use Qwen3-Embedding-0.6B[9] as the embedding model, and Qwen3-Reranker-0.6B[10] as the reranker (Zhang et al., 2025a). We chose these two models because some of our experimental setups require the entire dialogue history as the query, which imposes relatively high GPU memory demands, hence, we opted for smaller but more recent models that strike a strong balance between efficiency and performance. We first retrieve **15** candidate chunks from this pool using FAISSs IndexFlatL2 over L2-normalized embeddings, then reranked and selected the top **5** chunks for prompt construction. Retrieved chunks were formatted as numbered triple-quoted blocks and concatenated with the user query, and the models were instructed to answer strictly based on this context.

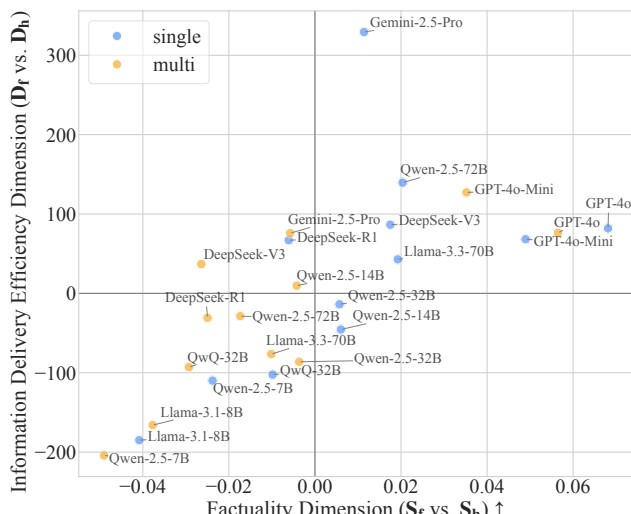

Figure 14: Model performance mapped onto the four-quadrant framework. The x-axis captures overall factuality, derived from the correlation between $S_f$ and $S_h$. The y-axis measures performance against the efficiency trade-off baseline, derived from the correlation between $D_f$ and $D_h$.

## H ADDITIONAL EXPERIMENTS

### H.1 NUMERICAL RESULTS IN MAIN EXPERIMENT

Table 12 presents the detailed numerical results for all models across the three evaluation dimensions. A general trend observed is a decline in performance for most models across a majority of metrics when transitioning from single-turn to multi-turn conversations. Notably, Factual Recall ($R_f$) and the token cost per hallucinated fact ($D_h$) are exceptions, showing improvements for most models. This may suggest that while multi-turn interactions prompt models to be more comprehensive and cover more ground-truth facts, this often comes at the cost of reduced precision and greater verbosity, which in turn dilutes the density of factual errors.

### H.2 FOUR-QUADRANT ANALYSIS

While the main text analyzes performance through separate scatter plots (Figure 4), the four-quadrant framework in Figure 14 offers a synthesized, holistic view. This framework is constructed from the two empirically established regression baselines discussed in Section 4.2.

The framework defines two new orthogonal dimensions based on these trends. The **Factuality Dimension (x-axis)** is derived from the synergistic relationship between $S_f$ and $S_h$ (Figure 4a). The goal is to create a single score where movement towards the plot's bottom-right region (higher $S_f$, lower $S_h$) is considered an improvement. Since the data's primary trend, captured by the top-left-to-bottom-right regression line, aligns with this desired trajectory, we use it as a directional basis. To quantify progress along this trajectory, we construct a baseline that is **perpendicular** to the regression line and passes through the data centroid. The 'Factuality Dimension' score is then the signed distance of each data point **to** this perpendicular baseline, with points on the bottom-right side receiving higher scores. This metric thus holistically captures a model's overall factuality.

The **Information Delivery Efficiency Dimension (y-axis)** is derived from the relationship between $D_f$ and $D_h$ (Figure 4b). Its value is the vertical residual: the signed distance from a data point to the regression line along the $D_h$ axis. Points above the line have positive values. A positive score thus quantifies the degree to which a model's token cost per contradicted fact ($D_h$) exceeds the

---

[9]https://huggingface.co/Qwen/Qwen3-Embedding-0.6B
[10]https://huggingface.co/Qwen/Qwen3-Reranker-0.6B

expectation set by its cost per correct fact ($\mathbf{D_f}$), measuring its ability to push the efficiency trade-off boundary. The detailed mathematical derivation of these dimensions is provided in Appendix H.3.

This synthesized view crystallizes the model behaviors discussed previously. For instance, **GPT-4o** occupies the right-hand side of the plot, confirming its strong factuality profile (high x-axis value) and its adherence to the established efficiency trade-off (y-axis value near zero). In stark contrast, **Gemini-2.5-Pro** (in the single-turn setting) is distinguished by a high positive y-axis value, visually confirming its status as a significant outlier that pushes the boundary, as its $\mathbf{D_h}$ is exceptionally high for its given $\mathbf{D_f}$. The **Qwen family** exhibits clear scaling effects, with larger models generally moving towards the upper-right. The transition to the more demanding multi-turn dialogue setting, however, challenges the models, causing a noticeable shift for most towards the lower-left, underscoring a degradation in both their overall factuality and their performance relative to the efficiency baseline.

### H.3 ON THE NON-TRIVIALITY OF OBSERVED METRIC CORRELATIONS

We demonstrate that the empirically observed correlations specifically, the negative correlation between factuality ($\mathbf{S_f}$) and hallucination ($\mathbf{S_h}$), and the positive correlation between efficiency metrics ($\mathbf{D_f}$ and $\mathbf{D_h}$)are non-trivial findings about model behavior, not mathematical artifacts of the metric definitions.

**Factuality vs. Hallucination ($\mathbf{S_f}$ vs. $\mathbf{S_h}$)** Consider $\mathbf{S_f} = \mathrm{HM}(\mathbf{P_f}, \mathbf{R_f})$ and $\mathbf{S_h} = \mathrm{HM}(\mathbf{P_{fc}}, \mathbf{R_m})$. At the instance level, their components are constrained because a ground-truth fact cannot be simultaneously supported and contradicted ($\mathcal{F}_j^+ \cap \mathcal{F}_j^- = \varnothing$), and a generated statement cannot be simultaneously correct and false ($\mathcal{S}_j^+ \cap \mathcal{S}_j^- = \varnothing$). This imposes constraints such as $\mathbf{R_f} + \mathbf{R_m} \leq 1$ and $\mathbf{P_f} + \mathbf{P_{fc}} \leq 1$.

However, these "sum-to-at-most-one" constraints on the components do not enforce a necessary monotonic relationship between the final F1 scores, $S_f$ and $S_h$. We demonstrate this with a minimal counterexample. Consider a dataset with one instance ($|\mathcal{D}| = 1$), 10 ground-truth facts ($|\mathcal{F}| = 10$), and a model generating 10 statements ($|\mathcal{S}| = 10$).

- **Case A:** The model correctly covers 5 facts with 5 statements and makes no contradictions ($|\mathcal{F}^+| = 5, |\mathcal{S}^+| = 5, |\mathcal{F}^-| = 0, |\mathcal{S}^-| = 0$). This yields $\mathbf{R_f} = 0.5, \mathbf{P_f} = 0.5 \implies \mathbf{S_f} = 0.5$, and $\mathbf{R_m} = 0, \mathbf{P_{fc}} = 0 \implies \mathbf{S_h} = 0$.

- **Case B:** The model's factuality remains the same ($|\mathcal{F}^+| = 5, |\mathcal{S}^+| = 5 \implies \mathbf{S_f} = 0.5$), but its other 5 statements are now false and contradict 5 distinct ground-truth facts ($|\mathcal{F}^-| = 5, |\mathcal{S}^-| = 5$). This yields $\mathbf{R_m} = 0.5, \mathbf{P_{fc}} = 0.5 \implies \mathbf{S_h} = 0.5$.

Since $\mathbf{S_f}$ can remain constant while $\mathbf{S_h}$ varies, no deterministic relationship (e.g., $\mathbf{S_h} = f(\mathbf{S_f})$) is imposed by the metric design. Therefore, the empirically observed negative correlation reflects a genuine behavioral pattern of current models, not an algebraic necessity.

**Information Delivery Efficiency ($D_f$ vs. $D_h$)** A similar analysis applies to the efficiency metrics, $\mathbf{D_f}(j) = \frac{T(a_j)}{|\mathcal{F}_j^+|}$ and $\mathbf{D_h}(j) = \frac{T(a_j)}{|\mathcal{F}_j^-|}$. While they share the same numerator (token count $T(a_j)$), their denominators are controlled by disjoint sets of ground-truth facts, $|\mathcal{F}_j^+|$ and $|\mathcal{F}_j^-|$, which are only loosely constrained by $|\mathcal{F}_j^+| + |\mathcal{F}_j^-| \leq |\mathcal{F}_j|$. A necessary positive correlation can be falsified by demonstrating that one metric can be held constant while the other varies.

Consider a fixed-length response with $T(a) = 100$ tokens and a context of $|\mathcal{F}| = 10$ facts.

- Let's fix the number of correctly covered facts at $|\mathcal{F}^+| = 5$. This fixes $\mathbf{D_f} = 100/5 = 20$. The number of contradicted facts, $|\mathcal{F}^-|$, can still vary from 0 to 5. As $|\mathcal{F}^-|$ changes, $\mathbf{D_h}$ takes values from $\infty$ (or the smoothed maximum) down to $100/5 = 20$. Thus, $\mathbf{D_f}$ is constant while $\mathbf{D_h}$ varies.

- Conversely, let's fix the number of contradicted facts at $|\mathcal{F}^-| = 2$. This fixes $\mathbf{D_h} = 100/2 = 50$. The number of correct facts, $|\mathcal{F}^+|$, can still vary from 0 to 8. As $|\mathcal{F}^+|$ changes, $\mathbf{D_f}$ varies from $\infty$ down to $100/8 = 12.5$. Thus, $\mathbf{D_h}$ is constant while $\mathbf{D_f}$ varies.

Table 13: Comparison of model performance using traditional n-gram-based metrics and our proposed core metrics. This table provides the detailed numerical results that complement the scatter plot analysis. Green indicates improvement, while red indicates degradation.

| Model | N-gram Based Metrics | | | | Our Proposed Core Metrics | | | |
|---|---|---|---|---|---|---|---|---|
| | BLEU − 4 ↑ | R − 1 ↑ | R − 2 ↑ | R − L ↑ | $S_f$ (%) ↑ | $S_h$ (%) ↓ | $D_f$ ↓ | $D_h$ ↑ |
| **Single-turn** | | | | | | | | |
| Gemini-2.5-Pro | 1.51 | 13.37 | 3.73 | 12.46 | 23.15 | 3.03 | 671 | 1623 |
| GPT-4o | 5.43 | 26.13 | 8.71 | 23.64 | 28.77 | 2.26 | 226 | 506 |
| GPT-4o-mini | 4.79 | 25.03 | 7.89 | 22.57 | 26.85 | 2.40 | 254 | 548 |
| DeepSeek-R1 | 1.55 | 14.21 | 3.75 | 13.25 | 21.55 | 4.11 | 603 | 1228 |
| DeepSeek-V3 | 2.29 | 17.94 | 4.64 | 16.52 | 23.80 | 3.21 | 343 | 740 |
| Qwen-2.5-72B | 3.52 | 21.88 | 6.77 | 19.95 | 23.97 | 2.44 | 358 | 821 |
| Llama-3.3-70B | 4.19 | 23.19 | 7.54 | 21.02 | 24.02 | 3.49 | 324 | 658 |
| Qwen-2.5-32B | 3.47 | 21.82 | 6.21 | 19.66 | 22.50 | 2.57 | 367 | 685 |
| QwQ-32B | 2.16 | 17.43 | 4.39 | 16.09 | 20.96 | 2.73 | 531 | 919 |
| Qwen-2.5-14B | 3.74 | 22.45 | 6.42 | 20.24 | 22.65 | 3.31 | 295 | 513 |
| Llama-3.1-8B | 3.72 | 23.57 | 7.74 | 21.24 | 18.02 | 4.01 | 405 | 589 |
| Qwen-2.5-7B | 3.24 | 21.33 | 6.08 | 19.35 | 19.67 | 3.57 | 366 | 589 |
| **Multi-turn** | | | | | | | | |
| Gemini-2.5-Pro | 1.38 | 13.27 | 3.45 | 12.34 | 21.62 | 4.33 | 695 | 1417 |
| GPT-4o | 4.12 | 23.22 | 7.18 | 21.10 | 27.63 | 2.48 | 254 | 555 |
| GPT-4o-mini | 3.34 | 21.49 | 6.40 | 19.57 | 25.54 | 2.87 | 344 | 782 |
| DeepSeek-R1 | 1.28 | 12.88 | 3.08 | 12.04 | 19.67 | 4.32 | 627 | 1177 |
| DeepSeek-V3 | 1.39 | 14.10 | 3.01 | 13.11 | 19.38 | 3.40 | 444 | 888 |
| Qwen-2.5-72B | 2.18 | 18.05 | 5.25 | 16.63 | 20.20 | 2.77 | 484 | 900 |
| Llama-3.3-70B | 3.06 | 21.09 | 6.46 | 19.28 | 21.16 | 4.25 | 437 | 760 |
| Qwen-2.5-32B | 2.57 | 18.97 | 5.41 | 17.36 | 21.62 | 3.02 | 435 | 746 |
| QwQ-32B | 2.10 | 17.25 | 4.04 | 15.74 | 19.17 | 3.92 | 473 | 815 |
| Qwen-2.5-14B | 2.91 | 19.75 | 5.75 | 18.04 | 21.60 | 3.22 | 425 | 824 |
| Llama-3.1-8B | 2.86 | 21.76 | 6.64 | 19.85 | 18.35 | 4.13 | 479 | 753 |
| Qwen-2.5-7B | 2.18 | 17.89 | 4.97 | 16.57 | 17.15 | 3.77 | 547 | 847 |
| **Diff (Multi vs. Single)** | | | | | | | | |
| Gemini-2.5-Pro | -8.6% | -0.7% | -7.4% | -1.0% | -6.6% | +42.6% | +3.6% | -12.7% |
| GPT-4o | -24.1% | -11.1% | -17.6% | -10.7% | -4.0% | +9.6% | +12.4% | +9.7% |
| GPT-4o-mini | -30.3% | -14.2% | -18.8% | -13.3% | -4.9% | +19.5% | +35.4% | +42.7% |
| DeepSeek-R1 | -17.7% | -9.3% | -17.9% | -9.1% | -8.7% | +5.2% | +4.0% | -4.2% |
| DeepSeek-V3 | -39.3% | -21.4% | -35.1% | -20.6% | -18.6% | +5.9% | +29.4% | +20.0% |
| Qwen-2.5-72B | -38.0% | -17.5% | -22.5% | -16.7% | -15.7% | +13.6% | +35.2% | +9.6% |
| Llama-3.3-70B | -27.1% | -9.1% | -14.4% | -8.3% | -11.9% | +21.8% | +34.9% | +15.5% |
| Qwen-2.5-32B | -25.9% | -13.1% | -12.9% | -11.7% | -3.9% | +17.6% | +18.5% | +8.9% |
| QwQ-32B | -2.8% | -1.0% | -7.9% | -2.1% | -8.5% | +43.7% | -10.9% | -11.3% |
| Qwen-2.5-14B | -22.1% | -12.0% | -10.5% | -10.9% | -4.6% | -2.7% | +44.1% | +60.6% |
| Llama-3.1-8B | -23.1% | -7.7% | -14.2% | -6.5% | +1.8% | +2.8% | +18.3% | +27.8% |
| Qwen-2.5-7B | -32.7% | -16.1% | -18.2% | -14.4% | -12.8% | +5.6% | +49.5% | +43.8% |

This independence shows that the shared numerator $T(a_j)$ is a potential confounding variable but does not create a deterministic relationship. The observed strong positive correlation between $\mathbf{D_f}$ and $\mathbf{D_h}$ is therefore an empirical finding about models' tendency towards uniform verbosity, not a mathematical artifact.

**Effect of Smoothing** Our smoothing procedure for zero-denominator cases (e.g., when $|\mathcal{F}_j^+| = 0$) replaces the undefined value with a dataset-level maximum. This imputes a constant for the metric on that specific instance, which does not establish a functional link between metrics. In summary, any observed systematic correlation between $(\mathbf{S_f}, \mathbf{S_h})$ or $(\mathbf{D_f}, \mathbf{D_h})$ should be interpreted as an empirical pattern reflecting inherent trade-offs in model behavior, not as a mechanical coupling arising from the metric design.

## H.4 SIGNIFICANCE IN MAIN EXPERIMENT

To formally test the directional changes observed in the multi-turn setting (Section 4.2), we conduct a series of one-sided Wilcoxon signed-rank tests across all evaluation metrics. For each metric, we pre-specify an expected direction that reflects the qualitative trends discussed in the main text: a degradation in factuality ($\mathbf{S_f}$), hallucination ($\mathbf{S_h}$), and efficiency for correct facts ($\mathbf{D_f}$), and a potential *improvement* in the sparsity of hallucinated facts ($\mathbf{D_h}$). Concretely, the null hypothesis ($H_0$)

Table 14: Overall results of the Wilcoxon signed-rank tests for Figure 4. Here, "Correct" means that the observed difference between Multi-turn and Single-turn is aligned with the pre-defined expected direction: Multi-turn $<$ Single-turn for $S_f$ (degradation in factuality) and Multi-turn $>$ Single-turn for $S_h$ and $D_f$ (degradation in hallucination and efficiency for correct facts) as well as for $D_h$ (an improvement, i.e., larger token cost per hallucinated fact). "Significant" means one-sided $p < 0.05$. Most models move in the expected direction and are significant on $S_f$, $D_f$, and $D_h$, whereas for $S_h$ the direction is usually correct but the significance is weaker, mainly because $S_h$ values are close to 0 for many models under both settings.

| Metric | Correct & significant | Correct & not significant | Opposite & significant | Opposite & not significant |
|---|---|---|---|---|
| $S_f$ | 10/12 | 1/12 | 0/12 | 1/12 |
| $S_h$ | 3/12 | 8/12 | 0/12 | 1/12 |
| $D_f$ | 10/12 | 1/12 | 0/12 | 1/12 |
| $D_h$ | 9/12 | 0/12 | 0/12 | 3/12 |

Table 15: Wilcoxon signed-rank test results for $S_f$ (one-sided). The expected direction is Multi-turn $<$ Single-turn. The column "Aligned with expectation?" indicates whether the mean difference between Multi-turn and Single-turn follows this expected direction. The "Significance" column uses *, ** and *** to denote one-sided $p < 0.05$, $p < 0.01$ and $p < 0.001$, respectively.

| Model | Multi-turn mean | Single-turn mean | Diff (M-S) | Aligned with expectation? | $p$-value (one-sided) | Significance |
|---|---|---|---|---|---|---|
| DeepSeek-V3 | 0.19 | 0.24 | -0.05 | Yes | 0.00 | *** |
| Qwen-2.5-72B | 0.20 | 0.24 | -0.04 | Yes | 0.00 | *** |
| Llama-3.3-70B | 0.21 | 0.24 | -0.03 | Yes | 0.00 | *** |
| Qwen-2.5-7B | 0.17 | 0.20 | -0.02 | Yes | 0.00 | *** |
| DeepSeek-R1 | 0.19 | 0.22 | -0.02 | Yes | 0.00 | *** |
| QwQ-32B | 0.19 | 0.21 | -0.02 | Yes | 0.00 | ** |
| GPT-4o-mini | 0.26 | 0.27 | -0.01 | Yes | 0.02 | * |
| Qwen-2.5-14B | 0.22 | 0.23 | -0.01 | Yes | 0.03 | * |
| Gemini-2.5-Pro | 0.22 | 0.23 | -0.01 | Yes | 0.03 | * |
| GPT-4o | 0.28 | 0.29 | -0.01 | Yes | 0.04 | * |
| Qwen-2.5-32B | 0.22 | 0.23 | -0.01 | Yes | 0.06 | |
| Llama-3.1-8B | 0.19 | 0.18 | 0.01 | No | 0.89 | |

states that there is no systematic difference between single-turn and multi-turn interactions, whereas the alternative hypothesis ($H_1$) specifies that the median paired difference follows a given direction: Multi-turn $<$ Single-turn for $S_f$, and Multi-turn $>$ Single-turn for $S_h$, $D_f$, and $D_h$. From an efficiency perspective, larger $D_f$ means that the model spends more tokens per *correct* fact (worse efficiency), whereas larger $D_h$ means that each *hallucinated* fact is more token-expensive and thus sparser per token, which is preferable from a safety/robustness standpoint. Table 14 summarizes the alignment of each model with these metric-specific hypotheses. At a high level, degradation in $S_f$ and $D_f$ is statistically significant for almost all models; for hallucination, we observe a consistent tendency for $S_h$ to increase and for $D_h$ to increase as well (hallucinations becoming sparser), although the strength of statistical evidence differs across metrics, as detailed below.

**Factuality ($S_f$)** The detailed results in Table 15 confirm a statistically significant decline in factual coverage. Among the 12 models, 11 show lower $S_f$ in the multi-turn setting, and this drop is significant at one-sided $p < 0.05$ for 10 of them. Models such as DeepSeek-V3, Qwen-2.5-72B, and Llama-3.3-70B exhibit highly significant declines ($p < 0.001$), reinforcing the conclusion that maintaining comprehensive factual recall becomes increasingly difficult as conversation depth grows.

**Hallucination ($S_h$): Zero-Inflation and Directional Consistency** The analysis of $S_h$ (Table 16) requires a more nuanced interpretation. Eleven out of twelve models exhibit changes that are aligned with the expected degradation direction (Multi-turn $>$ Single-turn) or remain unchanged, but only three models reach one-sided $p < 0.05$. This pattern is primarily driven by the sparsity and zero-inflation of hallucination events: many instances satisfy $S_h = 0$, so the majority of paired differences satisfy $S_h^{\text{multi}} - S_h^{\text{single}} = 0$. In non-parametric rank tests like Wilcoxon, such ties substantially reduce the effective sample size and thus statistical power. Importantly, among the non-tied pairs

Table 16: Wilcoxon signed-rank test results for $S_h$ (one-sided). The expected direction is Multi-turn > Single-turn. The column "Aligned with expectation?" indicates whether the mean difference between Multi-turn and Single-turn follows this expected direction. The "Significance" column uses *, ** and *** to denote one-sided $p < 0.05$, $p < 0.01$ and $p < 0.001$, respectively.

| Model | Multi-turn mean | Single-turn mean | Diff (M-S) | Aligned with expectation? | $p$-value (one-sided) | Significance |
|---|---|---|---|---|---|---|
| Gemini-2.5-Pro | 0.04 | 0.03 | 0.01 | Yes | 0.00 | *** |
| QwQ-32B | 0.04 | 0.03 | 0.01 | Yes | 0.00 | ** |
| Llama-3.3-70B | 0.04 | 0.04 | 0.01 | Yes | 0.02 | * |
| GPT-4o-mini | 0.03 | 0.02 | 0.00 | Yes | 0.09 | |
| Qwen-2.5-32B | 0.03 | 0.03 | 0.00 | Yes | 0.11 | |
| DeepSeek-R1 | 0.04 | 0.04 | 0.00 | Yes | 0.11 | |
| Qwen-2.5-72B | 0.03 | 0.02 | 0.00 | Yes | 0.25 | |
| DeepSeek-V3 | 0.03 | 0.03 | 0.00 | Yes | 0.26 | |
| GPT-4o | 0.02 | 0.02 | 0.00 | Yes | 0.27 | |
| Qwen-2.5-7B | 0.04 | 0.03 | 0.00 | Yes | 0.39 | |
| Llama-3.1-8B | 0.04 | 0.04 | 0.00 | Yes | 0.41 | |
| Qwen-2.5-14B | 0.03 | 0.03 | -0.00 | No | 0.75 | |

Table 17: Wilcoxon signed-rank test results for $D_f$ (one-sided). The expected direction is Multi-turn > Single-turn, corresponding to a degradation in efficiency for correct facts (more tokens per correct fact). The column "Aligned with expectation?" indicates whether the mean difference between Multi-turn and Single-turn follows this expected direction. The "Significance" column uses *, ** and *** to denote one-sided $p < 0.05$, $p < 0.01$ and $p < 0.001$, respectively.

| Model | Multi-turn mean | Single-turn mean | Diff (M-S) | Aligned with expectation? | $p$-value (one-sided) | Significance |
|---|---|---|---|---|---|---|
| Qwen-2.5-7B | 545.91 | 366.78 | 179.13 | Yes | 0.00 | *** |
| Qwen-2.5-72B | 486.41 | 357.49 | 128.92 | Yes | 0.00 | *** |
| GPT-4o-mini | 343.10 | 255.10 | 88.00 | Yes | 0.00 | *** |
| Qwen-2.5-14B | 426.54 | 295.47 | 131.07 | Yes | 0.00 | *** |
| DeepSeek-V3 | 443.71 | 340.33 | 103.38 | Yes | 0.00 | *** |
| Llama-3.3-70B | 439.03 | 323.99 | 115.04 | Yes | 0.00 | *** |
| Qwen-2.5-32B | 433.50 | 365.82 | 67.68 | Yes | 0.00 | *** |
| Llama-3.1-8B | 475.33 | 403.98 | 71.35 | Yes | 0.00 | *** |
| GPT-4o | 255.83 | 226.63 | 29.20 | Yes | 0.00 | *** |
| DeepSeek-R1 | 628.00 | 601.90 | 26.10 | Yes | 0.00 | ** |
| Gemini-2.5-Pro | 694.77 | 674.93 | 19.84 | Yes | 1.00 | |
| QwQ-32B | 468.97 | 527.21 | -58.23 | No | 1.00 | |

where a change does occur, the shift is consistently towards higher hallucination. Consequently, the large one-sided $p$-values mainly reflect a floor effect caused by rare hallucination events, rather than evidence against a degradation trend.

**Information Delivery Efficiency ($D_f$) and Hallucination Sparsity ($D_h$)** The clearest degradation appears in $D_f$ (Table 17): 11 of the 12 models require more tokens per correct fact in the multi-turn setting, and this increase is statistically significant for 10 models ($p < 0.05$). This confirms that, as conversations become multi-turn, models systematically become less efficient in how they allocate tokens to correct factual content. In contrast, the pattern for $D_h$ (Table 18) reflects an improvement in hallucination sparsity. For 9 models, the token cost per contradicted fact is significantly larger in the multi-turn setting (one-sided $p < 0.05$), meaning that hallucinated facts become more token-expensive and thus sparser per token. In other words, when hallucinations do occur, they tend to be more diluted within longer responses.

## H.5 VALIDATION OF THE MAXIMUM GENERATION LENGTH

Our evaluation includes models specifically designed for long Chain-of-Thought (CoT) reasoning, such as QwQ-32B, Gemini-2.5-pro and DeepSeek-R1. For these models, a methodological concern is that their inherently verbose reasoning might consume a disproportionate share of the 1024-token generation limit, leaving insufficient space for the final answer. To investigate this, we conduct a validation experiment in both single-turn and multi-turn settings.

Table 18: Wilcoxon signed-rank test results for $\mathbf{D_h}$ (one-sided). The expected direction is Multi-turn > Single-turn, which corresponds to a larger token cost per contradicted fact and hence sparser hallucinations per token (an improvement from a robustness perspective). The column "Aligned with expectation?" indicates whether the mean difference between Multi-turn and Single-turn follows this expected direction. The "Significance" column uses *, ** and *** to denote one-sided $p < 0.05$, $p < 0.01$ and $p < 0.001$, respectively.

| Model | Multi-turn mean | Single-turn mean | Diff (M-S) | Aligned with expectation? | $p$-value (one-sided) | Significance |
|---|---|---|---|---|---|---|
| Qwen-2.5-14B | 825.72 | 512.69 | 313.03 | Yes | 0.00 | *** |
| GPT-4o-mini | 783.58 | 545.69 | 237.89 | Yes | 0.00 | *** |
| Qwen-2.5-7B | 846.97 | 590.12 | 256.85 | Yes | 0.00 | *** |
| DeepSeek-V3 | 880.05 | 738.83 | 141.22 | Yes | 0.00 | *** |
| Llama-3.1-8B | 751.98 | 588.04 | 163.93 | Yes | 0.00 | *** |
| GPT-4o | 553.90 | 506.10 | 47.80 | Yes | 0.00 | *** |
| Qwen-2.5-72B | 902.65 | 822.77 | 79.88 | Yes | 0.00 | *** |
| Qwen-2.5-32B | 744.19 | 683.73 | 60.46 | Yes | 0.00 | *** |
| Llama-3.3-70B | 758.66 | 655.91 | 102.75 | Yes | 0.00 | *** |
| DeepSeek-R1 | 1176.21 | 1231.31 | -55.10 | No | 1.00 | |
| Gemini-2.5-Pro | 1414.47 | 1620.22 | -205.75 | No | 1.00 | |
| QwQ-32B | 815.24 | 918.85 | -103.61 | No | 1.00 | |

We compare the standard 1024-token limit QwQ-32B against QwQ-32B-2048, a variant with an 2048-token limit. The results, presented in Table 19, demonstrate a consistent trend across both settings: the extended generation capacity fails to provide a clear advantage. In both single-turn and multi-turn scenarios, increasing

Table 19: Performance comparison of QwQ-32B and QwQ-32B-2048. White and gray rows indicate the single-turn and multi-turn setting, respectively.

| Setting | $\mathbf{S_f}$ (%) ↑ | $\mathbf{S_h}$ (%) ↓ | $\mathbf{D_f}$ ↓ | $\mathbf{D_h}$ ↑ |
|---|---|---|---|---|
| QwQ-32B | 20.96 | 2.73 | 531 | 919 |
| QwQ-32B-2048 | 20.06 | 3.84 | 703 | 1556 |
| QwQ-32B | 19.17 | 3.92 | 473 | 815 |
| QwQ-32B-2048 | 17.26 | 3.91 | 806 | 1538 |

the token limit led to a degradation in the $\mathbf{S_f}$ and offered no substantive improvement in $\mathbf{S_h}$. Furthermore, the token efficiency per correct fact ($\mathbf{D_f}$) consistently worsened with the larger budget. This consistent pattern strongly suggests that the model's performance is not primarily constrained by its reasoning crowding out the answer space. The analysis thus validates our use of 1024 tokens as a sufficient and robust setting for the main experiments, regardless of the conversational context.

### H.6 COMPARISON TO TRADITIONAL N-GRAM-BASED METRICS

To provide further context for our proposed metrics, we analyze the relationship between $\mathbf{S_f}$ and two prevalent n-gram-based metrics: ROUGE-L and BLEU-4. The scatter plots in Figure 15 map the performance of evaluated models across these metrics. The numerical results are listed in Table 13

The analysis reveals a positive correlation between $\mathbf{S_f}$ and both ROUGE-L and BLEU-4. This alignment is expected, as higher factual accuracy often coincides with greater lexical overlap with reference texts. This finding suggests that our metric is directionally consistent with established evaluation paradigms.

A closer inspection of the plots, however, reveals a systematic deviation. We find that a cluster of models, particularly those optimized for CoT reasoning, are consistently undervalued by ROUGE-L and BLEU-4 relative to their $\mathbf{S_f}$ scores. Notably, this occurs even after programmatically removing the CoT reasoning steps, with all metrics assessing only the final answer. We hypothesize this discrepancy stems not from the reasoning text itself, but from subtle stylistic artifacts in the final synthesized answer. It is plausible that the CoT generation process implicitly influences the model's final output style, leading to differences in sentence structure or lexical choice compared to the reference. While these stylistic variations may not compromise the underlying factswhich our metric is designed to capture by operating on decomposed statementsthey can penalize scores for metrics sensitive to surface-level matching. This observation highlights the value of evaluation frameworks that can disentangle factual correctness from surface-level stylistic choices.

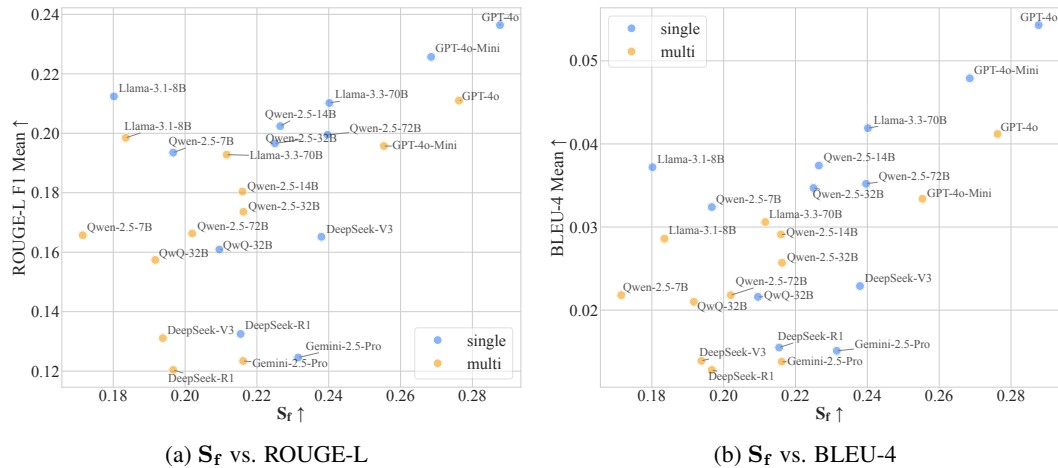

(a) $\mathbf{S_f}$ vs. ROUGE-L

(b) $\mathbf{S_f}$ vs. BLEU-4

Figure 15: Correlation between $\mathbf{S_f}$ and traditional lexical metrics. While a positive trend is observed, models optimized for CoT reasoning tend to be undervalued by n-gram metrics, potentially due to subtle stylistic differences in their final answers.

### H.7 INTERPRETABILITY ANALYSIS

To analyze how models use dialogue context in the final response, we attribute token-level importance by coupling attention weights with gradients computed only with respect to the last-turn answer. For a dialogue $d_k$, let the input token sequence for the final turn be $X^{(k)} = (x_1, \ldots, x_L)$, obtained by concatenating the history $H_k$, the final-turn question $q_{N_k}^{(k)}$, and the model's last-turn answer $a_{N_k}^{(k)}$. We denote the index sets of context, question, and answer tokens by $\mathcal{T}_{\text{ctx}}$, $\mathcal{T}_{\text{q}}$, and $\mathcal{T}_{\text{ans}}$, respectively, forming a partition of $\{1, \ldots, L\}$.

Let $\theta$ be model parameters. We define the target for attribution as the (token-averaged) log-likelihood of the last-turn answer:

$$s^{(k)} \;=\; \sum_{t \in \mathcal{T}_{\text{ans}}} \log p_\theta\big(x_t \mid x_{<t},\, H_k,\, q_{N_k}^{(k)}\big). \tag{5}$$

All gradients below are taken with respect to this $s^{(k)}$, so they reflect how changing attention would affect the probability of the *last-turn answer only*.

For each transformer layer $l \in \{1, \ldots, L_\ell\}$ and head $h \in \{1, \ldots, H\}$, let $A^{(l,h)} \in \mathbb{R}^{L \times L}$ be the row-stochastic attention matrix from the final forward pass (rows: query positions; columns: key positions). We compute its gradient

$$G^{(l,h)} \;=\; \frac{\partial s^{(k)}}{\partial A^{(l,h)}} \;\in\; \mathbb{R}^{L \times L}. \tag{6}$$

We then attribute a *source-side* importance to each token $j$ by aggregating, over layers, heads, and answer query positions $t \in \mathcal{T}_{\text{ans}}$, the signed gradient-weighted attention received by $j$:

$$I_j \;=\; \frac{1}{Z} \sum_{l=1}^{L_\ell} \sum_{h=1}^{H} \sum_{t \in \mathcal{T}_{\text{ans}}} A_{tj}^{(l,h)} \,\text{sign}\big(G_{tj}^{(l,h)}\big), \qquad Z \;=\; L_\ell \cdot H \cdot |\mathcal{T}_{\text{ans}}|. \tag{7}$$

A positive $I_j$ indicates that attending to token $j$ increases the likelihood of the last-turn answer, while a negative value indicates an inhibitory effect. (As a magnitude-sensitive variant, one may replace $\text{sign}(\cdot)$ with $\tanh(\alpha|G_{tj}^{(l,h)}|)$ or simply $|G_{tj}^{(l,h)}|$; we use the signed version in Eq. equation 7.)

In our final reporting, we aggregate only over the answer tokens that realize the useful atomic statement in the last-turn answer. From $a_{N_k}^{(k)}$ we extract a set of useful atomic statements $\mathcal{S}_k^+$ and use `Qwen2.5-7B-Instruct` to align each $s \in \mathcal{S}_k^+$ back to its minimal supporting span(s) in $a_{N_k}^{(k)}$, yielding

$$\mathcal{T}_{\text{ans}}^+ \;\subseteq\; \mathcal{T}_{\text{ans}}. \tag{8}$$

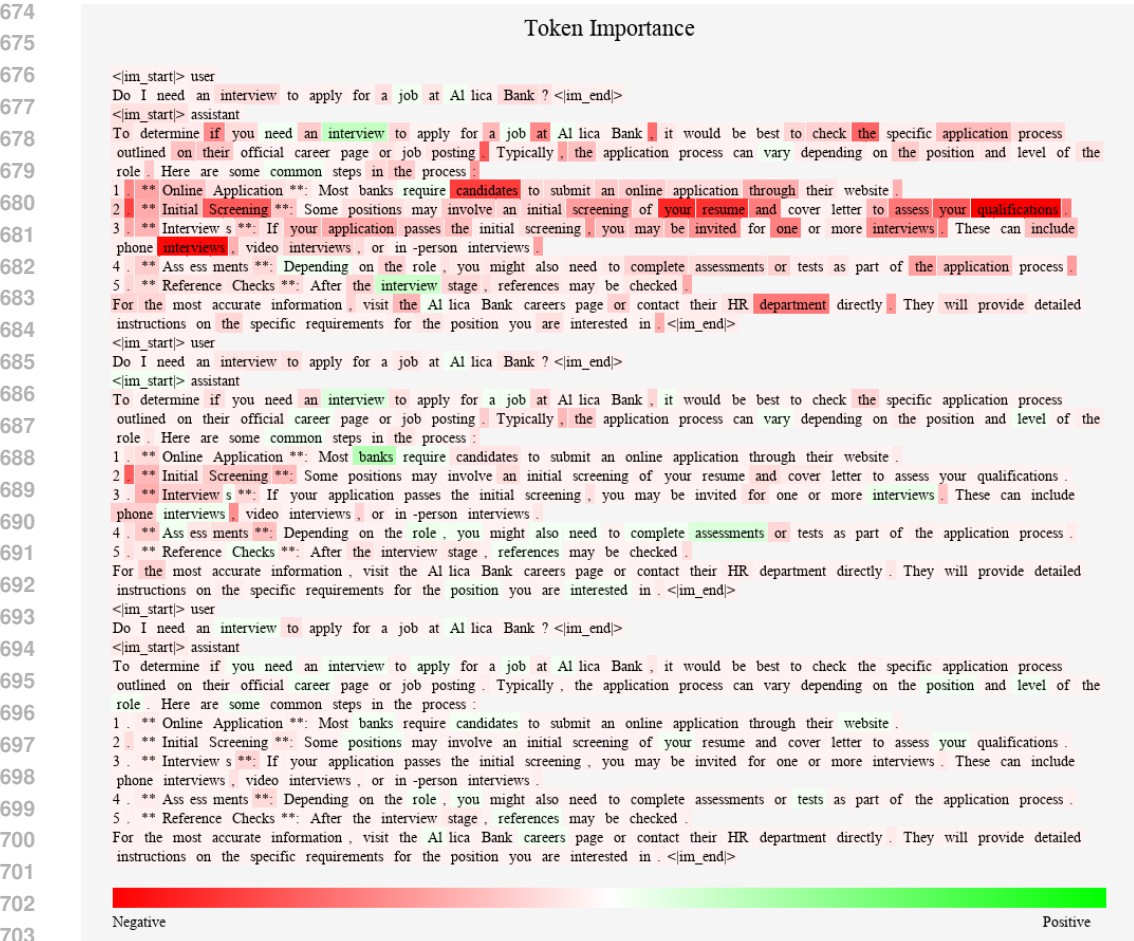

Figure 16: Visualization of contextual token importance. Green indicates tokens that positively contribute to the model's response, while red indicates tokens with a negative influence.

All gradients are still taken with respect to $s^{(k)}$ (Eq. 5), but the aggregation over query positions $t$ is restricted to $t \in \mathcal{T}_{\mathrm{ans}}^{+}$. We define the useful-span source-side importance as

$$I_j^{(+)} \;=\; \frac{1}{Z^{(+)}} \sum_{l=1}^{L_\ell} \sum_{h=1}^{H} \sum_{t \in \mathcal{T}_{\mathrm{ans}}^{+}} A_{tj}^{(l,h)} \, \mathrm{sign}\big(G_{tj}^{(l,h)}\big), \tag{9}$$

with the normalization

$$Z^{(+)} \;=\; L_\ell \cdot H \cdot \big|\mathcal{T}_{\mathrm{ans}}^{+}\big|. \tag{10}$$

We quantify the net contribution of the dialogue context to producing the useful statement(s) via

$$\overline{I}_{\mathrm{ctx}}^{(+)} \;=\; \frac{1}{|\mathcal{T}_{\mathrm{ctx}}|} \sum_{j \in \mathcal{T}_{\mathrm{ctx}}} I_j^{(+)}, \tag{11}$$

and analogously define $\overline{I}_{\mathrm{q}}^{(+)}$ and $\overline{I}_{\mathrm{ans}}^{(+)}$ if needed. Empirically, under this useful-span restriction, the contextual averages for **Qwen2.5-7B-Instruct** and **LLaMA3.1-8B-Instruct** are $-0.0025$ and $-0.0029$, indicating that context tokens exert a net negative (noise-like) influence on the helpful parts of the final answer. See Figure 16 for a visualization.

## H.8 ROBUSTNESS TO TOPIC DRIFT

While KNOWMT-BENCH primarily targets domain-focused, progressive expert consultations, real-world interactions sometimes involve spontaneous topic switching or "drift." To probe the impact

of such disjointed contexts and ensure our evaluation does not overestimate model robustness, we conducted a controlled experiment simulating topic drift.

**Experimental Setup.** We prepended a logically unrelated question (randomly sampled from a disjoint domain) before each question sequence in our standard evaluation set. This setup introduces irrelevant contextual noise, forcing the model to discern the current intent amidst a drifted history. We evaluate Qwen-2.5-7B under this setting compared to the original baseline.

**Results and Analysis.** The results are presented in Table 20. First, in the single-turn setting, performance remains essentially invariant, suggesting that unrelated prefixes have a negligible impact on the model's immediate instruction following. Second, in the multi-turn scenario, we observe a nuanced trade-off: topic drift leads to a simultaneous increase in both Factual Score ($S_f$: $17.15\% \rightarrow 20.51\%$) and Hallucination Score ($S_h$: $3.77\% \rightarrow 4.88\%$). This indicates that irrelevant context disrupts the model's generation pattern, potentially breaking the "conservative" constraint observed in standard multi-turn dialogues, thereby yielding more facts but at the cost of higher hallucination risk.

Crucially, these findings align with our core analysis in Section 4.3, confirming that the detrimental impact of *relevant misinformation* outweighs that of irrelevant noise or errors. Consequently, the exclusion of spontaneous topic drift does not substantially compromise the validity of our evaluation in knowledge-intensive scenarios, where robustness against relevant hallucinations is the primary concern.

Table 20: Impact of Topic Drift on Qwen-2.5-7B. We compare the standard setting against a "Topic Drift" setting where an unrelated question is prepended to the context.

| Setting | Mode | $S_f$ (%) | $S_h$ (%) | $D_f$ | $D_h$ |
|---|---|---|---|---|---|
| Original | Single-turn | 19.67 | 3.57 | 366 | 589 |
| Topic Drift | Single-turn | 19.65 | 3.49 | 367 | 590 |
| Original | Multi-turn | 17.15 | 3.77 | 547 | 847 |
| Topic Drift | Multi-turn | 20.51 | 4.88 | 537 | 833 |

## I  TOPIC TAXONOMY

We manually classified these topics into three categories: *institutions and products*, *policies and events*, and *concepts*. Below we formalize the category descriptions (kept from the original, translated to English) and provide filled examples in a unified **T/Q/A/R** format.

### I.1  ORGANIZATIONS & THEIR PRODUCTS/TOOLS

**Category Descriptions** This category covers various types of organizations and their products or services. For *insurance institutions*, the focus is on product details such as claim standards, premium/payment flows, and required documents, as well as the scope of the institution from official "About us" pages. For *banks*, examples include the materials required to handle services, the types of deposits, deposit safety in case of failure, and application conditions for specific credit cards. For *government departments*, the emphasis lies in service-handling procedures and official responsibilities. *Financial institutions (non-bank)* are described through their products and company details, while *other companies* are captured by their main businesses and notable events.

FILLED EXAMPLES  ORGANIZATIONS & THEIR PRODUCTS/TOOLS

**Insurance Institutions**
**T:** AXA
**Q:** What are the main businesses of AXA?
**A:** AXA's main businesses include property-casualty insurance business, life & savings business

and asset management business.

**R:** https://www.axa.com/en/about-us/what-we-do

**T:** Allstate

**Q:** What information do I need to provide to purchase auto insurance with Allstate?

**A:** When purchasing auto insurance with Allstate, you will need to provide the following information: 1. Vehicle Information: the make, model, and vehicle identification number for each car you want to insure. 2. Covered Drivers Information: the names of all drivers covered by the policy. 3. Current Insurance Information: your existing coverage and limits. 4. Personal Information: dates of birth, Social Security numbers, and driver's license numbers for each family member to be included in the policy. 5. Driving History: an overview of any past tickets or auto accidents incurred by family members you want to cover on the policy. 6. Background Information: whether you own or rent your home, and some details about your assets and income. 7. Coverage Types and Limits: the types of coverage you want.

**R:** https://www.allstate.com/resources/information-needed-for-insurance-quote

**Banks**

**T:** Bank of America

**Q:** What are the cash rewards like on this credit card, Bank of America Customized Cash Rewards?

**A:** The Bank of America Customized Cash Rewards credit card offers cashback as follows: 1. 1% base cashback on all eligible net purchases, meaning you earn $0.01 for every $1 spent. 2. 2% cashback on eligible purchases at grocery stores, wholesale clubs, supermarkets, meat and seafood markets, candy, nut, and confectionery stores, dairy stores, and bakeries, which includes the 1% base cashback plus an additional 1% bonus, but excludes supermarkets and small stores that sell groceries like drugstores. 3. 3% cashback on purchases in one chosen "Choice" category, which includes gas and EV charging stations (default), online shopping, dining, travel, drug stores, or home improvement/furnishings, with the 3% consisting of the 1% base cashback plus an additional 2% bonus. 4. Transactions such as balance transfers, cash advances, fees, interest charges, fraudulent transactions, and certain other fees do not earn cashback.

**R:** https://www.bankofamerica.com/credit-cards/terms-and-conditions/?campaignid=4071205&productoffercode=UN&locale=en_US

**T:** NatWest Bank

**Q:** What types of personal savings accounts are available at NatWest Bank?

**A:** NatWest Bank's Individual Savings Accounts include the Digital Regular Saver, Digital Regular Saver, Fixed Rate ISA, Fixed Term Savings, Flexible Saver, Stocks & Shares ISA and First Saver.

**R:** https://www.natwest.com/savings.html?intcam=HP-TTB-DEF-Default#productFilter

**T:** Silicon Valley Bank

**Q:** Is my deposit safe if Silicon Valley Bank fails?

**A:** If your deposits are with Silicon Valley Bank and meet the requirements of the Federal Deposit Insurance Corporation (FDIC), your deposits are safe. The FDIC provides insurance coverage of up to $250,000 per depositor, including principal and interest. If your deposits exceed this amount, additional coverage may apply based on different account ownership categories. You can use the FDIC's Electronic Deposit Insurance Estimator (EDIE) to verify if your deposits are fully covered. Please note, FDIC insurance does not cover investments such as stocks, bonds, or mutual funds.

**R:** https://www.svb.com/fdic/

**T:** Wells Fargo

**Q:** When I can't earn Rewards Points Bonus on my Wells Fargo travel rewards credit card

**A:** You will not earn Rewards Points on the following types of transactions with your travel rewards credit card: 1. Cash Advances and Equivalents: This includes ATM transactions, cash advances, money orders, prepaid gift cards, traveler's checks, wire transfers, and balance transfers. 2. Disputed or Illegal Transactions: Any purchases that are disputed, illegal, or violate the terms of the Credit Card Account agreement. 3. Fees and Interest: Any fees or interest charges that post to your Credit Card Account, such as annual fees, monthly fees, late fees, and returned payment fees. 4. Gambling Transactions: This includes any gambling-related transactions, such as online bets or wagers, casino gaming chips, lottery tickets, and off-track wagers.

**R:** https://www.wellsfargo.com/credit-cards/autograph-journey-visa/terms/

**Government Departments**

**T:** IRS

**Q:** What is the mission of the IRS?

**A:** The IRS mission is to provide America's taxpayers top quality service by helping them understand and meet their tax responsibilities and to enforce the law with integrity and fairness to all.

**R:** https://www.irs.gov/about-irs

**T:** DWP

**Q:** I am dissatisfied with my service at DWP, how do I make a complaint?

**A:** 1. If youd like to complain about any aspect of the service youve received, let the office you have been dealing with know as soon as possible. You can contact them by phone, in person or in writing. Universal Credit claimants can also use their journal. 2. You need to provide the necessary details, including your National Insurance number (unless you are an employer), your full name, address and contact details, the benefit you are complaining about, what happened, when it happened, and how it affected you, and what you want to happen to resolve the issue. 3. You can use the contact details on any recent letters weve sent you or use the contact information below. If you live in Northern Ireland, visit the Department for Communities website for more information.

**R:** https://www.gov.uk/government/organisations/department-for-work-pensions/about/complaints-procedure#contact-the-office-you've-been-dealing%20with

**Financial Institutions (non-bank)**

**T:** S&P 500

**Q:** What are the key components of Fidelity's iShares Core S&P 500 ETF?

**A:** The fund typically invests at least 80% of its assets in the component securities of the S&P 500 index or in investments that have economic characteristics substantially identical to those component securities. The remaining 20% of its assets may be invested in certain futures, options, swap contracts, cash, and cash equivalents.

**R:** https://digital.fidelity.com/prgw/digital/research/quote/dashboard/summary?symbol=IVV

**Other Companies**

**T:** TMTG

**Q:** What is the main business of Trump Media & Technology Group Corp.

**A:** TMTG's main businesses include Truth Social, a social media platform established as a safe harbor for free expression amid increasingly harsh censorship by Big Tech corporations, as well as Truth+, a TV streaming platform focusing on family-friendly live TV channels and on-demand content. TMTG is also launching Truth.Fi, a financial services and FinTech brand incorporating America First investment vehicles.

**R:** https://s3.amazonaws.com/sec.irpass.cc/2660/0001140361-25-004822.html

I.2    POLICIES, LAWS, OR EVENTS

**Category Description**    What is it? What does it include? Key details and clauses to note.

FILLED EXAMPLES  POLICIES, LAWS, OR EVENTS

**T:** Bitcoin legal

**Q:** What was the first country to make Bitcoin legal tender?

**A:** El Salvador is the first country in the world to make Bitcoin legal tender. On June 9, 2021, El Salvador's Congress passed the Bitcoin Law, making Bitcoin legal tender alongside the U.S. dollar, which went into effect on September 7 of the same year.

**R:**          https://legaljournal.princeton.edu/el-salvadors-bitcoin-law-contemporary-implications-of-forced-tender-legislation/

**T:** PPP Loan Forgiveness

**Q:** How can I qualify for the Full Forgiveness Terms of the First Draw PPP Loans based on the information provided?

**A:** To qualify for the Full Forgiveness Terms of the First Draw PPP Loans, the following conditions must be met: 1. Employee and compensation levels are maintained: During the 8– to 24–week period following loan disbursement, the business must maintain the same number of employees and compensation levels. 2. Loan proceeds are spent on eligible expenses: The loan funds must be spent

on payroll costs and other eligible expenses, such as rent, interest, and utilities. 3. At least 60% of the proceeds are spent on payroll costs: At least 60% of the loan amount must be used for payroll costs. By meeting these conditions, the loan may be fully forgiven.

**R:** https://home.treasury.gov/system/files/136/Top-line-Overview-of-First-Draw-PPP.pdf

## I.3 CONCEPTS

**Category Description**
**Currency**

- Exchange-rate history.
- Anchor/peg history and regimes.

**Other Concepts**

- What (definition)
- Why (motivation, use cases)
- Features (key characteristics)
- Concept explanation (intuition + precise definition)

FILLED EXAMPLES CONCEPTS

**T:** British pound
**Q:** What is the anchor of the British pound?
**A:** The pound's anchor has changed many times throughout history and can be divided into three main stages. The first stage was the gold standard, where the value of the pound was directly linked to gold reserves, ensuring its stability due to gold's intrinsic value. The second stage was the Bretton Woods system, established after World War II, where the pound was pegged to the US dollar, and the dollar itself was pegged to gold. The third stage began with the collapse of the Bretton Woods system, and since then, the value of the pound has been based on the economic strength and creditworthiness of the UK rather than any physical commodity like gold or silver.
**R:** https://www.britannica.com/money/money/The-decline-of-gold#ref1089594, https://www.britannica.com/story/how-are-currency-exchange-rates-determined

**T:** interest rate
**Q:** What do the nominal interest rate and real interest rate refer to in the Fisher Effect?
**A:** In the Fisher Effect, the nominal interest rate refers to the interest rate that does not account for inflation, which is the stated rate provided by financial institutions, reflecting how the amount of deposits or loans grows over time. The real interest rate, on the other hand, is the interest rate that takes inflation into account, indicating how the purchasing power of deposits or loans changes over time. The relationship between the two can be expressed by the Fisher equation, which states that the nominal interest rate is approximately equal to the real interest rate plus the expected inflation rate.
**R:** https://www.investopedia.com/terms/f/fishereffect.asp

## J PROMPTS

### J.1 BUILD QUESTIONS LIST PROMPTS

> **Seed Question Expansion Prompt (2-Turn)**
>
> ```
> Task:
> You will be given an original question (QO).
> Please choose an random appropriate template from the 8 predefined types below, and
>     extend the original question to two-turn questions (Q1 and Q2), where:
> - Q1 is a new question that introduces a different but related aspect of the topic
> - Q2 is completely identical to QO
> - Questions can be relatively natural and colloquial
> ```

```
Q1 and Q2 must seek completely non-overlapping information - they should cover distinct
    aspects of the topic without any duplication.

Available Template Types:

1. Existence + Details
 Q1: Does X have Y?
 Q2: Specific details about Y within X.

2. Definition/Concept + Details
 Q1: What is X or Y?
 Q2: A follow-up question about X or Y.

3. Introductory Question + Details
 Q1: A background question that introduces X.(not include X)
 Q2: A detailed question about X.

4. Comparison + Focus on One
 Q1: A comparison involving X.
 Q2: A deeper question about X specifically.

5. Different Angles on the Same Topic
 Q1: One common question about X.
 Q2: Another question about X from a different perspective.

6. Cause + Details
 Q1: Why did X happen?
 Q2: A detailed question about X.

7. Evolution + Current State
 Q1: How did X develop or evolve?
 Q2: What is the current status of X?

8. Conditional Trigger + Consequence
 Q1: What happens if X occurs?
 Q2: In that case, how would X affect Y?

Examples:

Input:
QO: What are the main businesses of AXA?

Action:
Choose template type: 2. Definition/Concept + Details

Output:
Q1: What kind of company is AXA?
Q2: What are the main businesses of AXA?

---

Input:
QO: How did the 2008 financial crisis affect AIG?

Action:
Choose template type: 6. Cause + Details

Output:
Q1: Why did the 2008 financial crisis happen?
Q2: How did the 2008 financial crisis affect AIG?

---

Input:
QO: How would a recession impact small businesses?

Action:
Choose template type: 8. Conditional Trigger + Consequence

Output:
Q1: What happens to the economy during a recession?
Q2: How would a recession impact small businesses?

---

Input:
QO:  What is the mission of the IRS?
```

```
Action:
Choose template type: 3. Introductory Question + Details

Output:
Q1: What is the U.S. federal government's tax agency?
Q2: What is the mission of the IRS?

---

Input:
QO: What information do I need to provide to purchase auto insurance with Allstate?

Action:
Choose template type: 1. Existence + Details

Output:
Q1: Does Allstate have auto insurance?
Q2: What information do I need to provide to purchase auto insurance with Allstate?

---

Input:
QO: What are the key components of Fidelity's iShares Core S&P 500 ETF?

Action:
Choose template type: 4. Comparison + Focus on One

Output:
Q1: Of Fidelity's ETF products, is the iShares Core S&P 500 ETF less risky or the ETP
    Fidelity Wise Origin Bitcoin Fund less risky.
Q2: What are the key components of Fidelity's iShares Core S&P 500 ETF?

---

Input:
QO: What is the current status of ChatGPT technology?

Action:
Choose template type: 7. Evolution + Current State

Output:
Q1: How has ChatGPT evolved since its initial release?
Q2: What is the current status of ChatGPT technology?

---

Input:
QO: How did the 2008 financial crisis affect AIG?

Action:
Choose template type: 6. Cause + Specific Impact

Incorrect Output:
Q1: Why did the 2008 financial crisis happen?
Q2: How did the 2008 financial crisis affect AIG?

---

Now, based on the input QO, choose the most suitable template and generate
    corresponding Q1 and Q2 that seek completely non-overlapping information, where Q2
     is completely identical to QO.

---

Input:
QO: {{QO_PLACEHOLDER}}
```

## Seed Question Expansion Prompt (3-Turn)

```
Task:
You will be given an original question (QO).
Please choose a random appropriate template from the 10 predefined types below, and
    extend the original question into a three-turn sequence (Q1, Q2, Q3), where:

- Q1 is a new question that introduces a different but related aspect of the topic
```

```
- Q2 is a follow-up that brings the conversation closer to the original question
- Q3 is completely identical to Q0
- Questions can be relatively natural and colloquial
- Q1, Q2, and Q3 must seek completely non-overlapping information  they should cover
    distinct aspects of the topic without duplication

Available Template Types:

1. Lead-in Question + Lead-in Question + Specific Detail
 Q1: A question that leads to Y
 Q2: A question that leads from Y to X
 Q3: A detailed question about X

2. Evolution + Current State + Specific Detail
 Q1: How did X develop or evolve?
 Q2: What is the current status or key traits of X?
 Q3: A specific question about X

3. Different Angles on the Same Topic
 Q1: One question about X
 Q2: Another question about X
 Q3: A third question about X

4. Definition + Existence + Specific Detail
 Q1: What is X?
 Q2: Does X have Y?
 Q3: Specific details about Y in X

5. Lead-in + Existence + Specific Detail
 Q1: A question that introduces X
 Q2: Does X have Y?
 Q3: Specific details about Y in X

6. Definition + Lead-in + Specific Detail
 Q1: A question about Y
 Q2: A related question that introduces X
 Q3: Specific details about X

7. Definition + Cause + Specific Detail
 Q1: What is X?
 Q2: Why did X happen?
 Q3: A specific consequence or detail about X

8. Different Angles + Existence + Specific Detail
 Q1: A general question about X
 Q2: Does X have a specific attribute Y?
 Q3: Detailed information about Y in X

9. Lead-in + Comparison + Specific Detail
 Q1: A question that introduces X
 Q2: Comparison between X and Y
 Q3: Specific detail about X

10. Definition + Definition + Specific Detail
 Q1: What is X?
 Q2: What is Y?
 Q3: A question about how Y functions in X

Examples:

Input:
Q0: What is the mission of the IRS?

Action:
Choose template type: 1. Lead-in + Lead-in + Specific Detail

Output:
Q1: How does the U.S. government fund its public services and programs?
Q2: What is the U.S. federal government's tax agency?
Q3: What is the mission of the IRS?

---

Input:
Q0: What does ChatGPT charge?

Action:
Choose template type: 2. Evolution + Current State + Specific Detail
```

```
Output:
Q1: How has ChatGPT evolved since its initial release?
Q2: What is the current status of ChatGPT technology?
Q3: What does ChatGPT charge?

---

Input:
QO: Will taking allopurinol affect my fertility?

Action:
Choose template type: 3. Different Angles on the Same Topic

Output:
Q1: What is allopurinol used for?
Q2: Will taking allopurinol help me prevent gout?
Q3: Will taking allopurinol affect my fertility?

---

Input:
QO: What information do I need to provide to purchase auto insurance with Allstate?

Action:
Choose template type: 4. Definition + Existence + Specific Detail

Output:
Q1: Can you introduce me to Allstate?
Q2: Does Allstate have auto insurance?
Q3: What information do I need to provide to purchase auto insurance with Allstate?

---

Input:
QO: What information do I need to provide to open a bank account at Chase?

Action:
Choose template type: 5. Lead-in + Existence + Specific Detail

Output:
Q1: What is the largest commercial bank in the United States?
Q2: Does Chase offer personal bank accounts?
Q3: What information do I need to provide to open a bank account at Chase?

---

Input:
QO: What are the main responsibilities of the World Health Organization (WHO)?

Action:
Choose template type: 6. Definition + Lead-in + Specific Detail

Output:
Q1: What is the United Nations?
Q2: What are the branches of the United Nations?
Q3: What are the main responsibilities of the World Health Organization (WHO)?

---

Input:
QO: How does climate change affect the insurance industry?

Action:
Choose template type: 7. Definition + Cause + Specific Detail

Output:
Q1: What is climate change?
Q2: Why is climate change becoming a global concern?
Q3: How does climate change affect the insurance industry?

---

Input:
QO: What privacy features does WhatsApp offer?

Action:
Choose template type: 8. Different Angles + Existence + Specific Detail
```

```
Output:
Q1: Can I use WhatsApp for international calls?
Q2: Does WhatsApp have built-in privacy protection?
Q3: What privacy features does WhatsApp offer?

---

Input:
QO: What is AWS's pricing model like?

Action:
Choose template type: 9. Lead-in + Comparison + Specific Detail

Output:
Q1: How do companies choose a cloud provider?
Q2: Which is bigger, Google Cloud or AWS?
Q3: What is AWS's pricing model like?

---

Input:
QO: What is the role of smart contracts in blockchain platforms?

Action:
Choose template type: 10. Definition + Definition + Specific Detail

Output:
Q1: What is a blockchain platform?
Q2: What is a smart contract?
Q3: What is the role of smart contracts in blockchain platforms?

---

Input:
QO: {{QO_PLACEHOLDER}}
```

### Detect Information Overlap Prompt

```
Q1: {question_1}
A1: {answer_1}

Q2: {question_2}
A2: {answer_2}

Do A1 and A2 contain more than 10% of the overlapping information? If yes, answer 'Yes
    '. If not, answer 'No'.

Purpose:
```

## J.2 DECOMPOSE PROMPT

### Decompose Prompt

```
# OVERALL INSTRUCTIONS
You are an expert in understanding logical relationships. This is a Semantic Content
    Unit (SCU) extraction task. Given a pair of Question and Answer, your goal is to
    create a list of self-contained and concise claims. Each claim should be able to
    stand alone and be independent of other claims. Your claims should encompass all
    the information present in the answer.

# TASK INSTRUCTIONS
- List of Possible Causes: For scenarios involving multiple entities like red flags,
    vaccines, symptoms, etc., generate separate claims for each entity. This increases
     the number of claims.
- OR Claims: When entities are presented in an "OR" context, treat them as distinct
    claims.
- IF Claims: When an "if statement" is present, preserve the "if statement" context
    while creating the claim.
```

```
- XOR Claims: When entities have an XOR logical relationship (e.g., treatment options),
    create separate claims for each option.
- Try your best to list all the information. Do not miss any information.
- Instead of summarizing the original answer, break it down.

# EXAMPLE CLAIM FORMAT
- List Format: "Possible cause for [CONDITION] in [DEMOGRAPHIC] can be [ENTITY]."
- OR Format: "Possible causes include: [ENTITY X], [ENTITY Y], and [ENTITY Z]."
- OR Format: "The [CONTEXT] of treatments such as [TREATMENT X], [TREATMENT Y], and [
    TREATMENT Z], is not well established."
- IF Format: "[CONTEXT], please seek medical attention if [CONDITIONS]."
- XOR Format: "Either take [TREATMENT X] or [TREATMENT Y], but not both."

# TASK EXAMPLE
Question: I am a 33-year-old female with right lower abdominal pain, what could it be?
Answer: Possible causes for right lower abdominal pain in a young female are
    Appendicitis, Inflammatory bowel disease, Diverticulitis, Kidney stone, urinary
    tract infection, Ovarian cyst or torsion, Ectopic pregnancy, Pelvic inflammatory
    disease, endometriosis. Please seek medical attention if the pain is sudden and
    severe, does not go away, or gets worse, is accompanied by fever, nausea and
    vomiting, or if you have noticed blood in urine or in stool.
Claims:
[
Possible cause for right lower abdominal pain in a young female: Appendicitis,
Possible cause for right lower abdominal pain in a young female: Ovarian cyst or
    torsion,
Possible cause for right lower abdominal pain in a young female: Ectopic pregnancy,
Possible cause for right lower abdominal pain in a young female: Pelvic inflammatory
    disease,
Possible cause for right lower abdominal pain in a young female: Kidney stone,
Possible cause for right lower abdominal pain in a young female: Urinary tract
    infection,
Possible cause for right lower abdominal pain in a young female: Diverticulitis,
Possible cause for right lower abdominal pain in a young female: Inflammatory bowel
    disease,
Possible cause for right lower abdominal pain in a young female: Endometriosis,
Please seek medical attention if the pain is sudden and severe,
Please seek medical attention if the pain is accompanied by fever,
Please seek medical attention if the pain is accompanied by nausea and vomiting,
Please seek medical attention if the pain is accompanied by blood in urine,
Please seek medical attention if the pain is accompanied by blood in stool,
Possible cause for right lower abdominal pain in a young female: Emotional stress
]

# TASK EXAMPLE
Question: So what does the non reactive mean for the hep a igm
Answer: Hep A IgM refers to a specific type of antibody called Immunoglobulin M (IgM)
    against the virus hepatitis A. When infected with hepatitis A, these antibodies
    are detectable at symptom onset and remain detectable for approximately three to
    six months. These antibodies might also be detectable in the first month after
    hepatitis A vaccination. A negative or non-reactive result means no IgM antibodies
     against hepatitis A found in your serum, meaning the absence of an acute or
    recent hepatitis A virus infection.
Claims:
[
A negative or non-reactive result means that there were no IgM antibodies against
    hepatitis A found in your serum,
The absence of IgM antibodies against hepatitis A in your serum indicates the absence
    of an acute or recent hepatitis A virus infection,
Hep A IgM refers to a specific type of antibodies called Immunoglobulin M (IgM) against
     the virus hepatitis A,
These antibodies might also be detectable in the first month after hepatitis A
    vaccination,
These antibodies remain detectable for approximately three to six months after
    infection,
When infected with hepatitis A, these antibodies are detectable at the time of symptom
    onset
]

# TASK EXAMPLE
Question: What medications are contraindicated for a pregnant woman with ulcerative
    colitis?
Answer: methotrexate (Otrexup, Rasuvo, RediTrex) and thalidomide (Contergan, Thalomid)
    are both considered contraindicated for treatment of UC in pregnancy. Possible
    treatment for UC during pregnancy include low-risk drugs such as aminosalicylates
    (sulfasalazine and mesalamine), immunomodulators (azathioprine, cyclosporine A, 6-
```

```
        mercaptopurine) and corticosteroids. Biological agents such as Infliximab,
        Adalimumab, Vedolizumab and Ustekinumab are generally avoided during pregnancy as
        their safety in pregnancy is not well established yet.
Claims:
[
Methotrexate (Otrexup, Rasuvo, RediTrex) is contraindicated for treatment of ulcerative
     colitis in pregnancy,
Thalidomide (Contergan, Thalomid) is contraindicated for treatment of ulcerative
     colitis in pregnancy,
Aminosalicylates (sulfasalazine and mesalamine) are considered low-risk drugs for
     treatment of ulcerative colitis during pregnancy,
Immunomodulators (azathioprine, cyclosporine A, 6-mercaptopurine) are considered low-
     risk drugs for treatment of ulcerative colitis during pregnancy,
Corticosteroids are considered low-risk drugs for treatment of ulcerative colitis
     during pregnancy,
Treatment for ulcerative colitis during pregnancy with biological agents such as
     Adalimumab is generally avoided during pregnancy as their safety in pregnancy is
     not well established yet,
Treatment for ulcerative colitis during pregnancy with biological agents such as
     Vedolizumab is generally avoided during pregnancy as their safety in pregnancy is
     not well established yet,
Treatment for ulcerative colitis during pregnancy with biological agents such as
     Infliximab is generally avoided during pregnancy as their safety in pregnancy is
     not well established yet,
Treatment for ulcerative colitis during pregnancy with biological agents such as
     Ustekinumab is generally avoided during pregnancy as their safety in pregnancy is
     not well established yet
]

Question: What is the most safe amount of Advil to take at one time
Answer: The safe amount of Advil (ibuprofen) to take at one time depends on various
     factors, including your age, weight, and overall health condition. Here are
     general guidelines for adults:\n\n- **Standard Dosage**: The typical adult dose
     for pain relief is 200 mg to 400 mg every 4 to 6 hours as needed.\n- **Maximum
     Daily Dose**: The maximum daily dose for adults is usually 1200 mg to 3200 mg,
     depending on the formulation and the manufacturer's recommendations. However, it's
      important to stay within the lower end of this range unless specifically advised
     otherwise by a healthcare provider.\n\nFor example:\n- For mild to moderate pain,
     you might start with 200 mg every 4 to 6 hours.\n- For more severe pain, you might
      take 400 mg every 4 to 6 hours.\n\nIt's crucial to follow the specific
     instructions on the package or those provided by your healthcare provider. Overuse
      can lead to serious side effects, including stomach ulcers, bleeding, and kidney
     problems.\n\nIf you have any underlying health conditions or are taking other
     medications, it's best to consult with a healthcare provider to determine the
     safest dosage for you. Always read the label and follow the recommended dosages
     carefully.
Claims:
[
The safe amount of Advil (ibuprofen) to take at one time depends on various factors,
     including your age, weight, and overall health condition.
The safe amount of Advil (ibuprofen) to take at one time for an adult typically ranges
     from 200 mg to 400 mg,
Adults should take the standard dosage of Advil every 4 to 6 hours as needed for pain
     relief,
The maximum daily dose of Advil for adults usually ranges from 1200 mg to 3200 mg,
     depending on the formulation and manufacturer's recommendations,
It's important to stay within the lower end of ranges from 1200 mg to 3200 mg unless
     specifically advised otherwise by a healthcare provider,
For mild to moderate pain, the initial dosage of Advil for adults is 200 mg every 4 to
     6 hours,
For more severe pain, the dosage of Advil for adults can be increased to 400 mg every 4
     to 6 hours,
Overuse of Advil can lead to serious side effects such as stomach ulcers, bleeding, and
     kidney problems,
It is crucial to follow the specific instructions on the package or those provided by a
     healthcare provider when taking Advil,
Consulting a healthcare provider is advisable if you have underlying health conditions
     or are taking other medications before determining the safest dosage of Advil,
Always read the label and follow the recommended dosages carefully when taking Advil.
]

# YOUR TASK
Question: {question}
Answer: {answer}
Claims:
```

## J.3 DETECT CONTRADICT PROMPT

---

**Detect Contradict Prompt**

```
# OVERALL INSTRUCTIONS
- You have a deep understanding of logical relationships, such as entailment and
    contradiction, to evaluate given triplets of (question, premise, hypothesis).

# TASK INSTRUCTIONS
Your goal is to determine whether the Premise effectively contradicts the corresponding
     Hypothesis. Carefully analyze each triplet, focusing on details, not introducing
    knowledge.
- If the premise and the hypothesis are unrelated or lack sufficient evidence to
    ascertain their truthfulness, label your answer as False.
- be vigilant in identifying cases where the premise doesn't rule out the possibility
    of an entity (e.g., vaccine, symptom) appearing in the hypothesis. In such cases,
    classify the answer as False.
- If the answer is true, the answer should include "The answer is True". If the answer
    is false, the answer should include "The answer is False".
- Approach each question methodically, considering the step-by-step process outlined
    below.

# INPUT DATA
Question: What does trich test for? Let's think step by step.
Premise: The term "trich test" can refer to two different medical tests, depending on
    the context. Here are the two possibilities:
Trichomoniasis Test: Trichomoniasis is a sexually transmitted infection (STI) caused by
     the parasite Trichomonas vaginalis.
The trichomoniasis test, also known as a trich test or trichomonas test, is used to
    detect the presence of this parasite in the body. The test is typically performed
    on a sample of vaginal discharge in women or urine in men.
Trichogram: A trichogram is a diagnostic test used to evaluate hair loss and assess the
     health and condition of hair follicles. It involves plucking a small number of
    hairs from the scalp and examining them under a microscope.
It's important to note that without additional context, it's difficult to determine
    which specific test you are referring to.
Hypothesis: Trichamoniasis- a parasitic infection that can cause your symptoms.
Answer: According to the premise "trich test" refer to two different medical tests. A
    Trichamoniasis test is one of them, which is used to detect this parasite's
    presence.
As stated in the hypothesis, the trich test is used to diagnose parasitic infections.
    Ths premise entails the hypothesis. The answer is False.
###
Question: Can diabetics eat sweets? Let's think step by step.
Premise: Individuals with diabetes are recommended to limit their consumption of sweets
     to one or two times per week.
It is also suggested being selective with desserts and to focus on foods with a low
    glycemic index, such as high fiber foods like whole grains and legumes, as well as
     certain lower sugar fruits like berries, melons, and apples.
Hypothesis: It is recommended that diabetics avoid sweets.
Answer: The premise suggests that diabetics can eat sweets but limit their consumption.
    According to the hypothesis diabetics should avoid sweets.
Diabetics are allowed to consume sweets according to the premise, but they are
    prohibited according to the hypothesis. There is a contradiction between the
    premise and the hypothesis. The answer is True.
###
Question: 25 yo female with right lower abdominal pain, what might be causing it? Let's
    think step by step.
Premise: Right lower abdominal pain in a 25-year-old female could be caused by a
    variety of medical conditions.
Some potential causes include: Ovarian cyst: a fluid-filled sac on the ovary - Ectopic
    pregnancy: a pregnancy that occurs outside the uterus.
Hypothesis: possible cause for right lower abdominal pain in a young female can be
    Appendicitis.
Answer: The premise lists several potential causes of right lower abdominal pain in a
    25-year-old female, not including appendicitis. The hypothesis states that
    Appendicitis could be a cause of right lower abdominal pain in a young female.
There is no direct contradiction between the premise and the hypothesis, as the premise
     does not exclude the possibility of appendicitis as the cause of the pain. The
    answer is False.
###
Question: Can a headache last longer than a few days? Let's think step by step.
Premise: Yes, it is possible. If you are experiencing a headache that lasts longer than
     a few days, it is important to see a doctor to get the appropriate treatment.
This will help to relieve the pain and prevent any further complications.
Hypothesis: It is not a cause for concern if a headache lasts longer than a few days.
```

```
Answer: This premise acknowledges that a headache can last for several days, but
    emphasizes that seeing a doctor to prevent further complications is important.
    According to this hypothesis, headaches lasting longer than a few days are not
    cause of concern.
There is a contradiction between the premise and hypothesis due to the premise
    emphasizing the importance of seeking medical consultation, while the hypothesis
    posits that there is no cause for concern. The answer is True.
###
Question: {question} Let's think step by step.
Premise: {llm_answer}
Hypothesis: {answer}
Answer:
```

## J.4 DETECT ENTAIL PROMPT

### Detect Entail Prompt

```
# OVERALL INSTRUCTIONS
- You have a deep understanding of logical relationships, such as entailment and
    contradiction, to evaluate given triplets of (question, premise, hypothesis).

# TASK INSTRUCTIONS
Your goal is to determine whether the Premise effectively entails the corresponding
    Hypothesis. Carefully analyze each triplet, focusing on details, not introducing
    knowledge.
- If the premise disagrees with, is unrelated to, or does not support the hypothesis,
    there is not enough evidence to determine whether it is true, and so you answer
    should be False.
- If the answer is true, the answer should include "The answer is True". If the answer
    is false, the answer should include "The answer is False".
- Approach each question methodically, considering the step-by-step process outlined
    below.

# INPUT DATA
Question: What does trich test for? Let's think step by step.
Premise: The term "trich test" can refer to two different medical tests, depending on
    the context. Here are the two possibilities:
Trichomoniasis Test: Trichomoniasis is a sexually transmitted infection (STI) caused by
    the parasite Trichomonas vaginalis.
The trichomoniasis test, also known as a trich test or trichomonas test, is used to
    detect the presence of this parasite in the body. The test is typically performed
    on a sample of vaginal discharge in women or urine in men.
Trichogram: A trichogram is a diagnostic test used to evaluate hair loss and assess the
    health and condition of hair follicles. It involves plucking a small number of
    hairs from the scalp and examining them under a microscope.
It's important to note that without additional context, it's difficult to determine
    which specific test you are referring to.
Hypothesis: Trichamoniasis- a parasitic infection that can cause your symptoms.
Answer: According to the premise "trich test" refer to two different medical tests. A
    Trichamoniasis test is one of them, which is used to detect this parasite's
    presence.
As the hypothesis suggested, the trich test is used to diagnose parasitic infections.
    The premise entails the hypothesis. The answer is True.
###
Question: Can diabetics eat sweets? Let's think step by step.
Premise: Individuals with diabetes are recommended to limit their consumption of sweets
    to one or two times per week.
It is also suggested to be selective with desserts and to focus on foods with a low
    glycemic index, such as high fiber foods like whole grains and legumes, as well as
    certain lower sugar fruits like berries, melons, and apples.
Hypothesis: After eating sweets, must monitor blood and sugar level
Answer: The premise suggests that diabetics can eat sweets but limit their consumption.
    The hypothesis highlights the necessity of monitor blood and sugar after eating
    sweets.
There is no relationship between the premise and hypothesis, therefore they do not
    entail one another. The answer is False.
###
Question: Can diabetics eat sweets? Let's think step by step.
Premise: Individuals with diabetes are recommended to limit their consumption of sweets
    to one or two times per week.
It is also suggested being selective with desserts and to focus on foods with a low
    glycemic index, such as high fiber foods like whole grains and legumes, as well as
    certain lower sugar fruits like berries, melons, and apples.
Hypothesis: It is recommended that diabetics avoid sweets.
```

```
Answer: The premise suggests that diabetics can eat sweets but limit their consumption.
    According to the hypothesis diabetics should avoid sweets.
The premise allows diabetics to consume sweets in moderate consumption, while the
    hypothesis prohibits them. There premise don't entail the hypothesis. The answer
    is False.
###
Question: What is the best hypertension treatment for patients who are also have Crohn'
    s disease? Let's think step by step.
Premise: For patients with Crohn's disease and hypertension, the recommended treatment
    is a combination of lifestyle changes and medication. The ACC/AHA recommends
    initiation of antihypertensive drug therapy at a BP \u2265130/80 mm Hg for adults
    with hypertension.
It is also important to monitor your blood pressure regularly to make sure that it is
    under control.
Hypothesis: reducing sodium intake, are the first-line treatment for hypertension in
    individuals with  Crohn's disease
Answer: The premise suggests that the recommended treatment for patients with diabetes
    and hypertension is a combination of lifestyle changes and medication, including
    antihypertensive drug therapy. The hypothesis focuses on reducing sodium intake as
     the first-line treatment.
A reduction in sodium intake could be a part of the lifestyle changes, but since it is
    not mentioned in the premise, the premise do not entail the hypothesis. The answer
     is False.
###
Question: 25 yo female with right lower abdominal pain, what might be causing it? Let's
     think step by step.
Premise: Right lower abdominal pain in a 25-year-old female could be caused by a
    variety of medical conditions.
Some potential causes include: - Appendicitis: inflammation of the appendix -
    Appendiceal abscess: a collection of pus in the appendix - Ovarian cyst: a fluid-
    filled sac on the ovary - Ectopic pregnancy: a pregnancy that occurs outside the
    uterus.
Hypothesis: possible cause for right lower abdominal pain in a young female can be
    Appendicitis.
Answer: The premise lists several potential causes of right lower abdominal pain in a
    25-year-old female, including appendicitis. The hypothesis states that
    Appendicitis could be a cause of right lower abdominal pain in a 25-year-old
    female.
Both the premise and hypothesis mention appendicitis as a possible cause of pain, so
    the premise entails the hypothesis. The answer is True.
###
Question: {question} Let's think step by step.
Premise: {llm_answer}
Hypothesis: {answer}
Answer:
```

# K  CASE STUDY

## K.1  CASE STUDY 1: ENHANCEMENT OF FACT ACCURACY THROUGH RETRIEVAL

### Case Study 1: Enhancement of Fact Accuracy Through Retrieval

```
Q: How long would I be on Saxenda before I could move to the next dose?

GT: ''It is recommended that adult patients start with a dose of 0.6 mg per day for one
     week. The dose is then increased by 0.6 mg each week over the following 4 weeks
    until the daily maintenance dose of 3 mg is reached. If a patient does not
    tolerate an increased dose during the dosing schedule, dose escalation can be
    delayed by one extra week.''

base_answer: ''The dosing schedule for Saxenda (liraglutide) is designed to gradually
    increase over time to achieve the desired therapeutic effect. Heres a general
    outline of how the dosing progression works:\n\n1. Initial Dose: Saxenda is
    typically started at a low dose, usually 0.6 mg once daily via subcutaneous
    injection. This initial dose is often given for 2 weeks to allow the body to
    adjust to the medication.\n\n2. First Increase: After the first 2 weeks, the dose
    is increased to 1.2 mg once daily. This step is also typically maintained for 2
    weeks to ensure the body continues to adapt.\n\n3. Second Increase: After another
    2 weeks at 1.2 mg, the dose is increased to 1.8 mg once daily. This final dose is
    then maintained for the duration of treatment unless otherwise directed by a
    healthcare provider.\n\nThe exact timing and duration can vary based on individual
     response and medical supervision. Some patients might stay on a particular dose
```

longer if they are experiencing side effects or if their healthcare provider decides it is necessary.\n\nIt's important to follow the specific instructions provided by your healthcare provider, as they will monitor your progress and adjust the dosing schedule as needed. Regular follow-up appointments are crucial to assess your response to the medication and make any necessary adjustments.\n\nIf you have any questions about your dosing schedule or if you experience any side effects, it's important to contact your healthcare provider for guidance.''

RAG_answer: ''For Saxenda, you would be on each dose for one week before moving to the next higher dose. Specifically, the dose escalation schedule is as follows:\n\n- Week 1: 0.6 mg daily\n- Week 2: 1.2 mg daily\n- Week 3: 1.8 mg daily\n- Week 4: 2.4 mg daily\n- Week 5 and onward: 3 mg daily\n\nAfter completing one week at each dose, you would then increase to the next higher dose. If you or a pediatric patient cannot tolerate the increased dose during the dose escalation process, you may delay the escalation for approximately one additional week.''

RAG_retrieved_chunks: "The following information has been retrieved and may be helpful:\n\n1.\n\"\"\"\n\nUse SAXENDA with a reduced-calorie diet and increased physical activity.\nSAXENDA is injected under the skin (subcutaneously) in your stomach area (abdomen), upper leg (thigh), or upper arm.\nDo not\ninject into a muscle (intramuscularly) or vein (intravenously).\nSAXENDA is injected 1 time each day, at any time during the day.\nStart SAXENDA with 0.6 mg per day in your first week. In your second week, increase your daily dose to 1.2 mg. In the third week, increase your daily dose to 1.8 mg. In the fourth week, increase your daily dose to 2.4 mg and in the fifth week onwards, increase your daily dose to the full dose of 3 mg. After that, do not change your dose unless your healthcare provider tells you to. Children may reduce their dose to 2.4 mg daily if the maximum dose is not tolerated.\nIf you miss your daily dose of SAXENDA, just take your next daily dose as usual on the following day. Do not take an extra dose of SAXENDA or increase your dose on the following day to make up for your missed dose. If you miss your dose of SAXENDA for\n3 days or more\n, call your healthcare provider to talk about how to restart your treatment.\n\nSAXENDA may be taken with or without food.\n\nChange (rotate) your injection site with each injection.\nDo not\nuse the same site for each injection.\nDo not share your SAXENDA pen with other people, even if the needle has been changed.\nYou may give other people a serious infection or get a serious infection from them.\nIf you take too much SAXENDA, call your healthcare provider or Poison Help line at 1-800-222-1222 or go to the nearest hospital emergency room right away.\n\nThrow away the used SAXENDA pen after 30 days.\nWhat are the possible side effects of SAXENDA?\nSAXENDA may cause serious side effects, including:\n\nSee What is the most important information I should know about SAXENDA?\n\ninflammation of the pancreas\n(\npancreatitis).\n\n\"\"\"\n\n2.\n\"\"\"\n\nTable 1. Dose Escalation Schedule\nWeek\nDaily Dose\n1\n0.6 mg\n2\n1.2 mg\n3\n1.8 mg\n4\n2.4 mg\n5 and onward\n3 mg\nAdult Patients\n\nFor adults, the recommended dosage of SAXENDA is 3 mg daily, lower dosages are for titration only.\nDiscontinue SAXENDA if the patient cannot tolerate the 3 mg dosage.\nIf patients do not tolerate an increased dose during dosage escalation, consider delaying dosage escalation for approximately one additional week.\nEvaluate the change in body weight 16 weeks after initiating SAXENDA and discontinue SAXENDA if the patient has not lost at least 4% of baseline body weight, since it is unlikely that the patient will achieve and sustain clinically meaningful weight loss with continued treatment.\n\nIn adult patients with type 2 diabetes, monitor blood glucose prior to starting SAXENDA and during SAXENDA treatment.\nPediatric Patients\n\nFor pediatric patients, the recommended maintenance dosage of SAXENDA is 3 mg daily. Pediatric patients who do not tolerate 3 mg daily may have their maintenance dose reduced to 2.4 mg daily. Discontinue SAXENDA if the patient cannot tolerate the 2.4 mg dose.\nIf pediatric patients do not tolerate an increased dose during dosage escalation, the dose may also be lowered to the previous level. Dosage escalation for pediatric patients may take up to 8 weeks.\nEvaluate the change in BMI after 12 weeks on the maintenance dose and discontinue SAXENDA if the patient has not had a reduction in BMI of at least 1% from baseline, since it is unlikely that the patient will achieve and sustain clinically meaningful weight loss with continued treatment.\n3 DOSAGE FORMS AND STRENGTHS\nInjection: 6 mg/mL clear, colorless solution in a 3 mL prefilled, single-patient-use pen that delivers doses of 0.6 mg, 1.2 mg, 1.8 mg, 2.4 mg, or 3 mg.\n\n\"\"\"\n\n3.\n\"\"\"\n\nHome\n/\nAbout Saxenda\n\n/ Dosing Schedule\nSaxenda\n\nDosing Schedule\nGet help staying on track with Saxenda\n\nby connecting with your prescribing health care provider, reviewing the daily dosing schedule below, or calling the Saxenda\n\nHotline at 1-844-845-6913 Monday through Friday 8:30\nAM\nto 6:00\nPM\nET.\nNeed reminders for when to take Saxenda\n\n? Try setting an alarm on a smartphone or AI assistant (like Alexa or Siri).\nReview the Saxenda\n\ndosing schedule\nThe Saxenda\n\ndaily dosing schedule was designed to help patients get used to their new medicine while also minimizing gastrointestinal side effects. Review the schedule below, and be sure to schedule a 4-month follow-up with your prescribing health care provider. This visit can be sooner if there are questions or concerns about dosing or side effects. As a reminder, a Saxenda\n\nprescription contains enough medicine for 30 days.\nBelow is a dosing schedule to guide you through the starting dosage of 0.6 mg to the dosage of 3 mg.\nSaxenda\n\nshould be taken exactly as prescribed. After

```
 starting Saxenda\n\n, provided there are no issues with tolerating it, the dose
should be increased weekly until the 3 mg dose is reached. If you or your teen
have trouble tolerating an increased dose, or if you have any other questions, be
sure to call your (or your teen's) health care provider. The Saxenda\n\ndose
should only be changed if your (or your teen's) health care provider advises to do
 so.\nNausea is the most common side effect when first starting Saxenda\n\n. Learn
 more about ways to manage it\nhere\n. For additional side effects, please refer
to the\nMedication Guide\n.\nIf you (or your teen) miss your daily dose of Saxenda
\n\n, just take the next daily dose as usual on the following day. Do not take an
extra dose of Saxenda\n\nor increase the dose on the following day to make up for
the missed dose. If Saxenda\n\nis not taken for 3 days or more, call your
respective health care provider to talk about how to restart treatment.\n\"\"\"\n4
.\n\"\"\"\nHow long each pen lasts depends on the dose you take. There is a scale
on the pen that shows you about how much Saxenda is left inside.\nWhat is the
average weight loss with Saxenda?\nIn clinical studies, a greater number of people
 who took Saxenda for 56 weeks lost 5 to 10% of their body weight, compared with
people who took a placebo. (A placebo is a substance that has no effect and is
used as a control when testing medications). This means participants lost 12 to 23
 pounds. People in the studies were overweight (BMI 27-29.9 kg/m2) or obese (BMI
greater than or equal to 30 kg/m2) before taking Saxenda.\nCan you drink alcohol
with Saxenda?\nThere is no specific warning about drinking alcohol with Saxenda.
Keep in mind that alcohol can lower blood sugar. If you are taking other drugs for
 diabetes, drinking alcohol increases your risk of hypoglycemia. In addition, many
 alcoholic drinks contain high levels of carbohydrates and sugar. If youre trying
to lose weight, you may want to avoid alcohol.\nHow long does it take for Saxenda
to work?\nSaxenda reaches its maximum concentration in the body 11 hours after
injection. Its recommended to follow up with your doctor 2 to 8 weeks after
starting Saxenda to see if its working.\nIf you have not lost 4% of your body
weight after 16 weeks, your doctor may tell you to stop taking it.\nIn children
ages 12 and up, Saxenda may be stopped after 12 weeks on the maintenance dose if
BMI has not decreased by 1%.\nWhy am I not losing weight on Saxenda?\nIt takes
time to lose weight with Saxenda. It may take about 8 weeks before you start to
see significant weight loss (about 5%) with Saxenda, but in the first 2 to 4 weeks
 you may lose about 2% to 4% of your weight. You should also adhere to a long-term
 reduced-calorie diet and exercise program as prescribed by your doctor for
maximum weight loss.\nContinue reading\nLiraglutide vs Semaglutide: How do they
compare?\n\"\"\"\n5.\n\"\"\"\nRefer to the accompanying Instructions for Use for
complete administration instructions with illustrations.\n\nInspect SAXENDA
visually prior to each injection. Only use if solution is clear, colorless, and
contains no particles.\n\nInject SAXENDA subcutaneously once daily at any time of
day, without regard to the timing of meals.\n\nInject SAXENDA subcutaneously in
the abdomen, thigh, or upper arm. No dose adjustment is needed if changing the
injection site and/or timing.\n\nRotate injection sites within the same region
in order to reduce the risk of cutaneous amyloidosis.\n\nIf a dose is missed,
resume the once-daily regimen as prescribed with the next scheduled dose. Do not
administer an extra dose or increase the dose to make up for the missed dose.\n\
nIf more than 3 days have elapsed since the last SAXENDA dose, reinitiate SAXENDA
at 0.6 mg daily and follow the dose escalation schedule in Table 3, to reduce the
occurrence of gastrointestinal adverse reactions associated with reinitiation of
treatment.\nDosage in Adults and Pediatric Patients Aged 12 Years and Older\n\
nInitiate SAXENDA with a dose of 0.6 mg daily for one week. Then follow the dose
escalation schedule in Table 3 to minimize gastrointestinal adverse reactions.\
nTable 3: Dose Escalation Schedule\nWeek\nDaily Dose\n1\n0.6 mg\n2\n1.2 mg\n3\n1.8
 mg\n4\n2.4 mg\n5 and onward\n3 mg\nAdult Patients\n\nFor adults, the recommended
dosage of SAXENDA is 3 mg daily, lower doses are for titration only.\n\
nDiscontinue SAXENDA if the patient cannot tolerate the 3 mg dose.\n\nIf patients
do not tolerate an increased dose during dose escalation, consider delaying dose
escalation for approximately one additional week.\n\nEvaluate the change in body
weight 16 weeks after initiating SAXENDA and discontinue SAXENDA if the patient
has not lost at least 4% of baseline body weight, since it is unlikely that the
patient will achieve and sustain clinically meaningful weight loss with continued
treatment.\n\nIn adult patients with type 2 diabetes, monitor blood glucose prior
to starting SAXENDA and during SAXENDA treatment.\nPediatric Patients\n\nFor
pediatric patients, the recommended maintenance dosage of SAXENDA is 3 mg daily.\n
\"\"\"\n\nPlease answer the following question based only on the information above
:\nHow long would I be on Saxenda before I could move to the next dose?"
```

The question concerns how long a patient typically stays on each dose of Saxenda (liraglutide) before increasing to the next one. The gold-standard text clearly provides the recommended escalation schedule: adults start at 0.6 mg per day for one week, and the dose is then increased by 0.6 mg each week over the next four weeks until reaching the 3 mg daily maintenance dose. If a patient has difficulty tolerating a specific dose, the escalation can be delayed by an extra week.

The base model's answer performs poorly in terms of factual accuracy because it does not reproduce this stepwise weekly schedule. Instead, it offers a generic description of how GLP-1 agonists are usually titrated, using vague statements such as starting at a low dose and often given for 2 weeks,

without providing the precise weekly increments or the fixed five-week escalation plan. Its statements focus on a general idea of gradual dose adjustment under medical supervision rather than the specific information contained in the official product guidelines, which explains why its factual score is essentially zero.

In contrast, the RAG-enhanced answer closely matches the gold-standard dosing schedule. It explicitly lays out the weekly plan0.6 mg in week one, followed by 1.2 mg, 1.8 mg, and 2.4 mg, and finally reaching 3 mg by week five. It also includes the key detail that dose escalation may be delayed by approximately one week if the patient does not tolerate the increase. This alignment with the official instructions results in a high factual score. The retrieval system appears to have pulled information directly from authoritative sources such as Drugs.com, enabling the model to provide a precise, accurate dose-escalation timeline rather than relying on general clinical intuition.

This case illustrates how retrieval greatly improves performance for topics where the correct answer depends on exact, non-negotiable detailssuch as fixed titration schedules for prescription medications. Without retrieval, the model tends to generalize from patterns it has learned, but with retrieval, it can reproduce the precise regimen specified in official guidelines.

## K.2 CASE STUDY 2: PERFORMANCE DEGRADATION RESULTING FROM MISALIGNED RETRIEVAL FOCUS

---

**Case Study 2: Performance Degradation Resulting from Misaligned Retrieval Focus**

Q: What are the two most prevalent short-term debt instruments in the money market?

GT: "Most Common Short-Term Bonds in Money Markets: T-Bills and Commercial Paper."

base_answer: "The two most prevalent short-term debt instruments in the money market are typically:\n\n1. Treasury Bills (T-Bills): These are short-term debt obligations issued by the government with maturities ranging from a few days to one year. They are considered very safe due to the backing of the government and are widely used for short-term investments and liquidity management.\n\n2. Commercial Paper: This is an unsecured promissory note issued by corporations with maturities ranging from a few days to 270 days. Commercial paper is used for short-term financing and is often issued by large, creditworthy companies to raise funds quickly.\n\nThese two instruments are highly liquid and are commonly traded in the money market, making them essential components of short-term investment strategies."

RAG_answer: "Based on the information provided, the two most prevalent short-term debt instruments in the money market are:\n\n1. Treasury Bills (T-bills) – Short-term debt obligations backed by the U.S. government, sold in denominations of $1,000 up to a maximum of $5 million, with maturities ranging from four to 52 weeks.\n2. Commercial Paper – Unsecured, short-term debt issued by large corporations to fund day-to-day operations, typically maturing within 270 days and issued at a discount to face value.\n\nThese instruments are widely recognized for their liquidity, low risk, and role in stabilizing interest rates and managing short-term funding needs."

RAG_retrieved_chunks: "The following information has been retrieved and may be helpful:\n\n1.\n\"\"\"\nTreasury notes and bonds are not included here. Bonds and other fixed-income instruments with longer terms are considered part of the capital markets.\nRepurchase Agreements (Repos)\n:\nThese assets involve the sale of securities with an agreement to repurchase them at a slightly higher price on a specific future dateoften the next day. They're essentially short-term loans, typically used by dealers in government securities. The securities serve as collateral, making repos relatively low-risk.\nCommercial Paper:\nCommercial paper consists of unsecured, short-term debt issued by large corporations to fund day-to-day operations. These instruments typically mature within 270 days and are issued at a discount to face value.\nWhile riskier than T-bills, commercial paper from top-rated companies often offers slightly higher yields.\nCDs:\nCDs have fixed terms ranging from a few weeks to several years and pay higher interest rates than standard savings accounts, though the depositor has to wait a period to obtain the funds back. Keep in mind that CDs that mature after one year aren't part of the money market.\nPrime funds typically invest in short-term obligations issued by banks and corporations. Two of the most common are listed below.\nBankers' Acceptances:\nThese are short-term debt instruments guaranteed by a bank, often used for international trade. When a bank accepts a\nbankers' acceptance\n, it assumes responsibility for paying the holder when the instrument matures. This bank guarantee makes them relatively safe investments.\nShort-Term Corporate

Bonds:\nFor inclusion in the money market, these corporate bonds have maturities of one year or less. This is debt issued by companies that offer a way for corporations to borrow money from investors for relatively brief periods, often to fund operations, finance projects, or refinance existing debt.\nFast Fact\nThe highest quality (and safest, lower yielding) bonds are commonly referred to as Triple-A bonds, while the least creditworthy are termed junk.\nTax-exempt funds are composed of short-term government-issued debt obligations. The interest income earned on these investments is exempt from federal taxes. Some may also be exempt from local and state taxes.\nMunicipal Bonds (Munis):\nTax-exempt money market funds are primarily municipal bonds or notes, which are issued by state and local governments. These are often tax-exempt at the federal level, making them attractive to investors in high tax brackets. The chart below shows the $6.57 trillion U.S.\n\"\"\"\n2.\n\"\"\"\nThey work through instruments like\ncommercial paper\n, Treasury bills (T-bills), and\ncertificates of deposit (CDs)\n. These instruments facilitate quick fund transfers and help to stabilize interest rates. They are often regarded as a haven for investors to park their surplus cash and keep the system liquid and stable.\nIn the money markets, banks, corporations, and government entities buy and sell financial instruments to manage liquidity. These transactions involve instruments like\nT-bills\nand commercial paper where terms are shorter and settlement is quick. This fast-paced activity helps participants to manage their short-term funding needs.\nMoney market operations are crucial for the level of liquidity and interest rates in the economy. They provide quick access to cash and stabilize interest rates so they are more predictable. The quick turnaround of funds allows investors to park their money temporarily and supports monetary stability.\nTypes and Examples of Money Markets\nMoney markets play a crucial role in the financial system, providing a place for institutions and individuals to park cash safely for short periods. These markets deal in highly liquid, short-term debt instruments, typically with maturities of one year or less. Let's explore the main types of money market instruments and how they function.\nGovernment money market funds primarily invest in short-term securities issued by the U.S. government, such as T-bills and other government-backed instruments. They are considered very safe and liquid, offering a lower yield but greater security compared with prime funds.\nT-Bills:\nThese are short-term debt obligations backed by the U.S. government and sold in denominations of $1,000 up to a maximum of $5 million. T-bills mature in four, eight, 13, 26, or 52 weeks.\nInvestors buy them at a discount and receive the full face value when they mature with the difference representing the interest earned. Treasury notes and bonds are not included here. Bonds and other fixed-income instruments with longer terms are considered part of the capital markets.\nRepurchase Agreements (Repos)\n:\nThese assets involve the sale of securities with an agreement to repurchase them at a slightly higher price on a specific future dateoften the next day. They're essentially short-term loans, typically used by dealers in government securities. The securities serve as collateral, making repos relatively low-risk.\nCommercial Paper:\nCommercial paper consists of unsecured, short-term debt issued by large corporations to fund day-to-day operations.\n\n\"\"\"\n3.\n\"\"\"\nMoney Markets vs. Capital Markets: What's the Difference?\nTable of Contents\nExpand\nTable of Contents\nAn Overview\nMoney Markets\nCapital Markets\nKey Differences\nAlternatives\nRegulation & Oversight\nFAQs\nThe Bottom Line\nMoney Markets vs. Capital Markets: An Overview\nMoney and capital markets are fundamental to the economy, serving investors and businesses alike.\nMoney markets\ndeal in short-term debt instruments, usually for one year or less. It's where governments, banks, and large corporations go to manage their immediate cash needs.\nCapital markets\ninvolve long-term securities, such as stocks and bonds, that mature in more than one year. This is where companies and governments raise funds for major projects and long-term growth.\nMoney markets are the lifeblood of day-to-day financial operations, while capital markets sustain long-term\neconomic growth\n. They differ in three ways: the types of financial instruments traded, the duration of investments, and the level of risk. While the money market prioritizes liquidity and safety, the capital market offers the potential for higher returns with increased risk. Below, we'll explore each market's characteristics and how they work.\nKey Takeaways\nMoney markets involve short-term lending that borrowers can tap into for cash for day-to-day operations\nCapital markets are geared toward long-term investing.\nMoney markets are less risky than capital markets, which can be more rewarding.\nBoth markets are subject to comprehensive regulation to ensure transparency, fairness, and stability.\nANGELA WEISS\n/ Contributor / Getty Images\nMoney Markets\nMoney markets are meant for short-term lending and borrowing, usually for a year or less. Its like a fast lane where businesses, governments, and\nfinancial institutions\ncan meet their quick funding needs. Thus, it is important for\nliquidity\nmanagement. These markets are known for their high liquidity, generally low risk, and ease of access to capital.\nThey work through instruments like\ncommercial paper\n, Treasury bills (T-bills), and\ncertificates of deposit (CDs)\n. These instruments facilitate quick fund transfers and help to stabilize interest rates. They are often regarded as a haven for investors to park their surplus cash and keep the system liquid and stable.\nIn the money markets, banks, corporations, and government entities buy and sell financial instruments to manage liquidity. These transactions involve instruments like\nT-bills\nand commercial paper where terms are shorter and settlement is quick.\n\n\"\"\"\n4.\n\"\"\"\nTax-exempt funds are composed of short-term government

```
        -issued debt obligations. The interest income earned on these investments is
        exempt from federal taxes. Some may also be exempt from local and state taxes.\
        nMunicipal Bonds (Munis):\nTax-exempt money market funds are primarily municipal
        bonds or notes, which are issued by state and local governments. These are often
        tax-exempt at the federal level, making them attractive to investors in high tax
        brackets. The chart below shows the $6.57 trillion U.S. money market broken down
        under the main headings used here:\nTip\nThe money market is far broader than
        money market funds or accounts available at banks and other financial institutions
        . While related, the latter is a mutual fund that invests in high-quality, short-
        term debt instruments and cash equivalents. Many are also insured by the\nFederal
        Deposit Insurance Corporation (FDIC)\n.\nCapital Markets\nCapital markets play a
        vital role in economic growth by channeling savings into productive investments.
        They are where longer-term securities are bought and sold. Companies and
        governments raise funds by issuing stocks (equities) and bonds (fixed-income
        securities). Investors can earn returns via value\nappreciation\nor distributions
        .\nTransactions enable individuals and institutions to tap into future
        opportunities. Investors buy long-term instruments like\nstocks and bonds\nfrom
        issuers in primary markets or trade them in secondary markets. This helps
        companies and governments get the funds they need for various projects and
        objectives. Investors hope to get returns through\ndividends\nor interest and
        potential appreciation.\nBuyers and sellers are matched through exchanges or\nover
        -the-counter (OTC)\nplatforms. Brokers and dealers play a key role in facilitating
         transactions and keeping things smooth. Pricing in the capital markets is driven
        by supply and demand, investor sentiment, and economic indicators.\nFacilitating
        the trade of financial assets helps set asset prices and ensures a certain degree
        of liquidity, allowing funds to move smoothly. Consequently, this market underpins
         business\nexpansion\nand bolsters the overall economy.\nTypes and Examples of
        Capital Markets\nCapital markets can be broken down into the primary and secondary
         markets. The primary market is where new securities are sold for the first time,
        such as when a company goes public with an\ninitial public offering (IPO)\n. This
        allows companies to raise capital directly from investors who buy these new shares
        .\n\"\"\"\n5.\n\"\"\"\nIf you want short-term, low-risk investments with quick
        returns, the money market is probably the way to go. Instruments like Treasury
        bills help you preserve capital and provide liquidity over shorter periods.\nMost
        investors have a long-term time horizon and turn to capital markets. Investing in
        stocks and/or bonds can build wealth and align with long-term financial goals
        while riding out market fluctuations.\nHow Do Geopolitical Events Affect Money
        Markets?\nGeopolitical events increase volatility and risk and cause a flight to
        safety in money markets as investors seek safe havens.\nWhat Role Do Central Banks
         Have in the Money Markets?\nCentral banks\ninfluence money markets by setting
        interest rates and conducting open market operations to manage liquidity. The U.S.
         Federal Reserve serves in this role in the U.S.\nWhy Are the Capital Markets
        Important for Startups?\nCapital markets provide startups with access to funding
        through IPOs and venture capital, fueling their growth.\nThe Bottom Line\nCapital
        and money markets are the fundamental pillars of the modern financial system, each
         serving distinct yet complementary roles. Capital markets, comprising stocks,
        bonds, and other long-term securities, enable businesses and governments to raise
        funds for long-term investments and expansion. These markets offer investors the
        potential for higher returns, but often with increased risk and\nvolatility\n.\
        nMoney markets, meanwhile, focus on short-term, highly liquid instruments such as
        Treasury bills and commercial paper. They serve as the economy's lubricant,
        facilitating short-term borrowing and lending and providing a relatively safe
        haven for cash management. While money market instruments typically offer lower
        returns, they provide essential liquidity and stability to the financial system.\
        nArticle Sources\nInvestopedia requires writers to use primary sources to support
        their work. These include white papers, government data, original reporting, and
        interviews with industry experts. We also reference original research from other
        reputable publishers where appropriate. You can learn more about the standards we
        follow in producing accurate, unbiased content in our\neditorial policy.\nStatista
        . \"\nLargest stock exchange operators worldwide as of March 2024, by market
        capitalization of listed companies\n.\"\nBoard of Governors of the Federal Reserve
         System. \"\nAbout Commercial Paper\n.\"\nCompare Accounts\nAdvertiser Disclosure\
        n\nThe offers that appear in this table are from partnerships from which
        Investopedia receives compensation. This compensation may impact how and where
        listings appear. Investopedia does not include all offers available in the
        marketplace.\n\"\"\"\n\nPlease answer the following question based only on the
        information above:\nWhat are the two most prevalent short-term debt instruments in
         the money market?"
```

This case looks at a situation where retrieval makes the answer worse. The question asks which two short-term debt instruments are the most common in the money market. The gold-standard answer is very short and simply states that the most prevalent instruments are Treasury Bills and Commercial Paper.

The base model, without retrieval, performs reasonably well and reaches a factual F1 score of about 0.62. It correctly identifies the two instruments and provides accurate explanations of their issuers,

maturities, and uses. Its statements clearly match the key facts in the gold-standard answer. Although the base answer contains more description and is longer than necessary, the evaluation system still recognizes that it accurately covers the essential information. With retrieval, the factual score drops dramatically to around 0.125. The answer produced with retrieval still names the correct instruments. It also includes additional details such as typical denominations and precise maturity ranges. The retrieved evidence itself even supports the idea that T-Bills and commercial paper are commonly used in the money market.

The problem comes from the answer's shift in emphasis. After incorporating retrieved content, the model focuses more on technical characteristics, such as the structure and numerical parameters of T-Bills. It focuses less on the central point that these two instruments are the most prevalent ones. Since the evaluation places strong weight on that specific fact about prevalence, the retrieved answer ends up covering the key point less directly. As a result, the factual match score becomes much lower, even though the content is technically correct and more detailed.

This case shows that retrieval does not always help. When the gold-standard answer highlights a specific categorical fact about importance or ranking, too much detail from retrieval can distract the model from stating that key fact clearly. The answer becomes more elaborate, but the essential information becomes blurred, which leads to a decline in factual accuracy during evaluation.

### K.3 CASE STUDY 3: NEUTRAL IMPACT OF RETRIEVAL ON REGULATORY INTERPRETATION

---

**Case Study 3: Neutral Impact of Retrieval on Regulatory Interpretation**

Q: "In producing the comprehensive cash flow projections for its U.S. operations, will an FBO be required to consolidate cash flows of entities subject to Regulation W, such as banks, with cash flows of entities not subject to Regulation W?"

GT: "Yes. As set forth in section 252.156(d) of Regulation YY, an FBO must produce comprehensive cash-flow projections for its combined U.S. operations. The comprehensive cash flow projections must include the cash flows of all entities included in the combined U.S. operations, regardless of whether the entities are subject to Regulation W."

base_answer: "In preparing comprehensive cash flow projections for its U.S. operations, an FBO (Financial Holding Company) is generally required to consolidate the cash flows of all its subsidiaries, including those subject to Regulation W, with the cash flows of entities not subject to Regulation W. This consolidation is typically required under Generally Accepted Accounting Principles (GAAP) and the rules governing financial reporting for holding companies.\n\nHere are some key points to consider:\n\n1. Consolidation Requirements: Under GAAP, an FBO must consolidate the financial statements of all its subsidiaries, including banks and other financial institutions, into its own financial statements. This means that the cash flows of all consolidated entities should be included in the FBO's cash flow projections.\n\n2. Regulation W: Entities subject to Regulation W, such as banks, are part of the FBO's consolidated financial statements. Therefore, their cash flows must be included in the FBO's consolidated cash flow projections. This includes any cash flows related to loans extended to insiders, which are subject to the provisions of Regulation W.\n\n3. Comprehensive Cash Flow Projections: When preparing comprehensive cash flow projections, the FBO should consider all sources of cash inflows and outflows, including those from its banking subsidiaries. This ensures a complete picture of the company's liquidity and financial health.\n\n4. Regulatory Compliance: While Regulation W itself does not specifically mandate the inclusion of cash flows in projections, the broader regulatory framework and the need for accurate financial reporting under GAAP require this consolidation.\n\n5. Disclosure Requirements: The FBO must ensure that its financial reports and disclosures comply with all applicable regulations, including those related to cash flow projections. This includes providing transparent and accurate information about the cash flows of all consolidated entities.\n\nIn summary, an FBO must consolidate the cash flows of all its subsidiaries, including those subject to Regulation W, with the cash flows of entities not subject to Regulation W, in order to produce comprehensive and accurate cash flow projections for its U.S. operations."

RAG_answer: "Based on the information provided, an FBO (Foreign Banking Organization) will be required to consolidate cash flows of entities subject to Regulation W, such as banks, with cash flows of entities not subject to Regulation W when producing comprehensive cash flow projections for its U.S. operations. Here's the reasoning:\n\n1. Regulation W Requirements: The regulations specify that an FBO must produce comprehensive cash-flow projections for its combined U.S. operations. This includes projecting cash flows arising from transactions with entities that

---

are not affiliates.\n\n2. Cash Flow Projections Scope: The cash flow projections must account for cash-flow needs and sources resulting from transactions between the U.S. intermediate holding company and entities that are not its affiliates. This implies that all relevant cash flows, regardless of whether the entities are subject to Regulation W, must be included in these projections.\n\n3. Net Internal and External Cash Flow Needs: The calculations for net internal and external stressed cash-flow needs involve detailed projections of cash flows from various transactions, including those with non-affiliates. These calculations are integral parts of the overall cash flow projection process.\n\nGiven these points, it is clear that the FBO must consider and include the cash flows of all relevant entities, whether or not they are subject to Regulation W, in its comprehensive cash flow projections for its U.S. operations. Therefore, the FBO will be required to consolidate cash flows of entities subject to Regulation W with those of entities not subject to Regulation W."

RAG_retrieved_chunks: "The following information has been retrieved and may be helpful :\n\n1.\n\"\"\"\n(d)Cash-flow pr ojections.\n(1)A for eign banking or ganization subject t o this subpar t must pr oduce compr ehensiv e cash-flow\nprojections for its combined U.S.The net external str essed cash-flow need for a\nU.S. intermediate holding company equals the diff erence between:\n(A)The pr ojected amount of cash-flow needs that r esults fr om tr ansactions between the U.S.\nintermediate holding company and entities that ar e not its affiliates; and\n(B) The pr ojected amount of cash-flow sour ces that r esults fr om tr ansactions between the\nU.S. intermediate holding company and entities that ar e not its affiliates.(C)Net intr agroup cash flow .For any giv en da y of the planning horiz on, the net intr agroup cash\nflow must equal the diff erence between:\n(1)The amount of pr ojected cash-flow needs r esulting fr om tr ansactions between a U.S .\nbranch or U.S. agency and the for eign bank' s non-U.S. offices and its affiliates; and\n(2)The amount of pr ojected cash-flow sour ces r esulting fr om tr ansactions between a\nU.S.12 CF R P art 25 2 (up t o d ate as of 12/14/2023)\nEnhanc ed P rudenti al S tand ards (R egul ation Y Y)12 CF R 25 2.156(b)(4)\n12 CF R 25 2.156( d)(1) ( enhanc ed displ ay) pag e 115 of 192\n\n(2)The for eign banking or ganization must establish a methodology for making cash-flow pr ojections for\nits combined U.S.t o the extent permitted b y applicable law .\n(e) Cash-flow pr ojections.\n(1)A bank holding company subject t o this subpar t must pr oduce compr ehensiv e cash-flow pr ojections\nthat pr oject cash flows arising fr om assets, liabilities, and off-balance sheet exposur es o ver, at a\nminimum, shor t- and long-term time horiz ons.\n\"\"\"\n2.\n\"\"\"\n(e)Cash-flow pr ojections.\n(1)A bank holding company subject t o this subpar t must pr oduce compr ehensiv e cash-flow pr ojections\nthat pr oject cash flows arising fr om assets, liabilities, and off-balance sheet exposur es o ver, at a\nminimum, shor t – and long-term time horiz ons. The bank holding company must update shor t-term\ncash-flow pr ojections daily and must update longer-term cash-flow pr ojections at least monthly .br anch or U.S. agency and the for eign bank' s non-U.S. offices and its affiliates.\n(D)Amounts secur ed b y highly liquid assets. For the purposes of calculating net intr agroup\ncash flow of the U.S. br anches and agencies under this par agraph, the amounts of\nintragroup cash-flow needs and intr agroup cash-flow sour ces that ar e secur ed b y highly\nliquid assets (as defined in paragraph (c)(7) of this section) must be ex cluded fr om the\ncalculation.\n\"\"\"\n3.\n\"\"\"\nFor purposes of this subpart, any company that would be an affiliate of a U.S. branch, agency, or commercial lending company of a foreign bank if such branch, agency, or commercial lending company were a member bank is an affiliate of the branch, agency, or commercial lending company if the company also is: (1) Directly engaged in the United States in any of the following activi-ties: \nVerDate Sep<11>2014 17:04 Jun 02,Any company that the Board determines by regula-tion or order, or that the appropriate Federal banking agency for the mem-ber bank determines by order, to have a relationship with the member bank, or any affiliate of the member bank, such that covered transactions by the member bank with that company may be affected by the relationship to the detriment of the member bank. (b) Affiliate with respect to a mem-ber bank does not include : (1) Subsidiaries. Any company that is a subsidiary of the member bank, un-less the company is: (i) A depository institution;[Reg. V, 72 FR 63758, Nov. 9, 2007, as amended at 74 FR 22642, May 14, 2009; 79 FR 30711, May 29, 2014] PART 223 TRANSACTIONS BE-TWEEN MEMBER BANKS AND THEIR AFFILIATES (REGULATION W) Subpart AIntroduction and Definitions Sec. 223.1 Authority, purpose, and scope. 223.2 What is an affiliate for purposes of sections 23A and 23B and this part?Provision of Regulation W Application (1) 12 CFR 223.2(a)(8)Affiliate includes a fi-nancial subsidiary.Does not apply. Savings association subsidiaries do not meet the statutory definition of financial subsidiary.(a) In general. In some situations in which a member bank purchases an asset from an affiliate, the asset pur-chase qualifies for an exemption under this regulation, but the member banks resulting ownership of the purchased asset also represents a covered trans-action (which may or may not qualify for an exemption under this part).\n\"\"\"\n4.\n\"\"\"\noffices and its U.S. and non-U.S. affiliates; and\n(2)The pr ojected amount of cash-flow sour ces that r esults fr om tr ansactions between\nthe U.S. br anches and agencies and entities other than the for eign bank' s non-U.S.\noffices and its U.S. and non-U.S. affiliates.\n(iv)Net internal str essed cash-flow need calculation \n(A)Gener al.The net internal str essed cash-flow need of the U.S. br

```
 anches and agencies of the\nforeign banking or ganization equals the gr eater of
:\n(1)The gr eatest daily cumulativ e net intr agroup cash-flow need o ver the
first 14 da ys of\nthe 30-da y planning horiz on, as calculated under paragraph (c
)(3)(iv)(B) of this\nsection; and\n(2)Zero.The net external str essed cash-flow
need for a\nU.S. intermediate holding company equals the diff erence between:\n(A)
The pr ojected amount of cash-flow needs that r esults fr om tr ansactions between
 the U.S.\nintermediate holding company and entities that ar e not its affiliates;
 and\n(B)The pr ojected amount of cash-flow sour ces that r esults fr om tr
ansactions between the\nU.S. intermediate holding company and entities that ar e
not its affiliates.br anch or U.S. agency and the for eign bank' s non-U.S.
offices and its affiliates.\n(D)Amounts secur ed b y highly liquid assets. For the
 purposes of calculating net intr agroup\ncash flow of the U.S. br anches and
agencies under this par agraph, the amounts of\nintragroup cash-flow needs and
intr agroup cash-flow sour ces that ar e secur ed b y highly\nliquid assets (as
defined in paragraph (c)(7) of this section) must be ex cluded fr om the\
ncalculation.(C)Net intr agroup cash flow .For any giv en da y of the planning
horiz on, the net intr agroup cash\nflow must equal the diff erence between:\n(1)
The amount of pr ojected cash-flow needs r esulting fr om tr ansactions between a
U.S.\nbranch or U.S.\n\"\"\"\n5.\n\"\"\"\n12, 2002, unless otherwise noted.
Subpart AIntroduction and Definitions  223.1 Authority, purpose, and scope. (a)
Authority. The Board of Governors of the Federal Reserve System (Board) has issued
 this part (Regulation W) under the authority of sections 23A(f) and 23B(e) of the
 Federal Reserve Act (FRA) (12 U.S.C. 371c(f), 371c1(e)) sec-tion 11 of the Home
Owners Loan Act (12 U.S.C.[Reg. V, 72 FR 63758, Nov. 9, 2007, as amended at 74 FR
22642, May 14, 2009; 79 FR 30711, May 29, 2014] PART 223TRANSACTIONS BE-TWEEN
MEMBER BANKS AND THEIR AFFILIATES (REGULATION W) Subpart AIntroduction and
Definitions Sec. 223.1 Authority, purpose, and scope. 223.2 What is an affiliate
for purposes of sections 23A and 23B and this part?(ii) If the Board determines
that a particular transaction is, in substance, a loan or extension of credit to
an affil-iate that is engaged in activities other than those described at 12 U.S.C
. 1467a(c)(2)(F)(i), as defined in 238.54 of Regulation LL (12 CFR 238.54), or the
 Board has other supervisory concerns concerning the transaction, the Board may
inform the savings association that the transaction is prohibited under this
paragraph (c)(1), and require the savings association to divest the loan, unwind
the transaction, or take other appropriate action.12 and the collateral
requirements of 223.14, and is otherwise permitted under this regulation; or (2)
Making reference to such a guar-antee, acceptance, letter of credit, or cross-
affiliate netting arrangement if otherwise required by law.  223.55 What are the
standards under which the Board may grant exemp-tions from the requirements of sec
-tion 23B?12, 2002, as amended at 73 FR 54308, Sept. 19, 2008; 73 FR 55709, Sept.
26, 2008; 74 FR 6226, 6227, Feb.\n\"\"\"\n\nPlease answer the following question
based only on the information above:\nIn producing the comprehensive cash flow
projections for its U.S. operations, will an FBO be required to consolidate cash
flows of entities subject to Regulation W, such as banks, with cash flows of
entities not subject to Regulation W?"
```

This question concerns whether a foreign banking organization must consolidate the cash flows of entities that are subject to Regulation W with those that are not, when producing comprehensive cash-flow projections for its U.S. operations. The gold-standard answer makes the requirement clear. Under Regulation YY section 252.156(d), an FBO must generate comprehensive cash-flow projections for its combined U.S. operations, and these projections must include all entities within that scope, regardless of whether they fall under Regulation W.

The base model, without retrieval, achieves a factual F1 score of about 0.44. Its answer relies mostly on accounting logic and the general principle of consolidated reporting. It notes that an FBO, under GAAP, consolidates the financial statements of its subsidiaries and therefore should include the cash flows of banks subject to Regulation W. Although the answer does not reference Regulation YY directly, it still captures part of the key idea in the gold-standard answer, namely that the cash flows of these entities must be included.

With retrieval, the factual score remains roughly the same at about 0.44. Retrieval brings in direct references to Regulation YY, and the retrieved statements explicitly note that an FBO must produce comprehensive projections for its combined U.S. operations and that these projections must include both entities subject to Regulation W and entities that are not. However, from the perspective of the evaluation metric, the base answer already covers the essential fact that these entities must be included. The retrieval-supported answer restates this using more formal regulatory language but does not introduce many new factual points that would increase coverage, so the score does not improve.

The retrieved source text confirms this pattern. In this case, retrieval mainly improves the clarity, structure, and alignment of the answer with the regulatory text, rather than changing the factual

content. As a result, while retrieval makes the answer more authoritative and closer to official wording, it does not provide a noticeable improvement under a fact-coverage evaluation framework.

