# OpenReview forum: "KnowMT-Bench: Benchmarking Knowledge-Intensive Long-Form Question Answering In Multi-Turn Dialogues"
_ICLR.cc/2026/Conference — ICLR 2026 Conference Withdrawn Submission_

### Official Review · Reviewer_jhxT · 2025-10-21

**Soundness:** 2
**Presentation:** 2
**Contribution:** 2
**Rating:** 4
**Confidence:** 3

**Summary:**

The paper introduces KnowMT-Bench , the first multi-turn long-form question answering (MT-LFQA) benchmark for knowledge-intensive domains (medicine, finance, law).
The research makes three main findings and contributions:
1. Performance Degradation: LLMs exhibit a significant drop in factual capability and information delivery efficiency when transitioning from single-turn to multi-turn dialogue.
2. Root Cause Identification: The decline in factual capability is primarily due to contextual noise generated by the model itself in previous turns, not merely dialogue length.
3. Mitigation Strategy: Retrieval-Augmented Generation (RAG) is an effective strategy that can alleviate and even reverse this factual degradation by grounding each
conversational step with external evidence.

**Strengths:**

**Originality**

- **First-of-its-kind benchmark**: KnowMT-Bench establishes the first benchmark specifically for Multi-Turn Long-Form Question Answering (MT-LFQA) in knowledge-intensive domains (medicine, finance, and law)

- **Novel evaluation setting**: Introduces a highly realistic dynamic evaluation framework where models must generate their own multi-turn conversational history from progressive question sequences

- **Real-world simulation**: Effectively replicates actual consultation scenarios and captures how contextual errors accumulate over conversation turns

**Methodology Quality**

- **Rigorous evaluation pipeline**: Develops a custom, fully automated, two-stage Natural Language Inference (NLI)-based evaluation system that ensures high-fidelity assessment of factual consistency

- **Overcomes prior limitations**: Addresses the shortcomings of superficial metrics used in previous work

- **Human validation**: Extensively validates the system's reliability through human expert annotations, demonstrating excellent agreement at both the answer decomposition and factual judgment stages

**Clarity**

- **Precise diagnosis**: Excels in identifying the specific causes of performance degradation through clear empirical analysis

- **Distinct factor separation**: Empirically distinguishes between:
  - **Dialogue length effects**: Drives verbosity and efficiency loss
  - **Contextual noise effects**: Drives factual accuracy decline

- **Enhanced interpretability**: This clear attribution enables targeted mitigation strategies and improves result interpretability

**Significance**

- **Critical guidance for LLM development**: Provides essential insights for advancing future large language models

- **Reveals evaluation gaps**: Fundamentally demonstrates that single-turn LFQA evaluations are inadequate for assessing real-world performance

- **RAG as powerful solution**: Establishes that Retrieval-Augmented Generation (RAG) is not merely a superficial fix, but a robust solution that can:
  - Mitigate factual degradation caused by conversational noise
  - Potentially reverse accuracy decline
  - Provide a clear pathway toward building more reliable conversational LLMs

**Weaknesses:**

**Limited Dialogue Complexity**

- **Rigid question structure**: Uses logically progressive question sequences (PQS) guided by fixed templates, limiting conversational diversity

- **Excludes critical phenomena**: Does not evaluate important real-world conversational dynamics, including:
  - Spontaneous topic switching
  - User correction of model errors
  - Handling of contradictory contextual information

- **Artificial noise filtering**: Actively screens out common forms of conversational noise

- **Overestimation risk**: May lead to inflated assessments of LLM robustness in less structured, dynamic real-world dialogues

- **Reduced real-world fidelity**: The restricted scope limits how well the benchmark represents actual consultation scenarios

**High Dependence on LLM-Based Evaluation**

- **Single model family dependency**: The entire quality assessment pipeline (both decomposition and judgment stages) relies exclusively on specific Qwen family LLMs

- **Potential evaluation bias**: This narrow dependency raises concerns about systemic biases inherent to the chosen model architecture

- **Limited generalizability testing**: Lacks validation across diverse LLM architectures

- **Need for cross-model validation**: Future work should incorporate alternative high-performing LLMs (e.g., GPT-4o) in the decomposition component to:
  - Verify consistency of factual scores across different models
  - Ensure impartiality of the evaluation framework
  - Confirm robustness of the assessment methodology

**Insufficient Mitigation Analysis**

- **Shallow prompt intervention exploration**: Section 5.3's analysis of prompt-based interventions lacks depth

- **Harsh trade-off identified**: The simple noise-disregard prompt yielded problematic results:
  - Improved factuality
  - Increased hallucination rates

- **Missed opportunities**: Failed to explore more sophisticated prompt engineering techniques, such as:
  - Self-correction mechanisms
  - Guided context selection strategies
  - Multi-stage reasoning prompts
  - Uncertainty acknowledgment frameworks

- **Lack of actionable guidance**: Does not provide practical, implementable solutions for practitioners seeking to mitigate multi-turn degradation

**Questions:**

1. Dialogue Realism and Scope: The benchmark uses pre-defined logically progressive question sequences (PQS) to ensure intent preservation. Do you believe this highly constrained format adequately captures real-world conversational challenges like user topic drift or the need for error correction, or does it potentially overstate the model's factual robustness?
2. For future work, consider including a Challenge Subset of dialogues that explicitly test robustness against user correction of model errors and contradictory contextual information.
3. Evaluation Pipeline Robustness (Decomposition Bias): The entire evaluation's quality hinges on the LLM Decomposer (Qwen2.5-32B-Instruct); Did you investigate the cross-architectural robustness of the decomposition step? Would using a different strong model (e.g., GPT-4o) as the decomposer yield the same atomic facts and resulting metrics,
or could there be an architectural bias?

---

> ### Author Response · Authors · 2025-11-21
> **Reply to Reviewer jhxT (Q1-Part1)**
>
> ### Q1: Dialogue Realism and Scope: The benchmark uses pre-defined logically progressive question sequences (PQS) to ensure intent preservation. Do you believe this highly constrained format adequately captures real-world conversational challenges like user topic drift or the need for error correction, or does it potentially overstate the model's factual robustness? (Also for Weakness 1: Limited Dialogue Complexity)
>
> We thank the reviewer for raising the concern about **Limited Dialogue Complexity and Reduced real-world fidelity**. Below we further clarify the design goals and coverage of **KnowMT-Bench** in terms of multi-turn construction and evaluation.
>
> First, **KnowMT-Bench follows a dynamic history generation evaluation protocol**: in each turn, the model must first generate its own answer, and these self-generated answers are then used as the dialogue history for subsequent turns. We fully acknowledge that an alternative setup—pre-defining the dialogue history and explicitly designing phenomena such as “error–correction” and contradiction handling—also has clear value. The two approaches emphasize different but complementary aspects: pre-defined histories make it easier to systematically cover correction and contradiction phenomena; dynamic histories more faithfully reflect real deployment, where the dialogue history is **typically generated by the same model itself and carries its characteristic noise**. More importantly, our experiments show that **if we directly use dialogue histories generated by a strong model as evaluation context, we systematically overestimate the true capability of the model under evaluation**. This indicates that dynamic evaluation provides a more conservative and realistic characterization of factuality and robustness under actual usage conditions, rather than artificially simplifying the dialogue difficulty.
>
> Second, regarding the reviewer’s point about **spontaneous topic switching / topic drift**, we agree that in real-world scenarios users do sometimes place **multiple, loosely related topics into a single dialogue thread** (e.g., switching consultation topics within the same session). Although the primary goal of KnowMT-Bench is to evaluate MT-LFQA in **domain-focused, progressive expert consultation** settings (where the main intent is usually stable), we also conduct a controlled **“topic shift”** experiment to probe the impact of real usage habits. Specifically, we prepend an **unrelated question (a randomly selected question from other domain)** before each PQS sequence to simulate topic drift. The results are as follows (Table R-Shift):
>
> | Model(Qwen-2.5-7B) | Mode   | $S_f$ (%)   | $S_h$ (%)   | $D_f$     | $D_h$    |
> | ------------------ | ------ | ----------- | ----------- | --------- | -------- |
> | original           | single | 19.67       | 3.57        | 366       | 589      |
> | topic drift        | single | 19.65       | 3.49        | 367       | 590      |
> | original           | multi  | 17.15       | 3.77        | 547       | 847      |
> | topic drift        | multi  | 20.51       | 4.88        | 537       | 833      |
>
> We observe that **topic drift does not significantly reduce factual ability** ($S_f$ remains comparable to the original setting), but it **increases hallucinations** (higher $S_h$) in the multi-turn scenario. This suggests that irrelevant context primarily raises hallucination risk, rather than directly harming factuality.
>
> Additionally, we observe that introducing topic shifts leaves single-turn performance essentially unchanged: both $S_f$ and $S_h$ remain almost the same as in the original setting, suggesting that the unrelated prefix has at most a marginal effect on the model’s focus in single-turn consultations. In contrast, the multi-turn case is more nuanced: topic drift raises both factual outputs and hallucinations simultaneously (higher $S_f$ and higher $S_h$), while $D_f$ and $D_h$ stay at a similar scale. This suggests that the unrelated prefix mainly alters the model’s generation pattern, leading to a higher proportion of turns that contain correct facts but also a higher proportion that contain hallucinations, rather than yielding a one-directional improvement or degradation. Therefore, topic drift does not introduce a systematic optimistic bias; instead, it exposes a realistic trade-off between factual bonus and hallucination risk, supporting that our evaluation does **not overestimate** models’ robustness in multi-turn knowledge-intensive dialogues.

---

> ### Author Response · Authors · 2025-11-21
> **Reply to Reviewer jhxT (Q1-Part2)**
>
> ......
>
> Finally, this observation is consistent with the conclusions from our other robustness analyses in the experiment in part 4 of the reply to reviewer RzT5's question 5: **the most powerful factor affecting multi-turn performance is “relevant incorrect history”, rather than irrelevant noise or irrelevant incorrect history**. Therefore, it is reasonable that topic drift does not substantially decrease model factual performance.
>
> In summary, KnowMT-Bench adopts pre-defined PQS to isolate **progressive, intent-preserving expert consultations**, which are a common form of real-world knowledge-intensive dialogues, but we do not claim to cover all open-ended conversational phenomena. Importantly, because our histories are **dynamically self-generated**, the benchmark does not rely on “clean” or oracle contexts; instead it captures robustness under realistic self-noise, and our strong-history control further shows that static/oracle histories would **overestimate** factual robustness. The added topic-drift results also indicate that even when realistic off-topic prefixes occur, they do not create a systematic optimistic bias, but reveal a nuanced fact–hallucination trade-off. We will clarify this scope–realism trade-off in the paper.

---

> > ### Author Response · Authors · 2025-11-21
> > **Reply to Reviewer jhxT (Q2)**
> >
> > ### Q2: For future work, consider including a Challenge Subset of dialogues that explicitly test robustness against user correction of model errors and contradictory contextual information.
> >
> > We agree with the reviewer that explicitly covering **user correction and contradiction-handling** phenomena is crucial for assessing real-world robustness. In this work, we primarily focus on progressive, knowledge-intensive consultations under **dynamically self-generated histories**, with the goal of characterizing how factuality degrades in such settings.
> >
> > As future work, we plan to introduce a dedicated **challenge subset** based on **pre-defined dialogue histories**: earlier turns will deliberately inject model errors or contradictions that are semantically related to the target question, and later turns will contain explicit user corrections. This will enable a systematic evaluation of models’ abilities to handle correction and contradictions in multi-turn, long-form, knowledge-intens dialogues.

---

> > > ### Author Response · Authors · 2025-11-21
> > > **Reply to Reviewer jhxT (Q3-Part1)**
> > >
> > > ### Q3: Evaluation Pipeline Robustness (Decomposition Bias): The entire evaluation's quality hinges on the LLM Decomposer (Qwen2.5-32B-Instruct); Did you investigate the cross-architectural robustness of the decomposition step? Would using a different strong model (e.g., GPT-4o) as the decomposer yield the same atomic facts and resulting metrics, or could there be an architectural bias? (Also for Weakness 2: High Dependence on LLM-Based Evaluation)
> > >
> > > We thank the reviewer for the valuable comment regarding the potential bias introduced by relying solely on Qwen-based models in our evaluation pipeline. We fully agree that **cross-model validation** is essential to ensure that the evaluation results are generalizable and fair.
> > >
> > > To address this concern, we conduct cross-model validation by **replacing Qwen with GPT-4o** at two different stages of the pipeline, the **Evaluator** and the **Decomposer**, and then re-evaluating four representative LLMs (GPT-4o, DeepSeek-V3, Llama-3.3-70B, and Qwen-2.5-72B) under both **single-turn** and **multi-turn** settings. See the results in **Table R1** and **Table R2** below:
> > >
> > > Our experiments show that the evaluation framework of **KnowMT-Bench** is highly robust:
> > >
> > > Overall, **all evaluation metrics show highly consistent behavior across model choices**. For clarity, we use the factual score (S_f) as a representative example in the analysis below; **other metrics (e.g., ($R_f, P_f, S_h$)) exhibit the same pattern of agreement**.
> > >
> > > #### 1. Robustness of the Evaluator
> > >
> > > We first replace the original Qwen-2.5-14B Evaluator with **GPT-4o**, while keeping the Decomposer fixed, and compare the resulting factual F1 scores ($S_f$).
> > >
> > > We observe **strong statistical consistency** between the two Evaluators:
> > >
> > > - **High correlation**
> > >   - Pearson correlation $r = 0.984$: indicates an strong linear relationship between the two sets of scores.
> > >   - Spearman rank correlation $\rho = 0.905$: shows that the **relative ranking** of models is almost identical under the two Evaluators.
> > >
> > > - **Stable ranking across settings**
> > >   As shown in Table R1, both Evaluators (Qwen-based and GPT-4o–based) consistently agree that GPT-4o is the top-tier model, and they preserve the overall separation between the proprietary model and the open-source models in both single-turn and multi-turn evaluation. The rankings of the open-source models only exhibit minor permutations, without changing our qualitative conclusions.
> > >
> > > #### 2. Stability of the Decomposer
> > >
> > > We further test the robustness of the **Atomic Fact Decomposition** stage by replacing the original Qwen-2.5-32B Decomposer with **GPT-4o**, while keeping the Evaluator fixed.
> > >
> > > Again, we find **near-perfect agreement**:
> > >
> > > - **High correlation**
> > >   - Pearson correlation $r = 0.996$: this extremely high value indicates that the atomic facts extracted by the Qwen-based Decomposer are highly aligned with those extracted by GPT-4o, and that the choice of Decomposer explains almost all variance in the final scores.
> > >   - Spearman rank correlation $\rho = 0.905$: confirms that changing the Decomposer does not significant alter the **relative comparison** between models.
> > >
> > > #### 3. Robustness of Core Findings
> > >
> > > Our main conclusion, that **models’ factual ability degrades in multi-turn dialogue**, remains stable under all validation configurations.
> > >
> > > As shown in Table R2, regardless of whether we replace the Evaluator, the Decomposer, or both, **all models** (including GPT-4o) exhibit a **consistent drop** in factual metrics $(R_f, P_f, S_f)$ when moving from the single-turn to the multi-turn setting.
> > >
> > > In summary, the evaluation framework of **KnowMT-Bench** is **not tied to a specific architecture**. While we chose the Qwen family primarily for **reproducibility (open weights)** and **computational efficiency**, our cross-model validation demonstrates that using a stronger proprietary model (GPT-4o) in place of Qwen leads to **the same qualitative conclusions**.
> > >
> > > We will include these detailed comparative results in the appendix of the revised paper to make this robustness more explicit.

---

> > > > ### Author Response · Authors · 2025-11-21
> > > > **Reply to Reviewer jhxT (Q3-Part2)**
> > > >
> > > > Table R1: Model Ranking Consistency across Evaluators and Decomposer Variants
> > > >
> > > > | Original Rank | Model (Mode)         | $S_f$ | GPT-4o Eval(Rank) | GPT-4o Decomp(Rank) |
> > > > |------|-------------------------|------------------------|-----------------------------|--------------------------------|
> > > > | 1    | GPT-4o (Single)         | 28.77%                 | 38.06% (1)                  | 29.54% (1)                     |
> > > > | 2    | GPT-4o (Multi)          | 27.63%                 | 37.86% (2)                  | 29.24% (2)                     |
> > > > | 3    | Llama-3.3-70B (Single)  | 24.02%                 | 32.95% (4)                  | 24.41% (5)
> > > > | 4    | Qwen-2.5-72B (Single)   | 23.97%                 | 32.37% (5)                  | 24.47% (4)                     |
> > > > | 5    | DeepSeek-V3 (Single)    | 23.80%                 | 33.82% (3)                  | 24.51% (3)                     |              |
> > > > | 6    | Llama-3.3-70B (Multi)   | 21.16%                 | 29.26% (6)                  | 21.50% (6)                     |
> > > > | 7    | Qwen-2.5-72B (Multi)    | 20.20%                 | 28.54% (8)                  | 20.97% (7)                     |
> > > > | 8    | DeepSeek-V3 (Multi)     | 19.38%                 | 29.19% (7)                  | 19.54% (8)                     |
> > > >
> > > >
> > > > Table R2: Cross-Model Validation – Detailed Metrics for Evaluator and Decomposer Variants
> > > >
> > > > | Model          | Setting | Configuration        | $S_f$ (%) | $S_h$ (%) | $D_f$  | $D_h$  |
> > > > |---------------|---------|----------------------|--------|--------|-----|-----|
> > > > | GPT-4o        | Single  | Original             | 28.77  | 2.26   | 226 | 506 |
> > > > | GPT-4o        | Single  | w/ GPT-4o Eval       | 38.06  | 3.36   | 179 | 434 |
> > > > | GPT-4o        | Single  | w/ GPT-4o Decomp     | 29.54  | 2.55   | 226 | 507 |
> > > > | GPT-4o        | Multi   | Original             | 27.63  | 2.48   | 254 | 555 |
> > > > | GPT-4o        | Multi   | w/ GPT-4o Eval       | 37.86  | 3.11   | 214 | 532 |
> > > > | GPT-4o        | Multi   | w/ GPT-4o Decomp     | 29.24  | 2.26   | 253 | 495 |
> > > > | DeepSeek-V3   | Single  | Original             | 23.80   | 3.21   | 343 | 740 |
> > > > | DeepSeek-V3   | Single  | w/ GPT-4o Eval       | 33.82  | 4.19   | 285 | 722 |
> > > > | DeepSeek-V3   | Single  | w/ GPT-4o Decomp     | 24.51  | 3.32   | 340 | 739 |
> > > > | DeepSeek-V3   | Multi   | Original             | 19.38  | 3.40   | 444 | 888 |
> > > > | DeepSeek-V3   | Multi   | w/ GPT-4o Eval       | 29.19  | 4.32   | 373 | 837 |
> > > > | DeepSeek-V3   | Multi   | w/ GPT-4o Decomp     | 19.54  | 3.40   | 445 | 877 |
> > > > | Llama-3.3-70B | Single  | Original             | 24.02  | 3.49   | 324 | 628 |
> > > > | Llama-3.3-70B | Single  | w/ GPT-4o Eval       | 32.95  | 5.12   | 272 | 628 |
> > > > | Llama-3.3-70B | Single  | w/ GPT-4o Decomp     | 24.41  | 3.58   | 326 | 653 |
> > > > | Llama-3.3-70B | Multi   | Original             | 21.16  | 4.25   | 437 | 760 |
> > > > | Llama-3.3-70B | Multi   | w/ GPT-4o Eval       | 29.26  | 6.00   | 383 | 732 |
> > > > | Llama-3.3-70B | Multi   | w/ GPT-4o Decomp     | 21.50  | 4.16   | 435 | 775 |
> > > > | Qwen-2.5-72B  | Single  | Original             | 23.97  | 2.44    | 358 | 821 |
> > > > | Qwen-2.5-72B  | Single  | w/ GPT-4o Eval       | 32.37  | 3.92   | 334 | 791 |
> > > > | Qwen-2.5-72B  | Single  | w/ GPT-4o Decomp     | 24.47  | 2.49   | 366 | 822 |
> > > > | Qwen-2.5-72B  | Multi   | Original             | 20.20  | 2.77    | 484 | 900 |
> > > > | Qwen-2.5-72B  | Multi   | w/ GPT-4o Eval       | 28.54  | 4.05   | 434 | 872 |
> > > > | Qwen-2.5-72B  | Multi   | w/ GPT-4o Decomp     | 20.97  | 2.86   | 484 | 900 |

---

> > > > > ### Author Response · Authors · 2025-11-21
> > > > > **Reply to Reviewer jhxT (Q4)**
> > > > >
> > > > > ### Q4: Insufficient Mitigation Analysis: Section 5.3’s prompt-based intervention exploration is shallow, with a harsh trade-off where a simple noise-disregard prompt improves factuality but increases hallucination rates; why were more sophisticated prompt engineering techniques (self-correction mechanisms, guided context selection strategies, multi-stage reasoning prompts, uncertainty acknowledgment frameworks) not explored, and how can the paper provide actionable, implementable guidance for mitigating multi-turn degradation? (Also for Weakness 3: Insufficient Mitigation Analysis)
> > > > >
> > > > > Following the reviewer’s concern that **Section 5.3 provides only shallow mitigation analysis and lacks actionable guidance**, we additionally evaluate three more sophisticated intervention strategies, and report their multi-turn effects together with the original setting:
> > > > >
> > > > > * **Summarization (Sum)**: before each turn, the LLM produces a global summary of the full dialogue history, and then answers based on this condensed summary.
> > > > > * **Context Selection (Select)**: before each turn, the LLM selects the most relevant fragments from the full history w.r.t. the current question, and answers using only these selected contexts.
> > > > > * **Iterative Self-Refinement (Self-Refine)**: following *Self-Refine*[a], the model first generates a draft, then produces self-feedback and iteratively revises the answer.
> > > > >
> > > > > All summarization / history selection / self-refinement steps are performed **by the same model**, without extra annotations or separate sub-models. We run new experiments with Qwen2.5-32B, and report the results below.
> > > > >
> > > > > **Table R1: Performance of advanced mitigation strategies under multi-turn settings**
> > > > >
> > > > > | Strategy | Turn | $S_f$ (%) | $S_h$ (%) | $D_f$ | $D_h$ |
> > > > > | - | - | -: | -: | -: | -: |
> > > > > | Original | Single | 22.50 | 2.57 | 367 | 685 |
> > > > > | Original | Multi  | 21.62 | 3.02 | 435 | 746 |
> > > > > | Add_Prompt | Multi  | 21.89 | 3.05 | 417 | 764 |
> > > > > | Select | Multi  | 24.10 | 3.19 | 319 | 590 |
> > > > > | Sum | Multi  | 24.05 | 3.28 | 259 | 419 |
> > > > > | Self-Refine | Multi  |  19.84 | 2.90| 525 | 899 |
> > > > >
> > > > > **Overall, the gains mainly manifest in $S_f$, while the impact on $S_h$ is relatively small.**
> > > > > Across all three advanced strategies, hallucination levels remain within a narrow band ($S_h$ fluctuates only between 2.9% and 3.3%), whereas factuality $S_f$ varies substantially. This indicates that mitigating multi-turn degradation is primarily about **recovering factuality**, rather than simply suppressing hallucinations.
> > > > >
> > > > > 1. **Summarization: shortening history alone already mitigates degradation and yields the best efficiency**
> > > > >    Summarization raises multi-turn $S_f$ to **24.05%** (vs. 21.62% original), and reduces $D_f$ to **259**. This shows that **model-driven compression of history** effectively restores factuality while cutting token cost, making it a strong choice under efficiency constraints.
> > > > >
> > > > > 2. **Selection: the strongest factuality gain via explicit history filtering**
> > > > >    Selection achieves the highest multi-turn factuality (**$S_f$ = 24.10%**) and lowers $D_f$ to 319. By enforcing an explicit “select-then-answer” mechanism, the model discards irrelevant/perturbative history segments before generation, producing the **most robust factual improvement** among all strategies.
> > > > >
> > > > > 3. **Self-Refine: fewer hallucinations but reduced factuality**\
> > > > >    Self-Refine yields the lowest hallucination score ($S_h$ = 2.90%), but also the highest hallucination density ($D_h$ = 899) and the largest number of factual segments ($D_f$ = 525), indicating that hallucinations occur in fewer turns but those answers are longer and contain more atomic facts overall. However, factuality drops to 19.84%, suggesting that in knowledge-intensive, non-RAG settings, self-refinement can trade a lower hallucination rate for weaker factual coverage and should be used with caution.
> > > > >
> > > > > #### Takeaways for Future Knowledge-Intensive Long-Form Dialogues
> > > > >
> > > > > These results provide direct implications for future models/systems in knowledge-intensive, long-form, multi-turn consultations:
> > > > >
> > > > > * **The key to mitigation is improving $S_f$.** Even a simple model-driven history summarization step can recover factuality while reducing cost.
> > > > > * **Explicit “selection-based history usage” is most effective..** Instead of relying on prompts to *hope* the model ignores noise, enforcing a model-driven context selection module yields the strongest robustness gains.
> > > > > * **Self-refinement without external evidence is not reliably beneficial.** In professional knowledge-intensive dialogues, Self-Refine–style mechanisms may sacrifice factuality, unless combined with external knowledge (e.g., RAG/tools) or more controlled refinement.
> > > > >
> > > > > We will revise Section 5.3 accordingly and incorporate these findings as concrete, implementable mitigation guidance in the paper.
> > > > >
> > > > > [a] Madaan, Aman, et al. "Self-refine: Iterative refinement with self-feedback." Advances in Neural Information Processing Systems 36 (2023): 46534-46594

---

### Official Review · Reviewer_J9J3 · 2025-11-02

**Soundness:** 3
**Presentation:** 3
**Contribution:** 3
**Rating:** 8
**Confidence:** 4

**Summary:**

The paper introduces KnowMT-Bench, the first benchmark specifically designed to evaluate multi-turn long-form question answering (MT-LFQA) in knowledge-intensive domains (medicine, finance, and law).

**Strengths:**

1. Novelty. This work explicitly targets multi-turn, knowledge-intensive long-form QA, which prior benchmarks only partially address.
2. Robust evaluation design. The NLI-based, human-validated pipeline is fine-grained into two parts and both automated, addressing weaknesses of LLM-as-a-judge methods.
3. Comprehensive ablation studies by dialogue length, domain, finetuning, and retrieval strategy.
4. Brings empirical insights by identifying contextual noise as the primary failure mode and empirically validating RAG as a strong mitigation.
5. Covers three specialized, high-stakes areas, showing generalization.

**Weaknesses:**

1. Although partially human-validated, using Qwen models as both decomposer and NLI judge + also evaluating Qwen models on the benchmark introduces potential bias.
2. Only RAG is tested, and other context-filtering or history-truncation approaches (e.g., summarization, re-encoding) are not.
3. Potential domain imbalance: Most data (≈ 579/801) come from finance; medical and legal subsets are smaller.
4. Bias in synthetic dialogs: Multi-turn sequences are partly LLM-generated, which could bake in stylistic artifacts or coherence patterns not reflective of real users.
5. No comparison to existing multi-turn factual QA datasets (e.g., ConvFinQA, Doc2Dial) under unified metrics to show cross-benchmark consistency.

**Questions:**

1. “Factual F1” and “Hallucination F1” are intuitive but may double-count correlated errors. One non-factual claim will both reduce S_f (by lowering P_f) and raise S_h (by increasing P_fc), so you are double-penalizing models with concise answer (less coverage of ground truths) but one error claim. On the other hand, models with verbose answers will give lots of facts where some are true and some are false, and will inflate both S_f and S_h. Evaluating information factuality vs. verbosity is difficult when both scores co-vary due to shared entailment/contradiction judgements. A future refinement might include separating coverage, error density, and truth-to-length ratio to better distinguish these behaviors.
2. When we evaluate one model, how should we holistically interpret the three dimensional metrics you introduced -- factuality, information delivery efficiency and hallucination metrics? What composition of these metrics indicate a good model?
3. What exact prompt was used in prompt-based mitigation (Table 4)? Why did hallucination rise here?

---

> ### Author Response · Authors · 2025-11-21
> **Reply to Reviewer J9J3 (Q1)**
>
> ### Q1: Although partially human-validated, using Qwen models as both decomposer and NLI judge + also evaluating Qwen models on the benchmark introduces potential bias.
>
> We fully agree with the reviewer that relying on a single model family in both the **decomposition** and **NLI judgment** stages may introduce systematic bias (e.g., favoring certain backbones or penalizing others). To address this concern, we conduct targeted bias checks for **both** components. The evaluator-bias validation was already reported in Appendix D.3 of the paper; here we (i) briefly restate that result for completeness, and (ii) add a new decomposer-bias validation following the same statistical design.
>
> **On potential bias of the Evaluator.**
> Following the analysis in Appendix D.3 / Table 7, we test whether the Qwen-based NLI Evaluator produces systematically different factual judgments across model families. Using ACC computed by the automatic NLI judge on the outputs of four representative LLMs (DeepSeek-V3, Llama-3-8B, GPT-4o, and Qwen-2.5-14B), a Kruskal–Wallis test yields
> $$
> H = 1.9552,\quad p = 5.82 \times 10^{-1},
> $$
> indicating **no significant overall difference** among models. Pairwise Mann–Whitney U tests further show that all comparisons have p-values well above both 0.01 and 0.05 (Table below), so we do not detect any statistically significant evaluator favoritism toward any backbone.
>
> **Evaluator pairwise Mann–Whitney U p-values**
>
> |                  | DeepSeek-V3 | GPT-4o | Llama-3-8B | Qwen-2.5-14B |
> | ---------------- | ----------: | -----: | ---------: | -----------: |
> | **DeepSeek-V3**  |           – | 0.4449 |     0.7612 |       0.1830 |
> | **GPT-4o**       |      0.4449 |      – |     0.6739 |       0.6020 |
> | **Llama-3-8B**   |      0.7612 | 0.6739 |          – |       0.3473 |
> | **Qwen-2.5-14B** |      0.1830 | 0.6020 |     0.3473 |            – |
>
>
> **On potential bias of the Decomposer.**
> Analogous to the evaluator check, we test whether the Qwen-based Decomposer systematically favors or penalizes some models when extracting atomic facts from their self-generated dialogue histories. For each model, we compute decomposition count error against human gold using sMAPE, then perform (1) a Kruskal–Wallis test across models and (2) pairwise Mann–Whitney U tests. The Kruskal–Wallis test yields
> $$
> H = 6.1147,\quad p = 1.062 \times 10^{-1},
> $$
> showing **no significant global difference** across model families. Pairwise tests again indicate that **no pair is significant at the stricter 0.01 level**, and only one pair reaches 0.05 with a small effect size. This suggests that any between-model variation in decomposition error is minor, and we find **no strong evidence that the Decomposer systematically favors or penalizes specific model families**.
>
> **Decomposer pairwise Mann–Whitney U p-values**
>
> |                  | DeepSeek-V3 | GPT-4o | Llama-3-8B | Qwen-2.5-14B |
> | ---------------- | ----------: | -----: | ---------: | -----------: |
> | **DeepSeek-V3**  |           – | 0.2373 |     0.4773 |       0.1039 |
> | **GPT-4o**       |      0.2373 |      – |     0.0858 |       0.8405 |
> | **Llama-3-8B**   |      0.4773 | 0.0858 |          – |   **0.0304** |
> | **Qwen-2.5-14B** |      0.1039 | 0.8405 | **0.0304** |            – |
>
> Taken together, the Kruskal–Wallis tests show no significant global differences for either stage, and the pairwise tests reveal no consistent pattern of favoritism. Therefore, both the Decomposer and the Evaluator behave **largely model-agnostically**, and our main conclusions are unlikely to be artifacts of Qwen-family bias.

---

> ### Author Response · Authors · 2025-11-21
> **Reply to Reviewer J9J3 (Q2)**
>
> ### Q2: Only RAG is tested, and other context-filtering or history-truncation approaches (e.g., summarization, re-encoding) are not.
>
> We additionally evaluate three more sophisticated intervention strategies, and report their multi-turn effects together with the original setting in **Table 1**
>
> * **Summarization (Sum)**: before generating each turn, the same LLM first produces a global summary of the full dialogue history, and the model then answers based on this condensed summary.
> * **Context Selection (Select)**: before answering each turn, the LLM automatically selects the most relevant fragments from the full history w.r.t. the current question, and answers using only these selected contexts.
> * **Iterative Self-Refinement (Self-Refine)**: following *Self-Refine*[a], the model first generates an initial draft, then produces self-feedback and iteratively revises the answer.
>
> All summarization / history selection / self-refinement steps are performed **by the same model itself**, without extra annotations or separate sub-models. We run new experiments with Qwen2.5-32B, and report the results below.
>
> **Table 1: Performance of advanced mitigation strategies under multi-turn settings**
>
> | Strategy      | Mode        | $S_f$ (%) | $S_h$ (%) |   $D_f$ |   $D_h$ |
> | ------------- | ----------- | --------: | --------: | ------: | ------: |
> | Original      | Single-turn |     22.70 |      2.73 |     366 |     684 |
> | Original      | Multi-turn  |     21.86 |      3.06 |     434 |     744 |
> | Add_Prompt    | Multi-turn  |     21.89 |      3.05 |     417 |     764 |
> | Select        | Multi-turn  |     24.10 |      3.19 |     319 |     590 |
> | Sum           | Multi-turn  |     24.05 |      3.28 |     259 |     419 |
> | Self-Refine    | Multi-turn  |     19.84 |      2.90 |     525 |     899 |
>
> **Overall, the gains mainly manifest in $S_f$, while the impact on $S_h$ is relatively small.**
> Across all three advanced strategies, hallucination levels remain within a narrow band ($S_h$ fluctuates only between 2.9% and 3.3%), whereas factuality $S_f$ varies substantially. This indicates that mitigating multi-turn degradation is primarily about **recovering factuality**, rather than simply suppressing hallucinations.
>
> 1. **Summarization: shortening history alone already mitigates degradation and yields the best efficiency**
>    Summarization raises multi-turn $S_f$ to **24.05%** (vs. 21.86% original), and reduces $D_f$ to **259**. This shows that **model-driven compression of history** effectively restores factuality while cutting token cost, making it a strong choice under efficiency constraints.
>
> 2. **Selection: the strongest factuality gain via explicit history filtering**
>    Selection achieves the highest multi-turn factuality (**$S_f$ = 24.10%**) and lowers $D_f$ to 319. By enforcing an explicit “select-then-answer” mechanism, the model discards irrelevant/perturbative history segments before generation, producing the **most robust factual improvement** among all strategies.
>
> 3. **Self-Refine: fewer hallucinations but reduced factuality**\
>    Self-Refine yields the lowest hallucination score ($S_h$ = 2.90%) and the sparsest hallucination density ($D_h$ = 899), indicating more cautious outputs. However, factuality drops to **19.84%**, suggesting that in knowledge-intensive, non-RAG settings, self-refinement can **hurt overall factuality** and should be used carefully.
>
> These results provide direct implications for future models/systems in knowledge-intensive, long-form, multi-turn consultations:
>
> * **The key to mitigation is improving $S_f$.** Even a simple model-driven history summarization step can recover factuality while reducing cost.
> * **Explicit “selection-based history usage” is the most effective direction.** Instead of relying on prompts to *hope* the model ignores noise, enforcing a model-driven context selection module yields the strongest robustness gains.
> * **Self-refinement without external evidence is not reliably beneficial.** In professional knowledge-intensive dialogues, Self-Refine–style mechanisms may sacrifice factuality, unless combined with external knowledge (e.g., RAG/tools) or more controlled refinement.
>
> We will revise Section 5.3 accordingly and incorporate these findings as concrete, implementable mitigation guidance in the paper.
>
> [a] Madaan, Aman, et al. "Self-refine: Iterative refinement with self-feedback." Advances in Neural Information Processing Systems 36 (2023): 46534-46594

---

> ### Author Response · Authors · 2025-11-21
> **Reply to Reviewer J9J3 (Q3, Q4, Q5)**
>
> ### Q3: Potential domain imbalance: Most data (≈ 579/801) come from finance; medical and legal subsets are smaller. (As the same as Q2 from Reviewer 7n4w and Q3 from Reviewer m7UK)
>
> We acknowledge the reviewer’s observation that, at the coarse-grained label level, the proportion of items tagged as “Finance” is relatively high. However, we would like to clarify that, from a capability perspective, the samples labeled as “Finance” do not belong to a single domain; they actually contain two essentially different types of tasks:
>
> - **Financial–legal / regulatory QA (~300 instances).**
>   These questions revolve around specific financial laws, regulatory rules, and compliance requirements, such as the applicability conditions of EU regulations or the calculation of regulatory thresholds in cross-border mergers and acquisitions. This subset mainly tests a model’s ability to understand legal provisions and perform legal reasoning, although the legal questions happen to be situated in financial scenarios. From the capability perspective, it is therefore closer to “law QA” than to traditional everyday financial knowledge.
>   For example, a financial–legal question is:
>   *“How should the turnover threshold of Article 20(3)(a) of Regulation (EU) 2022/2560 be calculated for concentrations consisting in the creation of a joint venture?”*
>
> - **Everyday finance consultation QA (~300 instances).**
>   This subset focuses on financial questions that ordinary users care about, such as whether bank accounts can still be used normally, explanations of product and service rules, and general interpretations of financial policies. Compared with the first subset, this mainly tests a model’s everyday financial knowledge.
>   For example, an everyday finance question is:
>   *“If Citizens Bank has closed, can I still use my deposit accounts normally?”*
>
> Therefore, although both types are statistically tagged under the “Finance” domain, in the model capability space they correspond to two different direction, legal reasoning ability and everyday financial knowledge, rather than a single homogeneous domain.
>
> ### Q4: Bias in synthetic dialogs: Multi-turn sequences are partly LLM-generated, which could bake in stylistic artifacts or coherence patterns not reflective of real users.
>
> To address the concern that our dialogue patterns might be “unnatural”, we conduct an additional empirical study **Complex Scenario Suitability Study** from StructFlowBench [a]. Specifically, we follow the setup of StructFlowBench, using **GPT-4o** as an automatic evaluator to rate 50 randomly sampled dialogues from each dataset on three dimensions (1–5 scale):
>
> - **Logical Coherence** – whether the dialogue is logically consistent without abrupt or unreasonable jumps.
> - **Goal Clarity** – whether the task goal is clearly conveyed and the intents of both user and system are transparent.
> - **Transition Naturalness** – whether transitions between turns are smooth and natural, without awkward or forced shifts.
>
> In addition, we define a **Confusion Factor (CF)** as a composite indicator: it is the proportion of dialogue samples whose *average* score across the three dimensions is **≥ 4**. CF thus serves as a proxy for how well the dialogue distribution aligns with real-world complex interactions. The experiment results are as follow:
>
> | Dataset                | Logical Coherence | Goal Clarity | Transition Naturalness | Overall Mean | Confusion Factor (CF) |
> |---|---|---|---|---|---|
> | **Ours (KnowMT-Bench)** | 4.72              | 4.46         | 4.12                   | 4.43         | 0.78                  |
> | **StructFlowBench** [a] | 4.78          | 4.46         | 4.18                   | 4.47         | 0.82                  |
> | **MT-Bench-101** [b]  | 4.50          | 4.52         | 4.50                   | 4.51         | 0.74                  |
>
> The data show that our benchmark (**overall mean 4.43, CF = 0.78**) is on par with two high-quality multi-turn dialogue benchmarks—StructFlowBench (**4.47, 0.82**) and MT-Bench-101 (**4.51, 0.74**). Notably, our **CF (0.78)** is even higher than that of MT-Bench-101 (0.74). These results provide strong evidence that the multi-turn dialogues generated from our question sequences are **reasonably natural** and well aligned with real-world complex interaction patterns.
>
> [a] Li, Jinnan, et al. "Structflowbench: A structured flow benchmark for multi-turn instruction following." ACL-Findings 2025.
>
> [b] Bai, Ge, et al. "Mt-bench-101: A fine-grained benchmark for evaluating large language models in multi-turn dialogues." ACL 2024.
>
> ### Q5: No comparison to existing multi-turn factual QA datasets (e.g., ConvFinQA, Doc2Dial) under unified metrics to show cross-benchmark consistency.
>
> We appreciate the reviewer’s suggestion. In future work, we will extend our evaluation metrics to more existing multi-turn factual QA benchmarks (e.g., ConvFinQA, Doc2Dial) for unified assessment and to verify cross-benchmark consistency.

---

> ### Author Response · Authors · 2025-11-21
> **Reply to Reviewer J9J3 (Q6, Q7)**
>
> ### Q6: “Factual F1” and “Hallucination F1” are intuitive but may double-count correlated errors. One non-factual claim will both reduce S_f (by lowering P_f) and raise S_h (by increasing P_fc), so you are double-penalizing models with concise answer (less coverage of ground truths) but one error claim. On the other hand, models with verbose answers will give lots of facts where some are true and some are false, and will inflate both S_f and S_h. Evaluating information factuality vs. verbosity is difficult when both scores co-vary due to shared entailment/contradiction judgements. A future refinement might include separating coverage, error density, and truth-to-length ratio to better distinguish these behaviors.
>
> We thank the reviewer for the intuition regarding metric coupling. We respectfully clarify that our framework introduces a **Neutral** category (irrelevant/unsupported info) which mathematically decouples these dimensions, as detailed in Appendix G.3.
>
> - **Mathematical Independence of $S_f$ and $S_h$:**
>     We are not in a binary (True/False) system. In our framework, $\mathbf{P_f} + \mathbf{P_{fc}} \le 1$ (not $=1$). A model can generate "fluff" (Neutral) that lowers Factuality ($S_f$) **without** increasing Hallucination ($S_h$).
>     For example, a response with "5 correct + 5 neutral statements" has the same $S_f$ as "5 correct + 5 contradictory statements," but a completely different $S_h$ (0 vs. 0.5). This proves they are not functionally locked. The observed negative correlation is thus an empirical finding about model behavior, not a mathematical artifact.
>
> - **"Double-Penalty" is Feature, Not Bug:**
>     We argue that a **Contradiction** warrants a dual penalty compared to a **Neutral** error. A Neutral error only fails **Utility** ($S_f$), whereas a Contradiction fails both **Utility** ($S_f$) and **Safety** ($S_h$). Our framework allows us to penalize active misinformation more severely than mere vagueness.
>
> - **Efficiency Dimensions ($D_f$ vs. $D_h$):**
>     While verbose models often inflate token counts, $D_f$ and $D_h$ are normalized by disjoint denominators (count of correct facts vs. count of false facts). One can vary independently from the other (e.g., a verbose but accurate model has high $D_h$ but low $D_f$). The observed correlation suggests that current LLMs tend to be "uniformly verbose" regardless of correctness.
>
>
> ### Q7: When we evaluate one model, how should we holistically interpret the three dimensional metrics you introduced -- factuality, information delivery efficiency and hallucination metrics? What composition of these metrics indicate a good model?
>
> We thank the reviewer for raising this important question about metric synthesis. To enable a holistic interpretation of these three dimensions, we actually developed a Four-Quadrant Framework (detailed in Appendix G.2 and visualized in Figure 11) that synthesizes these metrics into two orthogonal axes.
>
> - **1. How to Interpret Holistically:**
>    We map the metrics onto a 2D plane to visualize trade-offs:
>    * **The Factuality Dimension (X-axis):** Synthesizes **Factuality** ($S_f$) and **Hallucination** ($S_h$). A "Good Model" should move towards the right, indicating it simultaneously maximizes correct facts while minimizing hallucinations, breaking the typical trade-off where increased coverage leads to more errors.
>    * **The Efficiency Dimension (Y-axis):** Synthesizes the token costs ($D_f$ vs. $D_h$). A "Good Model" should move upwards, indicating it can convey correct information concisely ($D_f \downarrow$) without being uniformly verbose when hallucinating. This measures the model's utility per token.
>
> - **2. What Indicates a "Good Model"?**
>    In this framework, an ideal model resides in the **Top-Right Quadrant**.
>    * **Composition:** It achieves high $S_f$ and low $S_h$ (High Intrinsic Knowledge) while maintaining a low $D_f$ (High Information Density).
>    * **Example:** As shown in Figure 11, **GPT-4o** exemplifies a strong model by positioning itself on the far right (high factuality) while maintaining a balanced efficiency profile. Conversely, models that shift to the bottom-left in multi-turn settings demonstrate the degradation we aim to solve.
>
> We believe this Four-Quadrant view provides the holistic standard for evaluating future models in the MT-LFQA task.

---

> ### Author Response · Authors · 2025-11-21
> **Reply to Reviewer J9J3 (Q8)**
>
> ### Q8: What exact prompt was used in prompt-based mitigation (Table 4)? Why did hallucination rise here?
>
> **Exact prompt in Table 4.**\
> We apologize for not explicitly including the prompt text in the paper. The prompt used for the **prompt-based mitigation** setting in Table 4 is:
>
> > “You may refer to the previous conversation for context, but rely primarily on your internal knowledge.”
>
> We will add this prompt verbatim to the revised manuscript (Table 4 caption / Section 5.3) for full transparency and reproducibility.
>
> **Why did hallucination rise here?**\
> The slight rise in hallucination under the prompt-based mitigation setting is **not statistically significant**. On Llama-3.3-70B, **$S_h$ = 0.0432 (prompt-based) vs. 0.0423 (baseline)**, and a paired Wilcoxon Test shows no reliable difference (p = 0.928). Therefore, we attribute this small increase to randomness rather than a systematic prompt effect.

---

### Official Review · Reviewer_RzT5 · 2025-11-04

**Soundness:** 2
**Presentation:** 3
**Contribution:** 2
**Rating:** 2
**Confidence:** 3

**Summary:**

This paper introduces KnowMT-Bench, presented as the first benchmark for Multi-Turn Long-Form Question Answering (MT-LFQA) in knowledge-intensive domains. The authors construct 801 evidence-grounded QA instances across medicine, finance, and law, with dialogues spanning 2-5 turns. Question sequences are generated using 18 predefined templates (8 for 2-turn, 10 for 3-turn) and expanded via LLM generation with human refinement.
The evaluation employs a two-stage NLI-based pipeline: (1) decomposing answers into atomic facts using Qwen2.5-32B, and (2) judging factual consistency using Qwen2.5-14B, achieving 83.6% F1 agreement with human annotations. The authors evaluate 13 LLMs where models generate their own dialogue history, and assess performance across three dimensions: factuality (precision/recall of facts), hallucination (contradiction rates), and information delivery efficiency (tokens per fact).

**Strengths:**

- Paper presents a well-motivated problem and addresses a clear gap at the intersection of multi-turn dialogue, long-form answers, and knowledge-intensive domains. Table 1 effectively demonstrates no existing benchmark covers all three dimensions.

- Rigorous evaluation framework: Two-stage NLI-based pipeline with atomic fact decomposition achieves 83.6% F1 agreement with human judgments (Cohen's κ=0.80). The symmetric evaluation approach is efficient and interpretable.

- Dynamic evaluation setting: Models generate their own dialogue history, more faithfully capturing real deployment scenarios than static benchmarks.
- Comprehensive experiments: 13 models evaluated with systematic investigation of mitigation strategies. The "replace" experiment (Figure 6) provides evidence for the contextual noise hypothesis.
- High-quality data construction: Multi-stage human validation with authoritative sources and proper inter-annotator agreement metrics.

**Weaknesses:**

- Model-dependent evaluation: Paper relies on Qwen-2.5-32B for decomposition (18.1% SMAPE, 5.9% omission rate) and Qwen-2.5-14B for judgment. While Table 7 rules out self-preference, robustness to evaluator choice is not explored in detail. Further, only 100 dialogues used for human validation.

- Limited scale of eval data: Only 801 instances with severe domain imbalance (72% finance, 26% medicine). Average 2.98 turns (max 5) may not capture degradation in longer conversations. No significance tests or confidence intervals reported for main results.

- Artificial question sequences leads to generalizability concerns: Paper's approach of template-based generation (8-10 templates) with "no leakage" constraint creates unnatural conversation patterns. Real dialogues involve iterative refinement and reference to prior context. There is no analysis comparing template vs. natural conversations.

- Template benchmarking vs true multi-turn reasoning: Since templates create unnatural conversation history, it could mean that benchmark is only helping model better identify the template patterns vs true multi-turn reasoning. Benchmark uses only 8 templates for 2-turn and 10 templates for 3-turn dialogues, creating serious risk that models could likely exploit template patterns rather than demonstrate genuine multi-turn reasoning.

- Weak causal evidence: The claim that "contextual noise" causes degradation conflates history quality with model capability. "Replace" experiment only tests two weaker models with GPT-4o history (not vice versa). Alternative explanations (context length, position bias, different information) not ruled out.

- Shallow analysis: Missing error pattern analysis, failure mode characterization, and domain-specific insights. Table 2 shows domain results but minimal discussion. No qualitative examples of where/why RAG helps or fails.

**Questions:**

- Causality: Can you provide a synthetic/controlled experiment where model knowledge is verified in single-turn, then test multi-turn with systematic noise injection? This would establish causal evidence for "contextual noise."

- Multi-turn vs. long-context: Can you add baselines comparing (a) single-turn with concatenated history, (b) multi-turn with human-written history, and (c) multi-turn with ground-truth history? This would distinguish dialogue-structure effects from long-context effects.

- SFT: Why is supervised fine-tuning not explored given that (a) domain-FT helps (Table 2), (b) prompting shows promise but limitations, and (c) this is the standard approach for teaching models new behaviors?

---

> ### Author Response · Authors · 2025-11-21
> **Reply to Reviewer RzT5 (Q1)**
>
> ### Q1: Model-dependent evaluation: Paper relies on Qwen-2.5-32B for decomposition (18.1% SMAPE, 5.9% omission rate) and Qwen-2.5-14B for judgment. While Table 7 rules out self-preference, robustness to evaluator choice is not explored in detail. Further, only 100 dialogues used for human validation.
>
> We thank the reviewer for raising concerns about the **potential bias from relying on a single Qwen family in the evaluation pipeline** and about the **scale of human validation**. We agree that cross-model validation is crucial for ensuring the generality of our conclusions, and we have conducted additional analysis as follows.
>
> **(1) Cross-model robustness of the Evaluator: stable ranking and consistent core conclusion.**\
> To directly address the concern that “relying on a single model family as Evaluator may bias the conclusions,” we introduce **GPT-4o** as an alternative Evaluator, replacing only the Evaluator while keeping the Decomposer and all other components unchanged. We then re-evaluate four representative models (GPT-4o, DeepSeek-V3, Llama-3.3-70B, Qwen-2.5-72B) under both **single-turn** and **multi-turn** settings.
>
> We observe **strong statistical consistency** between the two Evaluators:
>
> * **High correlation with a systematic offset.** The $S_f$ from Qwen-2.5-14B and GPT-4o are almost linearly aligned (**Pearson (r=0.984)**), while GPT-4o consistently yields higher absolute values (Table R2), indicating a **global calibration shift rather than a change in relative evaluation**. Hence the cross-model comparisons and our conclusions remain stable.
> * **Stable ranking across settings.** The relative ordering of models is largely preserved (**Spearman $\rho = 0.905$**). As shown in Table R1, both Evaluators consistently identify GPT-4o as top-tier in both modes, while the open-source models exhibit only minor permutations when scores are close.
> * **Core conclusion unchanged.** Under both Evaluators, all models show a consistent degradation in factual performance from single-turn to multi-turn evaluation (Table R2). Therefore, our key finding—**multi-turn history degrades factual ability**—is not an artifact of a Qwen-specific Evaluator.
>
> **Table R1: Model Ranking Consistency across Evaluators**
>
> | Original Rank | Model (Mode)           | $S_f$ (Original Eval) | $S_f$ w/ GPT-4o Eval (Rank) |
> | --- | --- | --- | --- |
> | 1  | GPT-4o (Single)        | 28.77%                | 38.06% (1) |
> | 2  | GPT-4o (Multi)         | 27.63%                | 37.86% (2) |
> | 3  | Llama-3.3-70B (Single) | 24.02%                | 32.95% (4)|
> | 4  | Qwen-2.5-72B (Single)  | 23.97%                | 32.37% (5)|
> | 5  | DeepSeek-V3 (Single)   | 23.80%                | 33.82% (3) |
> | 6  | Llama-3.3-70B (Multi)  | 21.16%                | 29.26% (6) |
> | 7  | Qwen-2.5-72B (Multi)   | 20.20%                | 28.54% (8) |
> | 8  | DeepSeek-V3 (Multi)    | 19.38%                | 29.19% (7)  |
>
> **Table R2: Cross-Model Validation – Detailed Metrics for Evaluator Variants**
>
> | Model         | Setting | Configuration  | $S_f$ (%) | $S_h$ (%) | $D_f$ | $D_h$ |
> | --- | --- | --- | --- | --- | --- | --- |
> | GPT-4o        | Single  | Original       | 28.77     | 2.26      | 226   | 506   |
> | GPT-4o        | Single  | w/ GPT-4o Eval | 38.06     | 3.36      | 179   | 434   |
> | GPT-4o        | Multi   | Original       | 27.63     | 2.48      | 254   | 555   |
> | GPT-4o        | Multi   | w/ GPT-4o Eval | 37.86     | 3.11      | 214   | 532   |
> | DeepSeek-V3   | Single  | Original       | 23.80     | 3.21      | 343   | 740   |
> | DeepSeek-V3   | Single  | w/ GPT-4o Eval | 33.82     | 4.19      | 285   | 722   |
> | DeepSeek-V3   | Multi   | Original       | 19.38     | 3.40      | 444   | 888   |
> | DeepSeek-V3   | Multi   | w/ GPT-4o Eval | 29.19     | 4.32      | 373   | 837   |
> | Llama-3.3-70B | Single  | Original       | 24.02     | 3.49      | 324   | 628   |
> | Llama-3.3-70B | Single  | w/ GPT-4o Eval | 32.95     | 5.12      | 272   | 628   |
> | Llama-3.3-70B | Multi   | Original       | 21.16     | 4.25      | 437   | 760   |
> | Llama-3.3-70B | Multi   | w/ GPT-4o Eval | 29.26     | 6.00      | 383   | 732   |
> | Qwen-2.5-72B  | Single  | Original       | 23.97     | 2.44      | 358   | 821   |
> | Qwen-2.5-72B  | Single  | w/ GPT-4o Eval | 32.37     | 3.92      | 334   | 791   |
> | Qwen-2.5-72B  | Multi   | Original       | 20.20     | 2.77      | 484   | 900   |
> | Qwen-2.5-72B  | Multi   | w/ GPT-4o Eval | 28.54     | 4.05      | 434   | 872   |
>
> **(2) Scale of human validation (brief clarification).**
> Although we manually inspect 100 dialogue sessions, these dialogues come from both ground-truth references and real model-generated answers, and are expanded into **1,687 atomic statement–level judgment pairs** for annotation and alignment. Thus, at the *fine-grained factual level*, the amount of human supervision is sufficient to support our reliability analysis of the evaluation framework.

---

> > ### Author Response · Authors · 2025-11-21
> > **Reply to Reviewer RzT5 (Q2-Part1)**
> >
> > ### Q2: Limited scale of eval data: Only 801 instances with severe domain imbalance (72% finance, 26% medicine). Average 2.98 turns (max 5) may not capture degradation in longer conversations. No significance tests or confidence intervals reported for main results.
> >
> > #### 1. Regarding limited scale of eval data
> >
> > We agree that KnowMT-Bench is smaller than some broad multi-turn benchmarks. However, for LFQA, tasks that are more closely aligned with ours, the scale of our dataset is comparable. For example, the single-turn LFQA benchmark **FinTextQA** [a] contains 1,262 QA pairs, whereas our **KnowMT-Bench** includes 801 QA pairs.
> >
> > Compared with existing large-scale benchmarks, we intentionally make a trade-off between **“size”** and **“fine-grained, human-aligned, interpretable annotation”**, and deliberately prioritize the latter. Concretely, we devote most of our annotation budget to **atomic-fact–level labeling and verification**. For each multi-turn QA pair, we decompose the final answer into a set of atomic facts, and perform manual decomposition and consistency checks. Based on these atomic facts, we build a fine-grained evaluation protocol for multi-turn dialogue, which is more aligned with human judgments of “what was said correctly” and “where it went wrong” than directly using LLM-based scoring or traditional text-similarity metrics such as BLEU and ROUGE, and is also more interpretable.
> >
> > In addition, we invest substantial annotation effort to ensure that both the **decomposer** and the **evaluator** are highly consistent with human judgments. This allows our evaluation framework to measure not only **how much relevant knowledge the answer covers (recall)**, but also **how much of the content is factually correct (precision)**, resulting in a relatively comprehensive automatic evaluation framework for knowledge-intensive multi-turn dialogue.
> >
> > Our design is inspired by the LFQA evaluation framework **K-QA**, which also evaluates using atomic knowledge. However, K-QA focuses only on the **recall** dimension and annotates just 201 QA pairs. Building on this, we extend both the **evaluation dimensions** (by additionally modeling precision) and the **data scale**.
> >
> > #### 2. Regarding the domain imbalance
> >
> > We acknowledge the reviewer’s observation that, at the coarse-grained label level, the proportion of items tagged as “Finance” is relatively high. However, we would like to clarify that, from a capability perspective, the samples labeled as “Finance” do not belong to a single domain; they actually contain two essentially different types of tasks:
> >
> > - **Financial–legal / regulatory QA (~300 instances).**
> >   These questions revolve around specific financial laws, regulatory rules, and compliance requirements, such as the applicability conditions of EU regulations or the calculation of regulatory thresholds in cross-border mergers and acquisitions. This subset mainly tests a model’s ability to understand legal provisions and perform legal reasoning, although the legal questions happen to be situated in financial scenarios. From the capability perspective, it is therefore closer to “law QA” than to traditional everyday financial knowledge.
> >   For example, a financial–legal question is:
> >   *“How should the turnover threshold of Article 20(3)(a) of Regulation (EU) 2022/2560 be calculated for concentrations consisting in the creation of a joint venture?”*
> >
> > - **Everyday finance consultation QA (~300 instances).**
> >   This subset focuses on financial questions that ordinary users care about, such as whether bank accounts can still be used normally, explanations of product and service rules, and general interpretations of financial policies. Compared with the first subset, this mainly tests a model’s everyday financial knowledge.
> >   For example, an everyday finance question is:
> >   *“If Citizens Bank has closed, can I still use my deposit accounts normally?”*
> >
> > Therefore, although both types are statistically tagged under the “Finance” domain, in the model capability space they correspond to two different directions, legal reasoning ability and everyday financial knowledge, rather than a single homogeneous domain.

---

> > > ### Author Response · Authors · 2025-11-21
> > > **Reply to Reviewer RzT5 (Q2-Part2)**
> > >
> > > #### 3. Regarding the Average turns
> > >
> > > We acknowledge that, in our benchmark, the number of dialogue turns is indeed concentrated between 2 and 5. However, this distribution is not an arbitrarily chosen “shallow” interaction pattern; it is derived from an **empirical analysis of real-world interaction data for our target task**.
> > >
> > > Concretely, we start from a large human–ChatGPT dialogue dataset with topic annotations, **ShareGPT-Chinese-English-90k** [b]. From this dataset, we first filter out all dialogue histories related to medicine, law, and finance, and then use an LLM (Llama-3.3-70B) to further select those that correspond to knowledge-based long-form QA. On the resulting subset, we compute **the empirical distribution of dialogue length** (in turns): 2-turn (38.5%), 3-turn (38.2%), 4-turn (14.0%), 5-turn (5.4%), and the total proportion of dialogues longer than 5 turns is only ≈3.9%.
> > >
> > > Since the proportion of dialogues longer than 5 turns is below 5%, we group all such cases into the 5-turn category when constructing our benchmark. The final dialogue-length distribution used in our benchmark (2–5 turns) is therefore: 37.45%, 37.45%, 14.98%, 10.11%, which is **a direct modeling of the above empirical observation**.
> > >
> > > Therefore, although “real expert consultations” or other benchmarks in different application scenarios may exhibit longer interaction patterns, our analysis indicates that in current (at least LLM-based) knowledge-intensive QA settings, the vast majority of interactions are completed within 2–5 turns. Our benchmark is designed to **faithfully reflect the interaction patterns of this specific task**, rather than to enforce a pre-assumed length standard that may not be appropriate for this scenario.
> > >
> > > [a] Chen, Jian, et al. "Fintextqa: A dataset for long-form financial question answering." Proceedings of the 62nd Annual Meeting of the Association for Computational Linguistics (Volume 1: Long Papers). 2024.
> > >
> > > [b] shareAI. Sharegpt-chinese-english-90k bilingual human-machine qa dataset. https://huggingface.co/datasets/shareAI/ShareGPT-Chinese-English-90k, 2023.
> > >
> > > #### 4. Regarding the significance tests or confidence intervals reported for main results.
> > >
> > > We fully agree that the statistical analysis of performance comparisons should be strengthened. In the final version of the paper, we will add **statistical significance tests** (e.g., paired *t*-test or Wilcoxon signed-rank test) for key comparisons (such as the performance differences between single-turn and multi-turn settings in Fig. 4), and report the corresponding *p*-values to ensure the robustness of our conclusions.
> > >
> > > For Figure 4, we ran **one-sided paired Wilcoxon tests** for all experiments, testing whether the multi-turn setting is worse than the single-turn setting. At the 0.05 significance level, we summarize the results below (the full table will be provided in the appendix of the revised version):
> > >
> > > | Metric | significantly worse (*p* < 0.05) | %     | numerically worse | %     |
> > > | ------ | ------------------------- | ----- | ------------------- | ----- |
> > > | $S_f$  | 10/12                     | 83.3% | 11/12               | 91.7% |
> > > | $S_h$  | 3/12                      | 25.0% | 11/12               | 91.7% |
> > > | $D_f$  | 10/12                     | 83.3% | 11/12               | 91.7% |
> > > | $D_h$  | 9/12                      | 75.0% | 9/12                | 75.0% |
> > >
> > > For $S_h$, the statistical significance is weaker because many samples have zero hallucination, but in terms of the raw values, 11/12 (91.7%) of the models still show multi-turn performing worse than single-turn.
> > >
> > > For Figure 6, we ran **paired Wilcoxon tests** for the experiment. We find that in most cases, replacing the dialogue history of a weaker model with that of a stronger model such as GPT-4o leads to a **statistically significant performance gain**. Below we show the results on $S_f$ as an example:
> > >
> > > | Baseline     | Variant                   | Direction          | p |
> > > | ------------ | ------------------------- | ------------------ | ------: |
> > > | Qwen-2.5-7B  | with_gpt4o_history | variant > baseline | 0.0000 |
> > > | Llama-3.1-8B | with_gpt4o_history   | variant > baseline | 0.0002 |

---

> > > > ### Author Response · Authors · 2025-11-21
> > > > **Reply to Reviewer RzT5 (Q3-part1)**
> > > >
> > > > ### Q3: Artificial question sequences leads to generalizability concerns: Paper's approach of template-based generation (8-10 templates) with "no leakage" constraint creates unnatural conversation patterns. Real dialogues involve iterative refinement and reference to prior context. There is no analysis comparing template vs. natural conversations.
> > > >
> > > > We thank the reviewer for raising concerns about **generalization**, which touches on a core challenge in benchmark construction. We agree that any dataset that is not fully collected *in the wild* carries the risk of being “man-made.” Below we clarify our methodology, motivation, and data quality from four perspectives.
> > > >
> > > > #### 1. Methodological clarification: templates + human review
> > > >
> > > > We would like to emphasize that our data construction is **not** a fully automated, purely template-driven process. As described in Sec. 3.2 and App. D.2, our pipeline follows a **human-in-the-loop** paradigm:
> > > >
> > > > - **Templates as scaffolding.**
> > > >   Automatic generation (using a small set of 2-turn and 3-turn templates) is only used to produce **initial 2–3 turn question sequences** as a rough scaffold.
> > > >
> > > > - **Human review as the core quality control.**
> > > >   **All** automatically generated question sequences are then subjected to rigorous **human review, editing, and extension**. This step ensures that the final sequences are natural, logically coherent, and aligned with our design principles.
> > > >   This “automatic draft + human refinement” procedure is consistent with the current practice in multi-turn dialogue benchmarks such as **StructFlowBench** and **MT-Bench-101**, and our methodology is aligned with these SOTA works.
> > > >
> > > > #### 2. Motivation for the “no leakage” principle
> > > >
> > > > The reviewer questions whether the **no-leakage constraint** might hurt naturalness. In fact, this principle is introduced **precisely to improve naturalness**, not to undermine it.
> > > >
> > > > - **Avoiding unnatural redundancy.**
> > > >   In early experiments without constraints, LLMs tended to generate **front-loaded questions that simply restate the final target**. For example, when the final question is “What are the eligibility criteria for CBILS?”, the model often proposes prior questions like “Who can apply for CBILS?”.
> > > >   In realistic goal-oriented expert consultations, such redundant restatements of the same intent across turns are **unnatural**. Our no-leakage constraint is designed to prevent this kind of templated repetition and encourage genuinely incremental questioning instead.

---

> > > > > ### Author Response · Authors · 2025-11-21
> > > > > **Reply to Reviewer RzT5 (Q3-part2)**
> > > > >
> > > > > #### 3. Empirical evaluation: quantitative comparison with other multi-turn benchmarks
> > > > >
> > > > > To directly address the concern that our dialogue patterns may be “unnatural”, we conduct an additional empirical study **Complex Scenario Suitability Study** from StructFlowBench [a]. Specifically, we follow the setup of StructFlowBench, using **GPT-4o** as an automatic evaluator to rate 50 randomly sampled dialogues from each dataset on three dimensions (1–5 scale):
> > > > > - **Logical Coherence** – whether the dialogue is logically consistent without abrupt or unreasonable jumps.
> > > > > - **Goal Clarity** – whether the dialogue clearly conveys the task goal and keeps both user and system intent transparent.
> > > > > - **Transition Naturalness** – whether transitions between turns are smooth and natural, without awkward or forced shifts.
> > > > >
> > > > > We also define a **Confusion Factor (CF)** as a composite indicator: it is the proportion of dialogue samples whose *average* score across the three dimensions is **≥ 4**. CF serves as a proxy for how well a dataset aligns with **real-world complex interactions**.
> > > > >
> > > > > Our results are:
> > > > >
> > > > > | Dataset                | Logical Coherence | Goal Clarity | Transition Naturalness | Overall Mean | Confusion Factor (CF) |
> > > > > |------------------------|-------------------|--------------|------------------------|--------------|-----------------------|
> > > > > | **Ours (KnowMT-Bench)** | 4.72              | 4.46         | 4.12                   | 4.43         | 0.78                  |
> > > > > | **StructFlowBench** [c] | 4.78          | 4.46         | 4.18                   | 4.47         | 0.82                  |
> > > > > | **MT-Bench-101** [d]  | 4.50          | 4.52         | 4.50                   | 4.51         | 0.74                  |
> > > > >
> > > > > The results clearly show that our benchmark (**overall mean 4.43, CF = 0.78**) is on the **same quality level** as two high-quality multi-turn dialogue benchmarks, StructFlowBench (**4.47, 0.82**) and MT-Bench-101 (**4.51, 0.74**). Notably, our **CF (0.78)** is even *higher* than that of MT-Bench-101 (0.74). This provides strong evidence that multi-turn dialogues generated from our question sequences are **reasonably natural**, rather than being artifacts of rigid templates.
> > > > >
> > > > > [c] Li, Jinnan, et al. "Structflowbench: A structured flow benchmark for multi-turn instruction following." ACL-Findings 2025.
> > > > >
> > > > > [d] Bai, Ge, et al. "Mt-bench-101: A fine-grained benchmark for evaluating large language models in multi-turn dialogues." ACL 2024.
> > > > >
> > > > > #### 4. On “iterative refinement” and referencing prior context
> > > > >
> > > > > The reviewer notes that real-world dialogues often involve **iterative refinement** and **explicit references to prior context**. We fully agree, but we consider this to be a largely **orthogonal challenge**:
> > > > >
> > > > > - **Scope of KnowMT-Bench.**
> > > > >   Our benchmark is designed to evaluate models’ ability to **maintain and synthesize factual knowledge across multiple turns**—a crucial yet under-explored capability in the MT-LFQA setting.
> > > > >
> > > > > - **Complementarity to other benchmarks.**
> > > > >   Benchmarks like StructFlowBench focus more on **instruction-following and refinement**, such as editing or correcting previous outputs. Our task design is intentionally different: it aims to fill the gap in evaluating **knowledge-intensive long-form QA** in multi-turn scenarios, rather than duplicating existing instruction-following benchmarks.
> > > > >
> > > > > **Summary.**
> > > > > Our question sequences are constructed via a **carefully controlled human-in-the-loop process**, where the **no-leakage principle** is introduced to **improve**, not harm, naturalness. Empirical evidence further shows that the resulting dialogues are **on par with popular multi-turn benchmarks** in terms of coherence, clarity, and naturalness. We fully acknowledge that iterative refinement is an important challenge, and view it as a **complementary direction** that our work is designed to coexist with, rather than replace.

---

> > > > > > ### Author Response · Authors · 2025-11-21
> > > > > > **Reply to Reviewer RzT5 (Q4)**
> > > > > >
> > > > > > ### Q4: Template benchmarking vs true multi-turn reasoning: Since templates create unnatural conversation history, it could mean that benchmark is only helping model better identify the template patterns vs true multi-turn reasoning. Benchmark uses only 8 templates for 2-turn and 10 templates for 3-turn dialogues, creating serious risk that models could likely exploit template patterns rather than demonstrate genuine multi-turn reasoning.
> > > > > >
> > > > > > Please see our response to Q3 for a detailed discussion.

---

> > > > > > > ### Author Response · Authors · 2025-11-21
> > > > > > > **Reply to Reviewer RzT5 (Q5)**
> > > > > > >
> > > > > > > ### Q5: Weak causal evidence: The claim that "contextual noise" causes degradation conflates history quality with model capability. "Replace" experiment only tests two weaker models with GPT-4o history (not vice versa). Alternative explanations (context length, position bias, different information) not ruled out.
> > > > > > >
> > > > > > > We thank the reviewer for the detailed feedback on our analysis. We agree that the original phrasing “contextual noise causes degradation” was too strong. In the revised version, following the reviewer’s suggestion, we add a series of controlled studies designed to more clearly disentangle three interacting factors: context length, dialogue template structure, and history quality / contextual noise.
> > > > > > >
> > > > > > > Below we report the results for each experiment (all new tables will be included in the revised paper).
> > > > > > >
> > > > > > > #### 1. History truncation experiment:
> > > > > > >
> > > > > > > To disentangle **length effects** from other factors, we run a history truncation study on **Qwen-2.5-7B**. For each multi-turn answer, we truncate the *original generated response* to a maximum length of *full (1024) / 512 / 256 / 128* tokens, while keeping the questions, evaluation protocol, and model unchanged. The results are shown in Table R3:
> > > > > > >
> > > > > > > **Table R3: Qwen-2.5-7B history truncation experiment**
> > > > > > > (The first two columns are percentages, the last two are integer-valued efficiency metrics.)
> > > > > > >
> > > > > > > | Setting          | $S_f$ (%) | $S_h$ (%)   | $D_f$     | $D_h$    |
> > > > > > > |----------------  |---------- |-------------|-----------|----------|
> > > > > > > | multi (full)     | 17.15     | 3.77        | 547       | 847      |
> > > > > > > | multi, cut = 512 | 17.29     | 3.53        | 509       | 844      |
> > > > > > > | multi, cut = 256 | 19.09     | 3.90        | 426       | 633      |
> > > > > > > | multi, cut = 128 | 19.06     | 3.29        | 401       | 674      |
> > > > > > > | single           | 19.67     | 3.57        | 366       | 589      |
> > > > > > >
> > > > > > > We observe that:
> > > > > > >
> > > > > > > - As the **maximum length is reduced**, the model’s **factual performance** in the multi-turn setting **gradually approaches the single-turn level**, indicating that longer contexts do exert a negative effect on factuality. However, **reliability ($S_h$)** does **not** show a clear monotonic pattern across truncation levels.
> > > > > > > - At the same time, shorter maximum lengths lead to **higher density of correct information** in the output ($D_f$ decreases from 547 → 401), but also to a **higher density of hallucinations** ($D_h$ decreases from 847 → 633/674). This suggests that **information density is more sensitive to context length** than raw F1 scores alone would indicate.
> > > > > > >
> > > > > > > Therefore, we conclude that longer contexts contribute negatively to Factuality, more strongly so for information-density–oriented metrics. We will report these experimental findings in the revised version of the paper.

---

> > > > > > > > ### Author Response · Authors · 2025-11-21
> > > > > > > > **Reply to Reviewer RzT5 (Q5-Part2)**
> > > > > > > >
> > > > > > > > #### 2. Concatenated-history experiment:
> > > > > > > >
> > > > > > > > To directly test whether the degradation comes from the **multi-turn templated structure**, we add a **“single-turn + concatenated history”** experiment on two models:
> > > > > > > >
> > > > > > > > * **Qwen-2.5-7B**
> > > > > > > > * **Qwen-2.5-32B**
> > > > > > > >
> > > > > > > > For each dialogue, we concatenate *all previous turns plus the final user question* into **one long prompt**, and let the model regenerate the answer **once**. This keeps the **information content fixed** while removing **multi-turn templated structure effects**. Results are shown below.
> > > > > > > >
> > > > > > > > **Table R4a: Qwen-2.5-7B single / multi / concatenated comparison**
> > > > > > > >
> > > > > > > > | Setting              | $S_f$ (%)   | $S_h$ (%)   | $D_f$     | $D_h$    |
> > > > > > > > | --- | --- | --- | --- | --- |
> > > > > > > > | single               | 19.67       | 3.57        | 366       | 589      |
> > > > > > > > | multi (templated)    | 17.15       | 3.77        | 547       | 847      |
> > > > > > > > | concatenated history | 21.87       | 4.93        | 492       | 731      |
> > > > > > > >
> > > > > > > > **Table R4b: Qwen-2.5-32B single / multi / concatenated comparison**
> > > > > > > >
> > > > > > > > | Setting              | $S_f$ (%)   | $S_h$ (%)   | $D_f$     | $D_h$    |
> > > > > > > > | --- | --- | --- | --- | --- |
> > > > > > > > | single               | 22.50       | 2.57        | 367       | 685      |
> > > > > > > > | multi (templated)    | 21.62       | 3.02        | 435       | 746      |
> > > > > > > > | concatenated history | 26.13       | 4.75        | 365       | 663      |
> > > > > > > >
> > > > > > > > We observe consistent patterns across both experiments:
> > > > > > > >
> > > > > > > > * **Multi-turn templating introduces an extra negative structural effect.**
> > > > > > > >   With identical history information, switching from **templated multi-turn** to **concatenated single-shot** substantially improves factuality for both models:
> > > > > > > >
> > > > > > > >   * 7B: **17.15% → 21.87%**
> > > > > > > >   * 32B: **21.62% → 26.13%**
> > > > > > > >   (even higher than single-turn)
> > > > > > > >
> > > > > > > > * **However, concatenation also increases hallucinations.**
> > > > > > > >   F1_hall rises in both models under concatenation:
> > > > > > > >
> > > > > > > >   * 7B: **3.77% → 4.93%**
> > > > > > > >   * 32B: **3.02% → 4.75%**
> > > > > > > >
> > > > > > > > This suggests that removing the multi-turn template improves factual extraction, while the longer single-shot concatenated context introduces a hallucination trade-off.
> > > > > > > >
> > > > > > > > * **Information-density metrics further support the structural-effect hypothesis.**
> > > > > > > >   The templated multi-turn setting produces the **largest** $D_f$ / $D_h$ (i.e., the **lowest information density**) for both models, indicating that turn-by-turn templating leads to more verbose and less efficient use of history.
> > > > > > > >
> > > > > > > > Taken together, these two experiments show that the observed degradation in multi-turn LFQA is **not attributable to contextual noise alone**. Even under fixed information content, the **multi-turn templated conditioning itself** can hinder factual exploitation of history, while a concatenated single-shot format is more effective for factuality (albeit with a hallucination trade-off). We will include both tables and this interpretation in the revision.
> > > > > > > >
> > > > > > > >
> > > > > > > > #### 3. Cross-model bidirectional replace: history quality effects are architecture-agnostic**
> > > > > > > >
> > > > > > > > In the main paper we only reported the “weak model + GPT-4o history” replace case. Following the reviewer’s suggestion, we additionally conduct **bidirectional history swap** experiments among three models: **GPT-4o, Llama-3.1-8B, and Qwen-2.5-7B**. Results are shown in Table R5.
> > > > > > > >
> > > > > > > > **Table R5: Cross-model history replacement experiment**
> > > > > > > >
> > > > > > > > | Base model | History source    | $S_f$ (%)   | $S_h$ (%)   | $D_f$     | $D_h$    |
> > > > > > > > |---:|---|---:|---:|---:|---:|
> > > > > > > > | GPT-4o     | single            | 28.77       | 2.26        | 226       | 506      |
> > > > > > > > | GPT-4o     | multi (self)      | 27.63       | 2.48        | 254       | 555      |
> > > > > > > > | GPT-4o     | Llama-8B history  | 30.35       | 3.58        | 275       | 576      |
> > > > > > > > | GPT-4o     | Qwen-7B history   | 30.01       | 3.69        | 337       | 651      |
> > > > > > > > | Llama-8B   | single            | 18.02       | 4.01        | 405       | 589      |
> > > > > > > > | Llama-8B   | multi (self)      | 18.35       | 4.13        | 479       | 753      |
> > > > > > > > | Llama-8B   | GPT-4o history    | 20.99       | 4.03        | 459       | 654      |
> > > > > > > > | Qwen-7B    | single            | 19.65       | 3.49        | 367       | 590      |
> > > > > > > > | Qwen-7B    | multi (self)      | 17.15       | 3.77        | 547       | 847      |
> > > > > > > > | Qwen-7B    | GPT-4o history    | 20.74       | 3.54        | 428       | 734      |
> > > > > > > >
> > > > > > > > We observe that:
> > > > > > > >
> > > > > > > > - For the **weaker models (Llama-8B / Qwen-7B)**, replacing their own multi-turn histories with **GPT-4o histories** **consistently improves $S_f$** and also improves information density ($D_f$ and $D_h$  move closer to the single-turn level).
> > > > > > > >
> > > > > > > > - For **GPT-4o itself**, when we replace its history with histories generated by **weaker models**, $S_f$ **increases further** (up to 30.35%), but at the cost of **more hallucinations** and reduced information density. In other words, plugging low-quality histories into a strong model makes it produce more content to answer the question, but that extra content also brings more hallucinations and irrelevant information.

---

> > > > > > > > > ### Author Response · Authors · 2025-11-21
> > > > > > > > > **Reply to Reviewer RzT5 (Q5-Part3)**
> > > > > > > > >
> > > > > > > > > #### 4. Controlled noise injection (with concatenated variant):
> > > > > > > > >
> > > > > > > > > To further investigate whether contextual noise degrades factual ability, we design a **controlled noise injection study** that strictly follows the reviewer’s proposed protocol: first verify that the model knows the facts, and then systematically corrupt the history.
> > > > > > > > >
> > > > > > > > > (1) Step 1: Selecting “high quality samples” (fixed knowledge capability)
> > > > > > > > >
> > > > > > > > > From GPT-4o’s **single-turn** results, we select **85 “high quality samples”** satisfying:
> > > > > > > > > - $S_f \ge 0.6$ and S_h = 0$.
> > > > > > > > >
> > > > > > > > > On this subset, the single-turn baseline is:
> > > > > > > > > - $S_f = 73.26\%$, $S_h = 0.00\%$;
> > > > > > > > > - $D_f = 53$, $D_h = \infty$ (no hallucinations at all).
> > > > > > > > >
> > > > > > > > > These samples can be viewed as an approximate **upper bound under “noise-free” conditions**, where the model demonstrably possesses the required knowledge.
> > > > > > > > >
> > > > > > > > > (2) Step 2: Modifying only the previous-turn history (original multi-turn format)
> > > > > > > > >
> > > > > > > > > Keeping the **model**, **current question**, and **evaluation protocol** *fixed*, we vary **only the content of the previous turn** and construct three history types:
> > > > > > > > >
> > > > > > > > > 1. **Irrelevant Noise**
> > > > > > > > >    The history is a QA from a *different* domain; the answer is the ground-truth answer to that unrelated question (no errors, only irrelevant information).
> > > > > > > > >
> > > > > > > > > 2. **Irrelevant Error**
> > > > > > > > >    Same unrelated question as above, but the historical answer is a **fully incorrect** model rewrite of the ground truth, containing no relevant error.
> > > > > > > > >
> > > > > > > > > 3. **Relevant Error**
> > > > > > > > >    We take the **true previous-turn QA** for the current sample and ask the model to rewrite it into a **fully incorrect answer** that is semantically related but factually wrong.
> > > > > > > > >
> > > > > > > > > Under the original multi-turn format, the controlled-noise results are shown in Table R6.
> > > > > > > > >
> > > > > > > > > **Table R6: GPT-4o controlled noise injection (original multi-turn format, perfect-sample subset)**
> > > > > > > > >
> > > > > > > > > | Setting           | $S_f$ (%)   | $S_h$ (%)   | $D_f$     | $D_h$    |
> > > > > > > > > |-------------------|-------------|-------------|-----------|----------|
> > > > > > > > > | baseline (no history) | 73.26 | 0.00        | 53        | ∞        |
> > > > > > > > > | Irrelevant Noise   | 68.26       | 1.12        | 57        | 302      |
> > > > > > > > > | Irrelevant Error | 65.97       | 1.53        | 58        | 327      |
> > > > > > > > > | Relevant Error | 62.21       | 1.38        | 66        | 360      |
> > > > > > > > >
> > > > > > > > > We can see that:
> > > > > > > > >
> > > > > > > > > - These samples already prove that the model has **sufficient knowledge** in the single-turn setting, so the baseline is a reasonable “upper bound” under zero-noise conditions.
> > > > > > > > > - On top of this fixed knowledge capability, **changing only the history noise type** yields a **clear monotonic trend**:
> > > > > > > > >   $$
> > > > > > > > >   S_f: 73.26\% \;(\text{no history}) \;>\; 68.26\% \;(\text{Irrelevant Noise}) \;>\; 65.97\% \;(\text{Irrelevant Error}) \;>\; 62.21\% \;(\text{Relevant Error}).
> > > > > > > > >   $$
> > > > > > > > > - **Information density** ($D_f$, $D_h$) deteriorates as noise becomes stronger, indicating that answers become longer and less efficient.
> > > > > > > > > - For hallucinations, the baseline has **no hallucination at all** ($S_h = 0$, $D_f = \infty$), while all three noisy histories introduce **non-zero $S_h$**, showing that contextual noise **induces new hallucinations** even on originally “zero-hallucination” cases.
> > > > > > > > >
> > > > > > > > >
> > > > > > > > > (3) Step 3: Repeating the same experiment with concatenated history
> > > > > > > > >
> > > > > > > > > To remove any effect from the **multi-turn template itself**, we repeat the noise injection experiment under a **concatenated-history single-turn setting**. The model now receives a **single long prompt** (current question plus concatenated history), but the history content is the same as above. Results are shown in Table R7.
> > > > > > > > >
> > > > > > > > > **Table R7: GPT-4o controlled noise injection (concatenated-history format, perfect-sample subset)**
> > > > > > > > >
> > > > > > > > > | Setting (concatenated history) | $S_f$ (%)   | $S_h$ (%)   | $D_f$     | $D_h$    |
> > > > > > > > > |--------------------------------|-------------|-------------|-----------|----------|
> > > > > > > > > | Irrelevant Noise      | 68.82       | 1.12        | 57        | 302      |
> > > > > > > > > | Irrelevant Error   | 65.30       | 2.30        | 53        | 289      |
> > > > > > > > > | Relevant Error    | 60.99       | 2.53        | 70        | 286      |
> > > > > > > > >
> > > > > > > > > We observe that:
> > > > > > > > >
> > > > > > > > > - In the concatenated format, the **ordering of $S_f$ remains the same**:
> > > > > > > > >   $$
> > > > > > > > >   \text{Irrelevant Noise} > \text{Irrelevant Error} > \text{Relevant Error}
> > > > > > > > >   \quad (68.82\% > 65.30\% > 60.99\%),
> > > > > > > > >   $$
> > > > > > > > >   which closely matches the original multi-turn results.
> > > > > > > > > - The trends of **$D_f$** and **$D_h$** across the three noise types are also consistent with the original format: **stronger noise ⇒ worse efficiency and more hallucinations**.
> > > > > > > > > - This shows that even when we completely remove the multi-turn structure and keep only a long single-turn context, the **noise in the history content itself still systematically degrades factuality and increases hallucinations**.

---

> > > > > > > > > > ### Author Response · Authors · 2025-11-21
> > > > > > > > > > **Reply to Reviewer RzT5 (Q5-Part4)**
> > > > > > > > > >
> > > > > > > > > > #### 4. Controlled noise injection (with concatenated variant):
> > > > > > > > > > ......
> > > > > > > > > >
> > > > > > > > > > (4) Summary and adjustment
> > > > > > > > > >
> > > > > > > > > > Across the new controlled studies (1–4), we can now separate the degradation into three interacting factors, **context length**, **multi-turn template structure**, and **history noise**, and clarify their respective roles:
> > > > > > > > > >
> > > > > > > > > > * **Context length**
> > > > > > > > > >   The truncation study shows that shorter histories partially narrow the factuality gap between single-turn and multi-turn, indicating that longer contexts contribute negatively to factuality, especially for information-density–oriented metrics.
> > > > > > > > > >
> > > > > > > > > > * **Multi-turn template structure**
> > > > > > > > > >   When the *history information is held fixed*, moving from templated multi-turn to a single-shot concatenated prompt yields a large factual gain for both a weak and a strong LLMs. This demonstrates that the **turn-by-turn template itself** reduces effective history use, beyond length/content effects, although concatenation also brings a hallucination trade-off.
> > > > > > > > > >
> > > > > > > > > > * **History noise**
> > > > > > > > > >   Bidirectional cross-model swaps indicate that varying only the history source/content is systematically associated with changes in factuality and information-density metrics across different base models. Furthermore, our controlled noise-injection protocol provides evidence that is consistent with a causal interpretation: on samples where GPT-4o already answers perfectly in the single-turn setting, increasingly noisy previous-turn histories are accompanied by a monotonic drop in factual F1 and a rise in hallucinations, and these trends persist even after removing multi-turn structure via concatenation. Taken together, these results suggest that history noise is an important contributing factor, rather than merely a confound tied to a specific architecture or template.
> > > > > > > > > >
> > > > > > > > > > Putting these together, we will replace the earlier overly-strong claim *“contextual noise causes degradation”* with a more precise framing that attributes the effect to **joint influences** from multiple factors, e.g.:
> > > > > > > > > >
> > > > > > > > > > > “Our controlled experiments suggest that factual degradation and hallucinations in multi-turn settings arise from the **combined effects** of contextual noise, context length, and multi-turn dialogue structure.”
> > > > > > > > > >
> > > > > > > > > > ---
> > > > > > > > > >
> > > > > > > > > > #### Connection to J9J3 Q2 mitigation experiments
> > > > > > > > > >
> > > > > > > > > > In our response to Reviewer J9J3 Q2, we further explored practical strategies that **directly correspond to the above factors**, testing how multi-turn degradation can be mitigated by *leveraging or removing* length/structure burdens:
> > > > > > > > > >
> > > > > > > > > > * **Summarization (Sum):** before each turn, the same LLM first generates a global summary of the full dialogue history, and then answers based on this compressed context.
> > > > > > > > > >   This targets the **context-length factor** *and* the **multi-turn template structure**, by compressing long, turn-by-turn histories into a single coherent representation, thereby reducing both length burden and turn-by-turn overhead while preserving core information.
> > > > > > > > > >
> > > > > > > > > > * **Context Selection (Select):** before each turn, the LLM automatically selects the most relevant fragments of the full history for the current question, and answers using only these selected snippets.
> > > > > > > > > >   This targets the **length + noise interaction** *as well as* the **multi-turn template structure**, by filtering irrelevant or distracting turns and effectively relaxing the rigid turn-by-turn conditioning into a smaller, task-aligned context.
> > > > > > > > > >
> > > > > > > > > > ---
> > > > > > > > > >
> > > > > > > > > > All new tables and the above clarified conclusions (including the mitigation experiments) will be incorporated into the revised version of the paper.

---

> > > > > > > > > > > ### Author Response · Authors · 2025-11-21
> > > > > > > > > > > **Reply to Reviewer RzT5 (Q6, Q7, Q8, Q9)**
> > > > > > > > > > >
> > > > > > > > > > > #### Q6: Shallow analysis: Missing error pattern analysis, failure mode characterization, and domain-specific insights. Table 2 shows domain results but minimal discussion. No qualitative examples of where/why RAG helps or fails.
> > > > > > > > > > >
> > > > > > > > > > > Since the case study is too long, we will include it in the Appendix in our revised version. In that section, we will further provide a detailed analysis of Retrieval-Augmented Generation (RAG), illustrating under what conditions RAG improves model performance, when it leads to degradation, and when its influence is marginal.
> > > > > > > > > > >
> > > > > > > > > > > #### Q7: Causality: Can you provide a synthetic/controlled experiment where model knowledge is verified in single-turn, then test multi-turn with systematic noise injection? This would establish causal evidence for "contextual noise."
> > > > > > > > > > >
> > > > > > > > > > > We have discussed this question and provide the experiment in **Q5.4**
> > > > > > > > > > >
> > > > > > > > > > > #### Q8: Multi-turn vs. long-context: Can you add baselines comparing (a) single-turn with concatenated history, (b) multi-turn with human-written history, and (c) multi-turn with ground-truth history? This would distinguish dialogue-structure effects from long-context effects.
> > > > > > > > > > >
> > > > > > > > > > > For (a), we have discussed it in **Q5.2 Table R4a and R4b**.
> > > > > > > > > > >
> > > > > > > > > > > For (b) and (c), we appreciate this insightful suggestion as it would indeed help distinguish contextual noise effects. However, implementing these baselines requires collecting additional human-written histories or ground-truth annotations, which involves significant manual effort/resource and is not feasible within the short rebuttal window. We will consider exploring these settings in our future work.
> > > > > > > > > > >
> > > > > > > > > > > #### Q9: SFT: Why is supervised fine-tuning not explored given that (a) domain-FT helps (Table 2), (b) prompting shows promise but limitations, and (c) this is the standard approach for teaching models new behaviors?
> > > > > > > > > > >
> > > > > > > > > > > We appreciate the reviewer’s suggestion. However, our goal in this work is to evaluate models’ **overall robustness in broad, knowledge-intensive professional domains** such as medicine, finance, and law, rather than to adapt them to specific *sub-domains* (e.g., a particular disease or a specific financial product). In our view, such domain knowledge should be acquired during **pre-training** or **broad domain-level SFT**, rather than through targeted fine-tuning on narrow sub-domains. Otherwise, the evaluation would shift from measuring a model’s **general professional competence** to measuring how well it has been tuned to a particular niche, which undermines our intended objective.
> > > > > > > > > > >
> > > > > > > > > > > In the paper, we have already demonstrated the effectiveness of broad domain-level SFT (e.g., HuatuoGPT in Table 2), showing that such SFT indeed improves model performance on MT-LFQA. In contrast, performing SFT on specific sub-domains would bias the benchmark toward those sub-domains and no longer reflect the model’s **true performance in generic professional scenarios**. For this reason, we deliberately **do not conduct sub-domain SFT**, in order to preserve the consistency and interpretability of our evaluation goal.

---

### Official Review · Reviewer_m7UK · 2025-11-04

**Soundness:** 2
**Presentation:** 2
**Contribution:** 2
**Rating:** 4
**Confidence:** 4

**Summary:**

The authors introduce the new benchmark in multi-turn long-form question answering (MT-LFQA). They claim that it is the first benchmark in this field and its questions are spread across various domains - medicine, finance and law. The authors proposed a new data construction pipeline and using that, they were able to generate around 800 samples of data. Various LLMs were then evaluated on this dataset and they showed that LLMs fail to perform well on MT-LFQA tasks. Finally, they showed that the problem can be mitigated by using RAG, while testing other approaches such as domain-specific finetuning and prompt interventions.

**Strengths:**

1. The curation of the dataset was mostly automated except for the part where human evaluation was being used.
2. The authors tested various mitigation strategies to resolve the problem of poor performance of LLM on their dataset and concluded that RAG is the best approach.

**Weaknesses:**

1. KnowMT-Bench is the first benchmark - In introduction (L96,L97), the authors claim that KnowMT-Bench is the first benchmark to study multi-turn long-form question answering. However, there are some studies that have created very similar datasets as curated by the authors ref [a,b,c,d,e]. The authors have also mentioned one of these works along with some others in literature review (See supplementary) that have done MT-LFQA. Can authors point out how exactly their work is different from these works?


2. Low number of turns - The average number of turns in the curated dataset is less than 3 with the highest being 5. This is shallow and does not truly capture the long-horizon conversational dependencies found in real multi-turn expert consultations and they are often longer than 5 turns. Other datasets such as DialogBench [Ref. a and c] have an average number of turns higher than 7. So the practical applicability of dataset is still needed to be justified.

3.  Domain coverage - As mentioned in L180 and L181, the author's proposed data curation process yields 801 high-quality single-turn LFQA instances spanning three domains: finance (579), law (278), and medicine (209). One can note that the data composition is skewed i.e. 579 of 801 QA pairs are finance-related (which is over 70%). This represents a slightly biased evaluation as more questions are for finance.
Furthermore, the fields are limited to just 3 categories and the number of questions are relatively less. Existing benchmark [Ref. a] has more than 15 categories and has more than 10K questions.

4. Weak human alignment - In L249 and 250, the authors mentioned that the agreement between their NLI-based evaluator and the resulting gold annotations from majority voting among three annotators reached an F1-score of 83.6%. I feel that this is quite low and concerning also. A F1 score should be higher than 90% in order to show human alignment.

5. Lack of statistical rigor - Performance comparisons (e.g. in figs. 4,6) rely on descriptive statistics (scatter plots, correlations) without confidence intervals. Moreover, there are no tests across random seeds or variations in decomposition/evaluator models.



References
a) Ou et. al. - https://arxiv.org/abs/2311.01677,
b) Zheng et. al. - https://arxiv.org/pdf/2306.05685,
c) Li et. al. - https://arxiv.org/abs/1710.03957,
d) Zhao et. al. - https://arxiv.org/pdf/2304.09582
e) Feng et. al. - https://arxiv.org/abs/2109.12595

**Questions:**

There are few more comparisons that needs to be made as mentioned in weakness; also practicability and length of corpus seems low as well.

---

> ### Author Response · Authors · 2025-11-21
> **Reply to Reviewer m7UK (Q1)**
>
> ### Q1: KnowMT-Bench is the first benchmark - In introduction (L96,L97), the authors claim that KnowMT-Bench is the first benchmark to study multi-turn long-form question answering. However, there are some studies that have created very similar datasets as curated by the authors ref [a,b,c,d,e]. The authors have also mentioned one of these works along with some others in literature review (See supplementary) that have done MT-LFQA. Can authors point out how exactly their work is different from these works?
>
> We thank the reviewer for pointing us to these relevant references. Some of them are already discussed in our supplementary material; here, we provide a more detailed clarification of what makes **KnowMT-Bench** unique. Our central claim is that **KnowMT-Bench is the first benchmark specifically targeting multi-turn, long-form, factual-knowledge question answering (LFQA).**
>
> The works mentioned by the reviewer differ from our setting along one or more of the following key dimensions:
>
> - **Dialogbench:  Evaluating llms as human-like dialogue systems.** [a]
>   This work adopts a **multiple-choice** evaluation format: the model only needs to select the correct option rather than generate a long-form answer. This is fundamentally different from the **LFQA** task, which is defined as requiring the model to produce detailed, coherent, paragraph-style responses.
>
> - **MT-Bench** [b]
>   MT-Bench is an excellent multi-turn benchmark, but its questions are *“open-ended without reference answers”* and include many **non-factual** tasks, such as creative writing, role-play, and giving advice. In contrast, LFQA benchmarks (including **KnowMT-Bench**) are also open-ended in form, but their evaluation focus is **factuality**: they typically rely on high-quality long-form reference answers to enable fact-level evaluation.
>
> - **DailyDialog** [c]
>   DailyDialog mainly used for: (1) Response Generation, by given a dialog context, and the average ground truth utterance length is about **15 tokens**, which is far below the widely recognized LFQA benchmarks have an average ground truth length of **more than 50 tokens** (2) Dialogue Act / Intention Classification, Emotion Classification tasks, which is totally different from our task.
>
> - **Is chatgpt equipped with emotional dialogue capabilities?** [d]
>   This paper is **an empirical study of the emotional capabilities of ChatGPT**, and focus on: (1) Understanding Capability: Is ChatGPT capable of accurately understanding and interpreting the user’s emotions? (2) Generation Capability: Is ChatGPT capable of eliciting empathy or support towards such emotional states of the user? And the experimental results are based on existing datasets.
>
>   In contrast, our paper proposes a new systematical benchmark to evaluate a diverse suite of LLMs in MT-LFQA, we carefully construct a new dataset which contains 801 high-quality question sequences with human refinement and evaluate the LLMs in 3 dimensions: Factuality, Hallucination and Information Delivery Efficiency.
>
> - **MultiDoc2Dial** [e]
>   This work is a **document-grounded, closed-domain** QA task where the model answers based on a given document. By contrast, LFQA (and **KnowMT-Bench**) is traditionally defined as an **open-domain** setting, designed to evaluate a model’s ability to answer based on its **internal knowledge**, rather than retrieving information from a specific provided context.
>
> In summary, these prior works differ from our setting in at least one of the following aspects:
> - task form (LFQA),
> - evaluation focus (not primarily factuality),
> - answer length (not long-form), or
> - domain setting (closed-domain rather than open-domain).
>
> Therefore, **KnowMT-Bench** remains, to the best of our knowledge, the first benchmark that systematically evaluates models on **open-domain, multi-turn, long-form factual question answering**.
>
> [a] Ou, Jiao, et al. "Dialogbench: Evaluating llms as human-like dialogue systems." Proceedings of the 2024 Conference of the North American Chapter of the Association for Computational Linguistics: Human Language Technologies (Volume 1: Long Papers). 2024.
>
> [b] Zheng, Lianmin, et al. "Judging llm-as-a-judge with mt-bench and chatbot arena." Advances in neural information processing systems 36 (2023): 46595-46623.
>
> [c] Li, Yanran, et al. "DailyDialog: A Manually Labelled Multi-turn Dialogue Dataset." Proceedings of the Eighth International Joint Conference on Natural Language Processing (Volume 1: Long Papers). 2017.
>
> [d] Zhao, Weixiang, et al. "Is chatgpt equipped with emotional dialogue capabilities?." arXiv preprint arXiv:2304.09582 (2023).
>
> [e] Feng, Song, et al. "MultiDoc2Dial: Modeling Dialogues Grounded in Multiple Documents." Proceedings of the 2021 Conference on Empirical Methods in Natural Language Processing. 2021.

---

> ### Author Response · Authors · 2025-11-21
> **Reply to Reviewer m7UK (Q2)**
>
> ### Q2:  Low number of turns - The average number of turns in the curated dataset is less than 3 with the highest being 5. This is shallow and does not truly capture the long-horizon conversational dependencies found in real multi-turn expert consultations and they are often longer than 5 turns. Other datasets such as DialogBench [Ref. a and c] have an average number of turns higher than 7. So the practical applicability of dataset is still needed to be justified.
>
> We acknowledge that, in our benchmark, the number of dialogue turns is indeed concentrated between 2 and 5. However, this distribution is not an arbitrarily chosen “shallow” interaction pattern; it is derived from an **empirical analysis of real-world interaction data for our target task**.
>
> Concretely, we start from a large human–ChatGPT dialogue dataset with topic annotations, **ShareGPT-Chinese-English-90k** [f]. From this dataset, we first filter in all dialogue histories related to medicine, law, and finance, and then use an LLM (Llama-3.3-70B) to further select those that correspond to knowledge-based long-form QA. On the resulting subset, we compute **the empirical distribution of dialogue length** (in turns): 2-turn (38.5%), 3-turn (38.2%), 4-turn (14.0%), 5-turn (5.4%), and the total proportion of dialogues longer than 5 turns is only ≈3.9%.
>
> Since the proportion of dialogues longer than 5 turns is below 5%, we group all such cases into the 5-turn category when constructing our benchmark. The final dialogue-length distribution used in our benchmark (2–5 turns) is therefore: 37.45%, 37.45%, 14.98%, 10.11%, which is **a direct modeling of the above empirical observation**.
>
> Therefore, although “real expert consultations” or other benchmarks in different application scenarios may exhibit longer interaction patterns, our analysis indicates that in current (at least LLM-based) knowledge-intensive QA settings, the vast majority of interactions are completed within 2–5 turns. Our benchmark is designed to **faithfully reflect the interaction patterns of this specific task**, rather than to enforce a pre-assumed length standard that may not be appropriate for this scenario.
>
> [f] shareAI. Sharegpt-chinese-english-90k bilingual human-machine qa dataset. https://huggingface.co/datasets/shareAI/ShareGPT-Chinese-English-90k, 2023.

---

> ### Author Response · Authors · 2025-11-21
> **Reply to Reviewer m7UK (Q3)**
>
> ### Q3: Domain coverage - As mentioned in L180 and L181, the author's proposed data curation process yields 801 high-quality single-turn LFQA instances spanning three domains: finance (579), law (278), and medicine (209). One can note that the data composition is skewed i.e. 579 of 801 QA pairs are finance-related (which is over 70%). This represents a slightly biased evaluation as more questions are for finance. Furthermore, the fields are limited to just 3 categories and the number of questions are relatively less. Existing benchmark [Ref. a] has more than 15 categories and has more than 10K questions.
>
> We acknowledge the reviewer’s observation that, at the coarse-grained label level, the proportion of items tagged as “Finance” is relatively high. However, we would like to clarify that, from a capability perspective, the samples labeled as “Finance” do not belong to a single domain; they actually contain two essentially different types of tasks:
>
> - **Financial–legal / regulatory QA (~300 instances).**
>   These questions revolve around specific financial laws, regulatory rules, and compliance requirements, such as the applicability conditions of EU regulations or the calculation of regulatory thresholds in cross-border mergers and acquisitions. This subset mainly tests a model’s ability to understand legal provisions and perform legal reasoning, although the legal questions happen to be situated in financial scenarios. From the capability perspective, it is therefore closer to “law QA” than to traditional everyday financial knowledge.
>   For example, a financial–legal question is:
>   *“How should the turnover threshold of Article 20(3)(a) of Regulation (EU) 2022/2560 be calculated for concentrations consisting in the creation of a joint venture?”*
>
> - **Everyday finance consultation QA (~300 instances).**
>   This subset focuses on financial questions that ordinary users care about, such as whether bank accounts can still be used normally, explanations of product and service rules, and general interpretations of financial policies. Compared with the first subset, this mainly tests a model’s everyday financial knowledge.
>   For example, an everyday finance question is:
>   *“If Citizens Bank has closed, can I still use my deposit accounts normally?”*
>
> Therefore, although both types are statistically tagged under the “Finance” domain, in the model capability space they correspond to two different directionsm, legal reasoning ability and everyday financial knowledge, rather than a single homogeneous domain.
>
> Beyond domain labels, the reviewer also raises a valid question about dataset scale. We agree that KnowMT-Bench is smaller than some broad multi-turn benchmarks (e.g., [a], which includes more than 10K questions). However, for LFQA, tasks that are more closely aligned with ours, the scale of our dataset is comparable. For example, the single-turn LFQA benchmark **FinTextQA** contains 1,262 QA pairs, whereas our **KnowMT-Bench** includes 801 QA pairs.
>
> Compared with existing large-scale benchmarks, we intentionally make a trade-off between **“size”** and **“fine-grained, human-aligned, interpretable annotation”**, and deliberately prioritize the latter. Concretely, we devote most of our annotation budget to **atomic-fact–level labeling and verification**. For each multi-turn QA pair, we decompose the final answer into a set of atomic facts, and perform manual decomposition and consistency checks. Based on these atomic facts, we build a fine-grained evaluation protocol for multi-turn dialogue, which is more aligned with human judgments of “what was said correctly” and “where it went wrong” than directly using LLM-based scoring or traditional text-similarity metrics such as BLEU and ROUGE, and is also more interpretable.
>
> In addition, we invest substantial annotation effort to ensure that both the **decomposer** and the **evaluator** are highly consistent with human judgments. This allows our evaluation framework to measure not only **how much relevant knowledge the answer covers (recall)**, but also **how much of the content is factually correct (precision)**, resulting in a relatively comprehensive automatic evaluation framework for knowledge-intensive multi-turn dialogue.
>
> Our design is inspired by the LFQA evaluation framework **K-QA**, which also evaluates using atomic knowledge. However, K-QA focuses only on the **recall** dimension and annotates just 201 QA pairs. Building on this, we extend both the **evaluation dimensions** (by additionally modeling precision) and the **data scale**.

---

> ### Author Response · Authors · 2025-11-21
> **Reply to Reviewer m7UK (Q4)**
>
> ### Q4: Weak human alignment - In L249 and 250, the authors mentioned that the agreement between their NLI-based evaluator and the resulting gold annotations from majority voting among three annotators reached an F1-score of 83.6%. I feel that this is quite low and concerning also. A F1 score should be higher than 90% in order to show human alignment.
>
> We believe the **83.6% F1** agreement should be interpreted in the context of both **realistic performance ceilings** and **task difficulty**.
>
> 1. **Even on standard NLI benchmarks, *accuracy* is not necessarily above 90%.**
>
>    Prior work shows that very strong models do **not** always exceed 90% **accuracy** on classic NLI datasets. For example, **[g] (NAACL 2025)** reports that a top-tier model (e.g., Llama-3.1-405B) achieves an accuracy **slightly below 90%** on SNLI/MNLI. This suggests that **>90% accuracy** is not a universal expectation even in domain-agnostic, short-text NLI settings, and it provides a realistic reference point before considering harder domain-specific evaluations.
>
> 2. **Domain-specific and cross-domain NLI remains substantially harder in terms of F1.**
>
>    On difficult domain NLI tasks, state-of-the-art results typically lie in the **0.7–0.8 F1** range:
>
>    * **[h] (ACL-Findings 2025)** proposes a cross-scientific NLI benchmark where the best F1 is only **78.17%** (SciBERT-FT).
>    * **[i] (NAACL-Findings 2025)** reports, on a financial NLI task, an F1 of **78.62%** for the best LLM (Llama-3.1-70B), and **74.57%** for the best fine-tuned PLM (RoBERTa-L).
>      These results show that **F1 well below 90%** is the norm for domain-specific NLI even with strong models.
>
> 3. **Our evaluator faces an even more challenging setting.**
>
>    Our evaluator performs a **cross-domain NLI** task spanning *finance, law, and medicine*, and must decide whether a model-generated **long-form span** entails a given **atomic fact**. Compared with SNLI/MNLI, this setting involves **longer inputs, domain-specific terminology, and cross-domain distribution shifts**, making it substantially more difficult than standard NLI.
>
> Given these realistic ceilings and the difficulty of cross-domain long-form NLI, achieving **83.6% F1** (and **83.0% accuracy**) against human majority vote indicates **acceptable agreement with human judgments**. Therefore, our evaluator is a **reliable** evaluation tool rather than a “worrying” one.
>
> [g] Madaan, Lovish, et al. "Lost in inference: Rediscovering the role of natural language inference for large language models." NAACL 2025.
>
> [h] Shaik, Firoz, et al. "A MISMATCHED Benchmark for Scientific Natural Language Inference." ACL-Findings 2025.
>
> [i] Magomere, Jabez, et al. "FinNLI: Novel dataset for multi-genre financial natural language inference benchmarking." NAACL-Findings 2025.

---

> ### Author Response · Authors · 2025-11-21
> **Reply to Reviewer m7UK (Q5)**
>
> ### Q5: Lack of statistical rigor - Performance comparisons (e.g. in figs. 4,6) rely on descriptive statistics (scatter plots, correlations) without confidence intervals. Moreover, there are no tests across random seeds or variations in decomposition/evaluator models.
>
> We thank the reviewer for the suggestion regarding statistical rigor.
>
> 1. **Random seeds.**
>    For both the Decomposer and Evaluator models, we use **greedy decoding** in all experiments. This decoding procedure is **deterministic**, so the outputs are not affected by the random seed.
>
> 2. **Statistical tests.**
>    We fully agree that the statistical analysis of performance comparisons should be strengthened. In the final version of the paper, we will add **statistical significance tests** (e.g., paired *t*-test or Wilcoxon signed-rank test) for key comparisons (such as the performance differences between single-turn and multi-turn settings in Fig. 4), and report the corresponding *p*-values to ensure the robustness of our conclusions.
>
>    For Figure 4, we ran **one-sided paired Wilcoxon tests** for all experiments, testing whether the multi-turn setting is worse than the single-turn setting. At the 0.05 significance level, we summarize the results below (the full table will be provided in the appendix of the revised version):
>
>    | Metric | significantly worse (*p* < 0.05) | %     | numerically worse | %     |
>    | ------ | ------------------------- | ----- | ------------------- | ----- |
>    | $S_f$  | 10/12                     | 83.3% | 11/12               | 91.7% |
>    | $S_h$  | 3/12                      | 25.0% | 11/12               | 91.7% |
>    | $D_f$  | 10/12                     | 83.3% | 11/12               | 91.7% |
>    | $D_h$  | 9/12                      | 75.0% | 9/12                | 75.0% |
>
>    For $S_h$, the statistical significance is weaker because many samples have zero hallucination, but in terms of the raw values, 11/12 (91.7%) of the models still show multi-turn performing worse than single-turn.
>
>    For Figure 6, we ran **two-sided paired Wilcoxon tests** for the experiment. We find that in most cases, replacing the dialogue history of a weaker model with that of a stronger model such as GPT-4o leads to a **statistically significant performance gain**. Below we show the results on $S_f$ as an example:
>
>    | Baseline     | Variant                   | Direction          | p |
>    | ------------ | ------------------------- | ------------------ | ------: |
>    | Qwen-2.5-7B  | with_gpt4o_history | variant > baseline | 0.0000 |
>    | Llama-3.1-8B | with_gpt4o_history   | variant > baseline | 0.0002 |

---

### Official Review · Reviewer_7n4w · 2025-11-05

**Soundness:** 3
**Presentation:** 3
**Contribution:** 3
**Rating:** 6
**Confidence:** 3

**Summary:**

This paper introduces KnowMT-Bench, a benchmark designed to evaluate Multi-Turn Long-Form Question Answering (MT-LFQA) in domains such as medicine, finance, and law. It aims to address the gap between existing single-turn LFQA datasets and the complex multi-turn nature of real-world professional consultations. The benchmark consists of 801 multi-turn, evidence-grounded QA instances derived from authoritative sources. The work represents a solid benchmark built to simulate a realistic human-LLMs interaction. While it heavily relies on generated data with LLM, human annotation was used to validate the agreement, showing positive results. Findings show that in a multi-turn setup, there is a degradation of performances for all the models.

**Strengths:**

The work represents a solid benchmark to test models in an artificial setup, which represents a good proxy also for real-world use cases. The analysis provides enough in-depth information, and findings (although not novel) represent an interesting starting point to develop further research and models to solve the multi-turn setup.

**Weaknesses:**

The paper has its merits and overall it is a meaningful contribution for the research community. I have mainly two concerns: 1/ the size is limited. While this is a benchmark, it is not clear to me if the dataset contains a representative set of queries where to test generalization of models. 2/ The topic distribution seems to be skewed. Majority of questions are from finance.

I would like authors to provide more details on these choices.

**Questions:**

see Weaknesses

---

> ### Author Response · Authors · 2025-11-21
> **Reply to Reviewer 7n4w**
>
> ### Q1: the size is limited. While this is a benchmark, it is not clear to me if the dataset contains a representative set of queries where to test generalization of models.
>
> We appreciate the reviewer’s concern and would like to clarify that, although KnowMT-Bench is of moderate size, its questions are drawn from broad and professional real-world sources in three high-stakes domains rather than from synthetic prompts. **Concretely, KnowMT-Bench consists of three components: everyday finance QA, financial–legal / regulatory QA, and medical QA.**
>
> - **For everyday finance QA**, we extracted representative topics of people's interest from Google Trends in the “Finance” category and sub-categories over recent 5 years in major English-speaking markets (USA and UK), so that our questions reflect genuine consumer information needs (e.g., saving, investing, credit, retirement planning).
>
> - **For the financial–legal / regulatory QA**, we deliberately include regulations and official QA from multiple key jurisdictions, including the U.S. Securities and Exchange Commission, the Federal Reserve, European regulatory bodies, and the Hong Kong Monetary Authority, thus covering a diverse set of regulatory topics such as disclosure requirements, market conduct, prudential rules, and investor protection across different legal systems.
>
> - **For the medical QA, we adopt K-QA**, which aggregates patient–doctor questions from 15 medical QA websites; these cover common but clinically important conditions such as acute upper respiratory infection, urinary tract infection, acute sinusitis, COVID-19, and acute pharyngitis, and span different information needs (e.g., questions about problems/symptoms, treatments, and diagnostic tests).
>
> Taken together, by deriving everyday finance intents from Google Trends in major English-speaking markets, incorporating financial–legal questions from multiple key jurisdictions (e.g., SEC, Federal Reserve, European regulators, HKMA), and adopting medical QA from 15 patient–doctor websites covering diverse conditions and query types, we aim to make the queries in KnowMT-Bench representative of real-world information needs in these three high-stakes domains despite its moderate size.
>
> ### Q2: The topic distribution seems to be skewed. Majority of questions are from finance.
>
> We acknowledge the reviewer’s observation that, at the coarse-grained label level, the proportion of items tagged as “Finance” is relatively high. However, we would like to clarify that, from a capability perspective, the samples labeled as “Finance” do not belong to a single domain; they actually contain two essentially different types of tasks:
>
> - **Financial–legal / regulatory QA (~300 instances).**
>   These questions revolve around specific financial laws, regulatory rules, and compliance requirements, such as the applicability conditions of EU regulations or the calculation of regulatory thresholds in cross-border mergers and acquisitions. This subset mainly tests a model’s ability to understand legal provisions and perform legal reasoning, although the legal questions happen to be situated in financial scenarios. From the capability perspective, it is therefore closer to “law QA” than to traditional everyday financial knowledge.
>   For example, a financial–legal question is:
>   *“How should the turnover threshold of Article 20(3)(a) of Regulation (EU) 2022/2560 be calculated for concentrations consisting in the creation of a joint venture?”*
>
> - **Everyday finance consultation QA (~300 instances).**
>   This subset focuses on financial questions that ordinary users care about, such as whether bank accounts can still be used normally, explanations of product and service rules, and general interpretations of financial policies. Compared with the first subset, this mainly tests a model’s everyday financial knowledge.
>   For example, an everyday finance question is:
>   *“If Citizens Bank has closed, can I still use my deposit accounts normally?”*
>
> Therefore, although both types are statistically tagged under the “Finance” domain, in the model capability space they correspond to two different directions, legal reasoning ability and everyday financial knowledge, rather than a single homogeneous domain.

---

### Author Response · Authors · 2025-11-27
**Summary of Our Rebuttal (Part I)**

We would like to thank all of the reveiwers for their insightful suggestions to improve our paper. And we have try our best to response to each of the question raised by the reviewers. And here is the summary of our rebuttal.

### 1. Regarding to the dataset size and domain imbalance

While we have responded to these points in each reviewer-specific rebuttal, we would like to provide a brief summary clarification here regarding dataset size and topic distribution.

**(1) Does our benchmark contains a representative set of queries where to test generalization of models?**

- Firstly, We agree that KnowMT-Bench is smaller than some popular multi-turn benchmarks. However, for LFQA, tasks that are more closely aligned with ours, the scale of our dataset is comparable. For example, the single-turn LFQA benchmark FinTextQA contains 1,262 Single-turn QA pairs and KQA only contains 201 QA pairs, whereas our KnowMT-Bench includes 801 **multi-turn* QA pairs.

- Secondly, although KnowMT-Bench is of moderate size, it is built from broad and professional *real-world sources* across three high-stakes domains rather than synthetic prompts. Specifically, it covers (i) everyday finance questions grounded in what people care most about (based on Google Trends in major English-speaking markets), (ii) financial–legal/regulatory questions sourced from authoritative regulators across key jurisdictions, and (iii) medical questions drawn from patient–doctor QA collected from multiple medical websites (K-QA). Together, these components aim to make KnowMT-Bench representative of real-world information needs in finance, regulation, and healthcare despite its moderate scale.

**(2) Majority of questions are from finance. Is the topic distribution skewed?**

We acknowledge the reviewer’s observation that the coarse-grained “Finance” label accounts for a relatively large portion of the data. However, this label groups two qualitatively different capability types rather than a single homogeneous domain: (i) financial-legal/regulatory QA (\~300), which focuses on laws, regulations, and compliance in financial contexts and primarily tests legal understanding and reasoning; and (ii) everyday finance consultation QA (\~300), which reflects ordinary users’ practical financial questions and mainly tests everyday financial knowledge. Therefore, although both subsets are tagged as “Finance” for reporting, they correspond to distinct capability dimensions, legal reasoning vs. everyday financial knowledge, instead of one narrow finance domain.


### 2. Regarding to the potential bia and robustness of our evalution pipeline

Another concern is the potential bias and robustness of our evaluation pipeline. And we would like to make a brief summary clarification here.

**(1) Do the decomposer and evaluator exhibit bias?**

- To answer this question, We compared evaluation outcomes across multiple target models and found the metrics are tightly clustered (F1 roughly 0.81–0.85), and the model from the evaluator’s own family (Qwen2.5-14B) does not receive a disproportionately high score. We further ran non-parametric tests on model accuracy: pairwise Mann–Whitney U tests yield no significant differences at $p < 0.01$, and a Kruskal–Wallis test shows no overall difference ($H = 1.9552$, $p = 0.582$). Overall, we find no evidence that the evaluator systematically favors any model family.

- In addition, We measured decomposition count error against human gold using SMAPE, and similarly tested whether different target models are treated unevenly. The Kruskal–Wallis test indicates no significant global difference ($H = 6.1147, p = 0.1062$). Pairwise Mann–Whitney U tests again show no significant differences at the stricter $p < 0.01$ level; only one pair reaches $p < 0.05$ with a small effect size. This suggests any between-model variation in decomposition error is minor, with no strong evidence of systematic favoritism or penalization.

**(2) Is the evaluation pipeline robust?**

To test robustness, we perform cross-model validation by swapping key components of the pipeline. Specifically, we replace the Decomposer (Qwen-2.5-32B -> GPT-4o) while keeping the evaluator fixed, and replace the Evaluator (Qwen-2.5-14B -> GPT-4o) while keeping the decomposer fixed. The resulting scores and rankings remain highly consistent: replacing the decomposer yields near-perfect agreement (Pearson $r = 0.996$; Spearman $\rho = 0.905$), and replacing the evaluator also shows strong consistency (Pearson $r = 0.984$; Spearman $\rho = 0.905$). Model rankings exhibit only minor permutations, and the overall separation between stronger proprietary models (e.g., GPT-4o) and open-source models is preserved under both single-turn and multi-turn settings. Importantly, across all validation configurations, all models consistently show a drop in factual capability from single-turn to multi-turn, so our main qualitative conclusions remain unchanged.

---

### Author Response · Authors · 2025-11-27
**Summary of Our Rebuttal (Part II)**

### 3. Regarding to the statistic rigor

We thank for the reviewers to raise this concern, and we acknowledge that we have missed this in our initial version. Thus, we make several statistical tests and make a summary here.

**1. About the main results**

In response to the comment on statistical rigor, we have added one-sided Wilcoxon signed-rank tests over all instance pairs for each model and metric (Appendix G.4). For factuality and cost per correct fact ($\mathbf{S_f}$ and $\mathbf{D_f}$), 11 out of 12 models move in the expected degradation direction when going from single-turn to multi-turn (lower $\mathbf{S_f}$, higher $\mathbf{D_f}$), and these changes are statistically significant for 10 models ($p<0.05$), confirming that the multi-turn drop in Figure 4 is robust rather than due to noise. For hallucination and its density counterpart ($\mathbf{S_h}$ and $\mathbf{D_h}$), 11 out of 12 models also change in the expected direction (higher $\mathbf{S_h}$ and mostly higher $\mathbf{D_h}$), and 3 models (for $\mathbf{S_h}$) and 9 models (for $\mathbf{D_h}$) reach significance. The weaker statistical signal on $\mathbf{S_h}$ is mainly due to the data distribution: $\mathbf{S_h}$ is exactly zero on most instances, so $\mathbf{S_h}^{\text{multi}} - \mathbf{S_h}^{\text{single}} = 0$ for a large fraction of pairs, which creates many ties and reduces the power of the Wilcoxon test. Overall, the new analysis supports our main claim that multi-turn interactions systematically reduce factuality and increase the token cost per correct fact, while the trends for hallucination metrics remain directionally consistent with our descriptive findings.

**2. About the other experiment results**

We include the statistic test results of other experiments in our rebuttal.

### 4. Other questions

For other questions, we detailly reply to each reviewer and we thank their great suggestions for improving our paper.

### 5. Our updated submission

We have refined our submission based on the valuable reviews from all of the reviewers and here we give a summary our the improvement in our updated submission:

- We improve the abstract and the introduction to make a more clear finding about the source of the performance degradation of LLMs in multi-turn dialogues

- We rewrite Section 4.3 and add additional experiments to make a deep analysis for different possible causes of this phenomenon

- We rewrite Section 5.3 to add more expriments on prompt-based mitigation strategies and give a more detailed analysis

- We improve Section 6 to enhance the soundness of our paper

- In Appendix D.3.2 and D.3.3, we add comprehensive experiments to test the bias of our evaluation pipeline

- In Appendix F, we add a Complex Scenario Suitability Study to validate that our synthetic multi-turn dialogues align well with real-world interaction patterns and do not introduce unnatural artifacts.

- In Appendix H.4, we add formal statistical significance tests for the main results.

- In Appendix H.8, we add an experiment on topic drift to show that spontaneous topic shifts do not compromise the validity of our findings in knowledge-intensive settings.

- In Appendix K, we add more case studies and provide additional qualitative analysis of these cases.

---

### Author Response · Authors · 2025-12-03
**Statement of Significance**

Dear ICLR 2026 AC, SAC, and PC,

We sincerely appreciate the additional time and effort you are dedicating to the review process. For your convenience, we provide a brief statement of significance for our submission.

To the best of our knowledge, KnowMT-Bench offers the first systematic benchmark for multi-turn long-form QA in knowledge-intensive domains. We complete the atom-based LFQA evaluation paradigm by introducing human-validated automatic decomposition of answers and NLI-based atomic verification. This lets us measure both how many correct facts a model produces (Factual F1, $\mathbf{S_f}$) and how many hallucinated facts it introduces (Hallucination F1, $\mathbf{S_h}$). We also add an efficiency view with cost per correct fact ($\mathbf{D_f}$) and token cost per contradicted fact ($\mathbf{D_h}$), so the metrics capture not only what a model knows, but also how it communicates that knowledge.

We reveal two robust correlations: factuality and hallucination are negatively correlated (higher $\mathbf{S_f}$ tends to accompany lower $\mathbf{S_h}$), and the token cost per correct fact and per contradicted fact are positively correlated (models that are concise for correct content also tend to be concise when making mistakes). These patterns give practitioners clear targets: raise $\mathbf{S_f}$ while reducing $\mathbf{S_h}$, and decouple efficiency by lowering $\mathbf{D_f}$ yet keeping $\mathbf{D_h}$ high, so correct facts are delivered concisely while misleading content remains sparse. Under a dynamic evaluation protocol, we also show a clear degradation from single-turn to multi-turn across diverse models. We trace this decline to self-generated contextual noise (especially relevant misinformation), growing context length, and dialogue structure, and we demonstrate effective mitigations, including per-round retrieval (RAG) and prompt-based interventions. These findings provide a reliable yardstick and concrete guidance for building robust conversation systems in high-stakes applications.

Additionally, we provide a summary of our rebuttal for your reference.

---

### Note · Authors · 2026-01-26

I have read and agree with the venue's withdrawal policy on behalf of myself and my co-authors.

---

### Meta-Review · Area_Chair_7cSB · 2026-01-11

**Summary:**

The paper introduces KnowMT-Bench to evaluate multi-turn, knowledge-intensive dialogues. While the motivation is recognized, the primary concerns focus on the benchmark's quality: the dataset is limited in scale, heavily skewed towards Finance, and features shallow dialogue depth.

Reviewers noted that the predefined, logically progressive questions fail to capture the complexity of real-world interactions, such as spontaneous topic drifts or user corrections.

**Reviewer Concerns:**

The rebuttal effectively addressed technical rigor, including statistical significance (Wilcoxon tests), evaluator bias,
 and expanded mitigation analysis.

However, fundamental concerns regarding the dataset's scale, domain imbalance, and its artificial/constrained nature remain outstanding, as these represent core structural limitations of the benchmark.

**Reviewer Scores:**

Reviewer jhxT is likely to increase score (from 4 to 6) as the rebuttal directly addressed their requests for cross-model validation.
J9J3 is expected to maintain a positive rating.

However, 7n4w, m7UK, and RzT5 will likely maintain their scores,  because the core issues of dataset representativeness and the artificial nature of dialogues were not fully resolved through the rebuttal.

---

### Decision · Program_Chairs · 2026-01-26

Reject